# Precursor of disintegration of Greenland's largest floating ice tongue

Angelika Humbert[1,2], Veit Helm[1], Niklas Neckel[1], Ole Zeising[1], Martin Rückamp[3,1], Shfaqat Abbas Khan[4], Erik Loebel[5], Jörg Brauchle[6], Karsten Stebner[6], Dietmar Gross[7], Rabea Sondershaus[8], and Ralf Müller[8]

[1]Alfred-Wegener-Institut Helmholtz-Zentrum für Polar- und Meeresforschung, Bremerhaven, Germany
[2]University of Bremen, Department of Geosciences, Bremen, Germany
[3]Geodesy and Glaciology, Bavarian Academy of Sciences and Humanities, Munich, Germany
[4]DTU Space, National Space Institute, Technical University of Denmark, Department of Geodesy and Earth Observations, Copenhagen, Denmark
[5]Technische Universität Dresden, Institut für Planetare Geodäsie, Dresden, Germany
[6]German Aerospace Center, Institute of Optical Sensor Systems, Berlin, Germany
[7]Division of Solid Mechanics, Institute for Mechanics, Technical University of Darmstadt, Darmstadt, Germany
[8]Division of Continuum Mechanics, Institute for Mechanics, Technical University of Darmstadt, Darmstadt, Germany

**Correspondence:** Angelika Humbert (angelika.humbert@awi.de)

**Abstract.** The largest floating tongue of Greenland's ice sheet, Nioghalvfjerdsbræ, has been relatively stable with respect to areal retreat until 2022. Draining more than 6% of the ice sheet, a disintegration of Nioghalvfjerdsbræ's floating tongue and subsequent acceleration due to loss in buttressing, is likely to lead to sea level rise. Therefore, the stability of the floating tongue is a focus of this study. We employed a suite of observational methods to detect recent changes at the calving front. We found that the calving style has changed since 2016 at the southern part of the eastern calving front, from tongue-type calving to a crack evolution initiated at frontal ice rises reaching $5-7\,\mathrm{km}$ and progressing further upstream compared to 2010. The calving front area is further weakened by an area upstream of the main calving front that consists of open water and an ice mélange that has substantial expanded, leading to the formation of a narrow ice bridge. These geometric and mechanical changes may be a precursor of instability of the floating tongue. We complement our study by numerical ice flow simulations to estimate the impact of future ice-front retreat and complete ice-shelf disintegration on the discharge of grounded ice. These idealised scenarios reveal that a loss of the south-eastern area of the ice shelf would lead to 0.2% increase of ice discharge at the grounding line, while a sudden collapse of the frontal area (46% of the floating tongue area) will enhance the ice discharge by 5.1% due to loss in buttressing. Eventually, a full collapse of the floating tongue increases the grounding line flux by 166%.

## 1 Introduction

The Greenland Ice Sheet (GrIS) has undergone major mass loss since the mid 1990's, with an acceleration of sea level contribution starting in the early 2000's (Shepherd et al., 2020) when outlet glaciers in the south accelerated and retreated (Joughin

et al., 2004; Howat et al., 2008). In the last decade, mass loss has reached northern Greenland with a significant contribution from ice dynamics beside the negative surface mass balance (Khan et al., 2022). Only three floating glacier tongues are left to date (e.g. Hill et al., 2017), namely Petermann Glacier, Ryder Glacier and Nioghalvfjerdsbræ (79°N Glacier, 79NG). The other outlet glaciers became tidewater glaciers.

The largest floating tongue in Greenland is the 79NG (∼70 km length and ∼20 km width, Fig. 1a), draining an ice sheet area of 6.28% containing an ice volume of 0.58 m sea level equivalent (SLE) (Krieger et al., 2020). The floating tongue of 79NG has two calving fronts (see Fig. 1b), one in the north towards the Djimphna Sound (earlier this part of the 79NG was named Spalte Glacier) and one eastern calving front. The latest calving event at the northern front took place in 2020. All calving events still since the 1980's followed a similar pattern, with one lateral rift growing and widening over numerous years.

Parts of the eastern calving front are grounded on ice rises, acting as pinning points (blue areas in Fig. 1b,c denoted with IR) and it is hence an ideal location to understand the impact of ice rises on stabilising the ice front. In the following we use the term pinning points and ice rises as synonyms. The floating ice has an ice thickness of about 80-100 m in the vicinity of the calving front. Variations of the eastern calving front position in the past have not gone beyond an imaginary line between these pinning points in upstream direction (Khan et al., 2014) until 2013.

Calving and basal melting are the predominant mass loss mechanisms of ice shelves and floating tongues. This study is focusing on calving only. The style of calving varies widely and is often governed by the existence of ice rises acting as pinning points (e.g. Thomas et al., 1979; Wang et al., 2022). When the ice mass moves past an ice rise, cracks are forming and generate rifts (cracks that penetrate through the entire ice thickness), which typically grow laterally into the floating ice shelf. Eventually this leads to the detachment of an iceberg. This calving style can be found at many locations (e.g. Berger et al., 2016) and in the following we denote it as tongue-type calving. This style of calving is different from that at the floating tongues in North-Greenland (Hill et al., 2018), where friction at the fjord walls are initiating cracks laterally and ice rises at the calving front are not playing any role.

Crack formation and crack propagation are associated with a variety of mechanisms. One of the most important of them are stress peaks leading to material failure by exceeding the material strength. Another is fatigue failure where cracks occur and grow due to cyclic loading. In glaciers and ice shelves also hydrofracturing is possible, describing the propagation of a crack due to water inside crevasses.

Concerning the deformation of the cracks, three distinct crack modes are known in fracture mechanics (Gross and Seelig, 2017). So-called mode I (opening mode) describes the crack propagation under tensile loading where the crack faces move away from each other perpendicular to the crack faces. Associated is a local stress state at the crack tip with the 1st principal stress (tension) perpendicular to the crack faces and the 2nd principal stress in crack propagation direction. Mode II (shear or sliding mode) is characterised by a relative displacement of the crack faces in crack propagation direction. Here, a local stress state with the direction of maximum shear stress coinciding with crack propagation direction is associated. The respective principal stress directions are tilted by 45°. Finally, mode III (tearing mode) is characterised by a shear loading acting out of plane.

A tongue-type calving style is formed by ice rises inducing laterally dominant shear stresses that initiate mode II cracks (Fig. 2a). Once the ice moved past the ice rises, tensile stresses become large enough for crack propagation as mode I cracks. The floating tongue downstream the ice rises is incised laterally on both sides and eventually one of those initial cracks is reaching a critical limit and propagates further, disrupting the entire ice vertically as well as horizontally and leading to the detachment of an iceberg. This type of calving is still taking place north of 79.5°N (north of IR1, see Fig. 1, Fig. 2a and Khan et al. (2014)).

Tongue-type calving is very distinct from disintegration events where ice shelves or floating tongues are experiencing catastrophic fragmentation events. During these events, a large part of the floating tongue is shattered and a massive number of icebergs is produced in a short period of time (Braun et al., 2009). Prominent break-up events were observed for example at Larsen-B (Rack and Rott, 2004), Wordie (Doake and Vaughan, 1991) and Wilkins (Humbert et al., 2010) ice shelves at the Antarctic Peninsula, but are also identified in geological records of Pine Island Bay (Jakobsson et al., 2011).

In Greenland, there are evidences that the floating extensions buttressing the inland ice flow. The former floating tongue of Jakobshavn Isbræ has disintegrated in 2003 which led to acceleration of the outlet glacier and an increase in seasonality of glacier speeds (Joughin et al., 2012).

The most recent disintegration event in Greenland has taken place at Zacharias Isbræ (Zacharias Isstrømen, ZI, Fig. 1a), which lost the majority of its floating tongue since 2012 (Khan et al., 2014). The glacier experienced an ice flow acceleration and doubling in ice discharge and has turned into a tidewater glacier after this event (Mouginot et al., 2015, 2019). The speed-up is reaching up to 200 km into the inland (Khan et al., 2022). Curiously, the neighbouring 79NG remains almost stable (we use stable as synonym to unchanged) and experienced minor ice discharge increase. Over the last decades (1972-2018) an ice discharge increase of 10% has been observed on NEGIS (Mouginot et al., 2019).

ZI and its neighbouring glacier 79NG are the two major outlet glaciers of the Northeast Greenland Ice Stream (NEGIS), the only ice stream of the GrIS. Since ZI has already disintegrated, investigating the stability regime of 79NG is a first step towards to estimate future sea level projections.

A transition in calving regime can potentially destabilise the calving front and eventually trigger the disintegration of the floating tongue (Matsuoka et al., 2015). Although large calving events are normal mass loss processes and are not considered as catastrophic events, the change in load situation may lead to response in stresses of the inland ice glaciers. A retreat of the floating tongue might imply a reduction of the buttressing exerted on the upstream part and perhaps lead to increased ice discharge. This has been shown in projections of the discharge of Petermann Glacier for an upcoming calving event (Hill et al., 2018; Rückamp et al., 2019). A break-up event of Petermann's floating tongue is found by means of simulations to enhance ice discharge and may be not recoverable (Åkesson et al., 2021, 2022).

Also for Pine Island Glacier, Antarctica, a larger calving event has been inferred to be a significant contribution to the ice flow acceleration since 2018 (Joughin et al., 2021), whereas ice flow modelling showed that a moderate retreat of the ice front has a minor influence on the ice discharge across the grounding line (De Rydt et al., 2021).

Our aim is to investigate how and why calving at the eastern front of 79NG has changed. To this end, we are detecting calving front position, crack formation and propagation, and compare this to principal stresses prior to the crack formation.

**Table 1.** Overview of the data used in this study in respective years. (CF = calving front position, CZ = chaos zone)

| | 1975 | 1980 | 1985 | 1990 | 1996 | 2000 | 2010 | 2013 | 2014 | 2016 | 2018 | 2019 | 2020 | 2021 | 2022 | application |
|---|---|---|---|---|---|---|---|---|---|---|---|---|---|---|---|---|
| Landsat | x | x | x | x | | x | | x | | | | | | | | CF position, CZ, velocity |
| ASTER | | | | | | | x | | x | | | | | | | CF position, CZ |
| Sentinel 2 | | | | | | | | | | x | x | x | x | x | | CF position, CZ |
| TerraSAR-X | | | | | | | | x | | | | | | | | CF position, grounded spot |
| Sentinel 1 | | | | | | | | | | | | | | x | | interferogram |
| ERS 2 | | | | | x | | | | | | | | | | | interferogram |
| Canon | | | | | | | | x | | | | | | x | | grounded spot, Crack E |
| MACS | | | | | | | | | | | | | | | x | Crack E |
| Laserscanner | | | | | | | | x | | | | | | x | | grounded spot, thinning at CF |
| EMR | | | | | | | | x | | | | | | | | ice geometry |
| UWB | | | | | | | | | | | | | | x | | ice geometry, crack depth |

Furthermore, we explore which role thinning of the floating tongue played in the retreat of the calving front. In a final step we will assess which impact a further retreat of the calving front will have on the 79NG's contribution to sea level change by means of numerical perturbation experiments.

As our study is relying on an extensive data basis, we provide an overview in Table 1. The last column of the table is briefly summarising what the data is used for. We mainly use optical, but also radar imagery, as well as SAR interferometry and products such as velocity fields. Airborne data in this study comprises ice penetrating radar, laser scanner (ALS) and optical imagery. For increasing the readability of the text, we present the details of the data processing only in the Appendix A.

The observational datasets are complemented by numerical modelling simulations which aim to assess the instantaneous
velocity response to a retreating floating tongue (Section 3 and Appendix B). Based on this variety of datasets we aim to investigate whether recent changes of the 79NG are indicating a regime change, what exactly is the cause of the changes and how this will impact the stability of 79NG.

The text is organised as follows: We first demonstrate that the calving style has changed, present a fracture mechanical assessment of the floating tongue at the calving front and discuss the evolution of a weak zone that potentially influences
the floating tongue stability. We continue with estimating the impact of future (large) calving events on 79NG's sea level contribution by means of ice flow modelling (details are presented in the Appendix B). Finally, we compare our findings with disintegration or catastrophic calving/disintegration events at other floating tongues and ice shelves.

## 2 Transition in calving regime

Comparing the calving front locations in satellite imagery allows us to investigate whether the type of calving has changed.
Figure 2 displays the calving front in 2000, 2010 and 2020. While in 2000 (Fig. 2a) a tongue exists between the two ice rises and calving is initiated by the lateral rifts downstream of the ice rises IR1 and IR2, the calving front in 2020 (Fig. 2c) is characterised by rifts forming in upstream direction (A and D in Fig. 2c).

Satellite imagery going back until 1975 shows that the tongue-type calving style has been present at least for 28 years until summer 2003. This is in line with the findings of Khan et al. (2014). After 2003, the calving front retreated to a nearly straight line between the two ice rises (similar to Fig. 2b) and remained in this shape until 2011.

Figure 2a indicates that the lateral rifts are first growing towards the centre of the tongue (in across flow direction) between the ice rises, before changing orientation towards the spot marked with a blue star. This rift geometry, leading to tongue-type calving, is also found back to 1975 and icebergs until 2003 have been detached by these rifts, breaking L-shaped/wedge-shaped icebergs off.

Based on the data analysed here, we suggest that between the two ice rises a small grounded spot at the location of the blue star (Fig. 2a) existed until 2013. That spot has become ungrounded early in 2014. Our inference is based on the following observations: The surface structure as seen from the airborne optical camera on 08 August 2013 (Fig. 3a) is consistent with this spot being grounded as the surface elevation is $12\,\mathrm{m}$ higher than the surrounding floating tongue (ALS in Fig. 3b) and the flow velocity is low (Fig. 6d). Evidence for ungrounding of this location comes from different data sources: A comparison of satellite imagery from 2013 (Fig. 3c) and 2014 (Fig. 3d) shows that this surface buckle is moving downstream. Furthermore, ALS elevation data are lacking the dome-like structure at the same location in 2021 (Fig. 4). In 2013 the radargram clearly indicates reflection from a grounded spot, as presented in Fig. 5a. Interestingly enough, ice penetrating radar shows in 2021 at the location that was formerly grounded still thinner ice (see Fig. 5, location of profiles shown in Fig. 1d). This further supports the argumentation that this area was ungrounding. Since 2013 such a dome-like structure was not newly formed, underlining this spot being in a transition to a new state. An episodic change between grounding and ungrounding seems to be rather unlikely based on our data. To conclude, we infer that a grounded spot existed at least for four decades and became afloat early 2014.

Ungrounding can be a result from two processes: (i) thinner ice approaching the shallow bathymetry or (ii) thinning of the ice locally. Repeat ALS elevation data from 2013 and 2021 reveal a difference of the ice surface of about $1.5\,\mathrm{m}$ north and almost no change south of that grounded spot (Fig. 4). We are aware that the surface elevation difference is only slightly above the tidal range, expected to be in the range of $1\,\mathrm{m}$ based on measurements of Reeh et al. (2000), Christmann et al. (2021) and FES2014b ocean tide model (Lyard et al., 2006), however, the thinning of the ice rises alone (left and right in Fig. 4) exemplifies that surface melting was taking place in these eight years.

In addition to the ungrounding, we observe severe changes in the crack evolution at the calving front over the last years/decades. For the sake of clarity, we have denoted each crack with a label from A to F (Fig. 1c). Between 5–7 July 2016 crack A has been formed over a length of about $3\,\mathrm{km}$ and widened since. The shorter cracks B and C were formed two years later in June 2018.

Crack D formed at the southern ice rise between 1–13 March 2019 (3.5 km length). Both, Crack A and D are actually rifts that propagate through the full ice depth. The widening of both cracks over the course of three years is substantial: Crack A from 20 to 180 m between 2016-09-09 and 2019-09-07 and Crack D from 65 to 284 m between 2019-09-02 and 2022-08-20. The evolution of Crack E is more difficult to obtain from satellite imagery: between May to August 2019 it was formed in several episodes from two sides, starting from Crack B and IR2. Although Canon optical data show in 2021 that the northern and southern branch are close to join (shown in high resolution in Fig. A3a marked by a red arrow), there is no indication that they are rifts that propagate through the full ice depth, which we discuss below in more detail. The situation is similar in summer 2022 , which is revealed by high resolution imagery of the airborne MACS camera system presented in Fig. A3b. Crack F evolved in fall 2020 and has also propagated slightly in winter 2021/22. Crack C has extended in early July 2022, while the crack at IR1 just north of Crack A has propagated end of July 2022 towards the junction of Crack A (visible in Fig. 1c). Most importantly, the crack evolution is reaching far upstream of the ice rises, marking the first event of this kind since the observational era at 79NG. Currently, the crack tip locations of the cracks D and F are located about $5 - 7$ km further upstream than the calving front in 2010. Despite not yet being calved off, the rifts are already now intersecting the ice and as a consequence, the stresses in the floating tongue are changing, which will be discussed below.

The crack and rift evolution presented above is a complex process for which we develop next a fracture mechanical interpretation. We investigate the crack evolution and unveil the modes under which the cracks are formed and discuss how that changed in the recent past.

Fig. 6 shows the first and second principal stresses between the ice rises with their corresponding directions calculated from a velocity field of 2014-16 by means of inverse modelling. Furthermore, the figure displays the maximum shear stress and its direction. As pinning points are acting as barriers for the ice flow, their upstream side is characterised by compressive stresses. Consequently, shear zones exist between the compressive stress zone of each ice rise and the main flow between the two ice rises. The maximum shear stress descends towards zero in the main flow area. This setting enabled the formation of mode II cracks, as Crack A is one. Crack A runs into an area with low stresses (principal as well as shear stresses), which is likely the reason for why the crack stops propagating (see Fig. 6).

To identify the mode of Crack D we cannot use any principal stress field, as due to lack of data coverage, we only have a stress field prior to the formation of Crack A and the initiation of Crack A very likely had a large influence on the stress field. We do however observe that the crack faces move apart like it is typical for mode I cracks, as can be seen in Fig. 1c.

Interestingly enough, the direction of crack propagation is neither along surface rivers or lakes nor along remnants of historic crevasses. We only find in the propagation of Crack A one instance, where the running crack is joining for $\sim 200$ m a river shore and deviates from it thereafter again. Crack arrest is not coinciding with rivers or lakes, as we find evidence for cracks to propagate across meandering rivers and lakes, and propagating further. One may wonder why rivers and lakes are not disturbing crack propagation. In comparison with the ice thickness of about 80 m an $1 - 2$ m deep river or $3 - 4$ m (based on ALS data) deep lake is still a minor change in thickness. From our perspective, this surface topography can be seen as a surface roughness, but without an additional stress concentration, a surface roughness alone, is not controlling the propagation of a crack.

The ice penetrating radar data from 2021 (overview of flight lines is presented in Fig. 1d) across Crack E revealed that the crack is not intersecting the ice entirely in vertical direction at the end of July 2021 (Fig. 5, A1 and A2). However, surface imagery (Fig. A3) from the on-board camera in 2021, as well as optical satellite imagery (see Appendix A), show a thin surface crack at the same time (dashed line in Fig. 1c). In summer 2022, we find lakes ponding over this thin line and the surface ponds are not getting drained by Crack E. Forming a vertically non-intersecting crack is characteristic for a fatigue fracture. Fatigue fractures are initiated from cyclic loading situations on short time scales. The principal normal stresses between the ice rises are sufficiently low (Fig. 6a,b) to allow the existence and propagation of fatigue cracks. Here, tides are suggested to be the reason for the cyclic loading situation. To support this, we explore fringe patterns of interferograms from March 2021 (Fig. 7). Remarkably, there are clear hinge zones around IR1 and IR2, despite the fractures intersecting the ice vertically. We suggest that at IR2, the floating ice tongue is pushed intensively against the ice rise, building a solid contact between the floating and grounded ice. At IR1, we suggest that there are still some intact connections between the floating part and the grounded ice of IR1. As a result, the deflection of the ice plate is a relative motion in vertical direction which leads to cyclic loading at Crack E. This deflection is presumably small enough, so that the crack is not yet critical, otherwise the crack would have propagated through the entire ice thickness already.

None of the cracks is a hydrofracture. Hydrofractures are basically crevasses (pre-existing fractures) filled with water, either from surface melt draining into them as on Antarctic ice shelves (e.g. Scambos et al., 2000) or are facilitating supraglacial lake drainage (e.g. Das et al., 2008; Chudley et al., 2019). Crack A-D and F are newly formed cracks that grow vertically and horizontally at once - they are rifts. As an example, Crack D was initiated in March when no melt water was available at all. Also, the newly formed cracks do not follow the remnants of old crevasses, which hydrofractures would have done. Crack E is the only crack that has a potential for becoming a hydrofracture. Formed in 2019 it has so far survived three years without propagating vertically through. Although the floating tongue at the calving front is densely covered with melt ponds in summer, the lakes are small and shallow so the stress due to the water filling the crack has not yet been large enough to initiate hydrofracture. Hence, we conclude that the rifts are not hydrofractures but initiated by stresses (not due to water pressure) exceeding the material strength.

We further consider the potential future evolution of the calving front. We anticipate that calving along the cracks D–F–C–A will detach about $20\,\mathrm{km}^2$ of ice but leave the ice bridge (IB, light purple in Fig. 1b; area $\sim 55\,\mathrm{km}^2$) unaffected. The remaining ice bridge has a calving front on both sides, one facing towards the open ocean to the east, another one towards a bay in the west. The southern calving front produces icebergs of smaller size that are trapped in the bay, encaptured in winter by a seasonal mélange, whereas in summer this mélange opens up. We denote this area chaos zone (CZ in Fig. 1b) and investigate next the evolution of this zone, which we display in Fig. 8. In 1990 this zone consisted of $14\,\mathrm{km}^2$ open area and further $17\,\mathrm{km}^2$ of fractured ice. In 2000 the open area including icebergs was $21\,\mathrm{km}^2$ and about $19\,\mathrm{km}^2$ being fractured. This increased by August 2010 to $38\,\mathrm{km}^2$ and $16\,\mathrm{km}^2$ open and fractured areas respectively. In September 2020, the open area was $50\,\mathrm{km}^2$ with $15\,\mathrm{km}^2$ being fractured. The total area increased by $34\,\mathrm{km}^2$ or 52% within 30 years, with $18\,\mathrm{km}^2$ in the time when tongue-type calving was last observed, which was in 2004. Also the area of the calving front of this CZ has almost doubled from $15\,\mathrm{km}^2$ to

$28\,\text{km}^2$. The extent of the ice bridge in flow direction developed from 1996 $7.5\,\text{km}^2$, 2013 and 2016 $5\,\text{km}^2$ to $3.8\,\text{km}^2$ in 2021, which is half the width in 25 years with 35% of loss in the past five years.

The situation of the ice bridge at the 79NG is comparable to the ice bridge at Wilkins Ice Shelf (WIS) in Antarctica, where a similar bridge was formed between stabilising islands (Braun et al., 2009). The WIS ice bridge had also two calving fronts, one supported by a thick ice mélange, while the other had only open ocean or winter sea ice and thus no support. The ice bridge had a width of $6\,\text{km}^2$ prior to its rupture and collapsed in 2008. The ice bridge at 79NG is narrower than the one at WIS but has the 'advantage', that two smaller glaciers from the south are draining into the ice bridge (see Fig. 1b,d). This is leading to compression, which is visible in optical imagery and ALS elevation as a bulging zone (Fig. 1d).

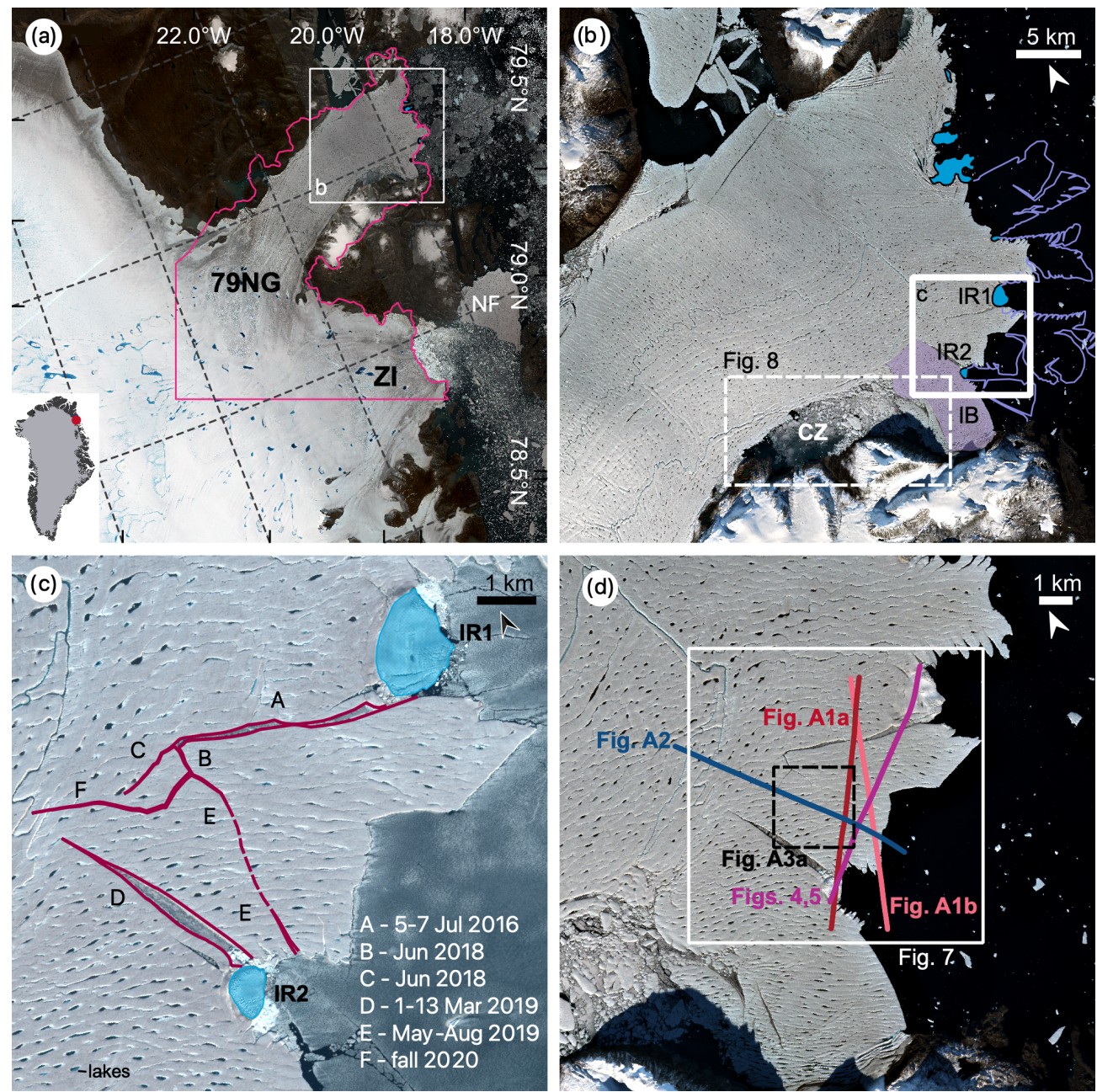

**Figure 1.** Overview of the eastern calving front of 79NG. (a) displays the 79NG and its location in Greenland. The pink line outlines the modelling domain. (b) The eastern calving front of 79NG with pinning points at the ice front shaded in blue. CZ indicates the southern chaos zone and IB marks an ice bridge (see text for details). The boxes indicate subsets used in (c) and Fig. 8. The purple lines represent the calving front from 1980-09-24. (c) Red lines denote cracks as by 2022-08-20 with names A-F mentioned in the main text. The dashed line denotes that the crack does not intersect the ice in the vertical direction entirely. The legend gives the dates or time frame of crack formation. (d) Overview of the UWB profiles displayed in Fig. 5, A1 (in redish color), the ALS profile from Fig. 4, the profile crossing Crack E from Fig. A2 (blue) and the area shown in Fig. 7 (white) and Fig. A3 (gray). The background images are from Sentinel-2: (a) 2021-08-11, (b,d) 2021-09-02 and (c) 2022-08-20.

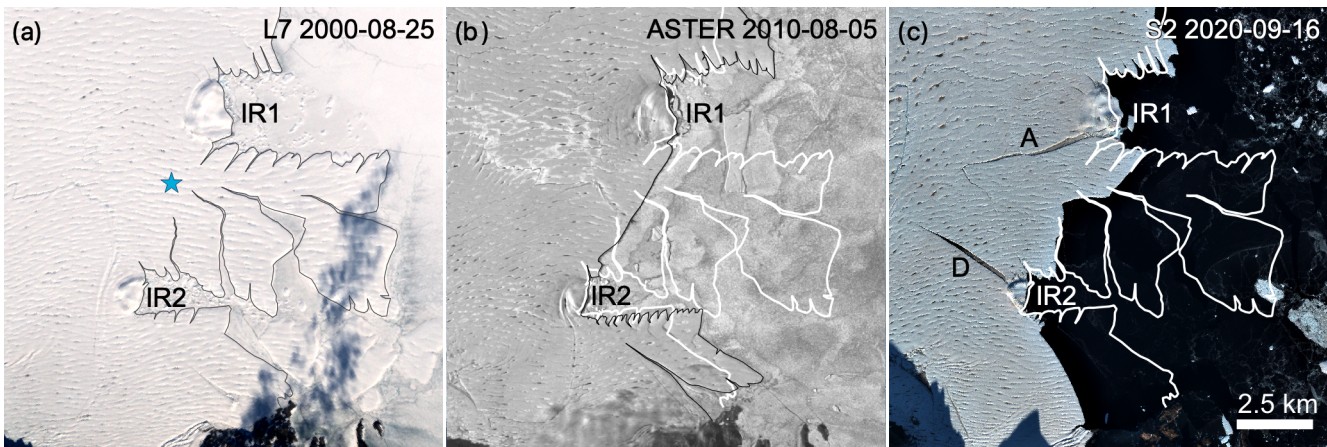

**Figure 2.** Evolution of the calving front and crack areas over two decades based on optical satellite imagery of (a) Landsat-7, (b) ASTER and (c) Sentinel-2 missions, respectively. The blue star denotes an area that has been partially grounded (see main text). In panels (a,b) the calving front is highlighted with a thin black line and in (b,c) the calving front of panel (a) is superimposed in white color.

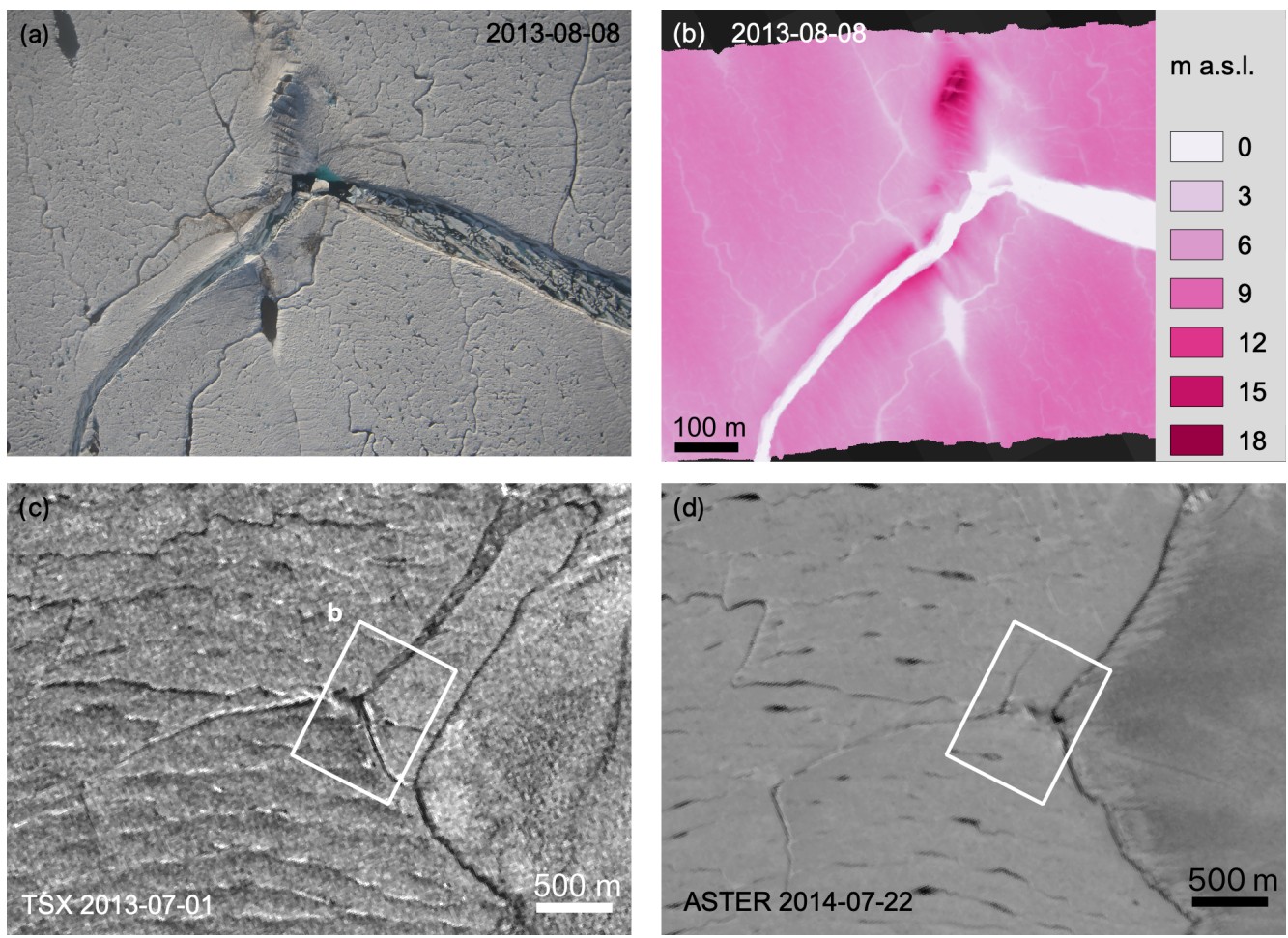

**Figure 3.** Former grounded spot. (a) photo from the Canon camera onboard Polar 5 crossing the grounded spot; (b) ALS surface elevation over at the same area as panel (a). The location of the profile is presented in Fig. 1d. (c & d) satellite imagery during grounding (c) and after ungrounding (d). Panel (c) and (d) are showing the area of panel (b) as an inset.

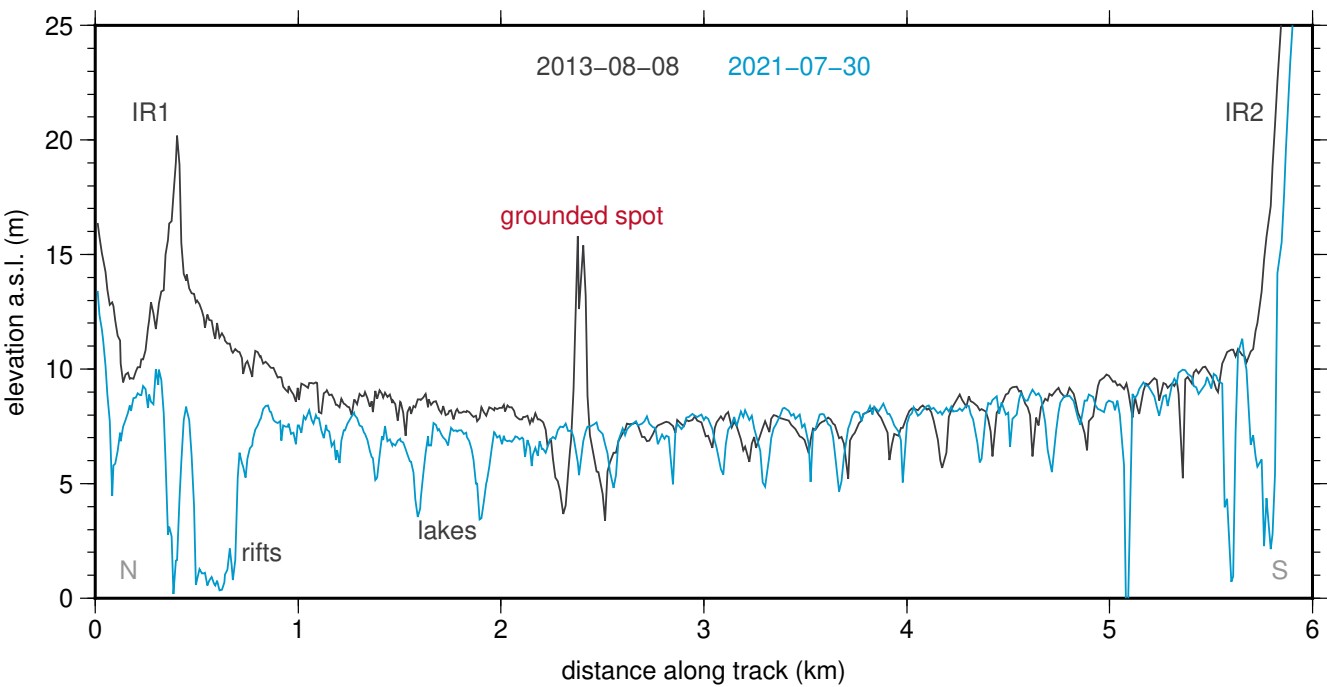

**Figure 4.** Thinning along a profile in the vicinity of the calving front between 2013 (grey line) and 2021 (blue line) based on ALS data. The location denoted with grounded spot corresponds to the same feature in Fig. 3 and the profile location is shown in Fig. 1d. Elevations below 3 m belong to rifts in the vicinity of ice rise margins.

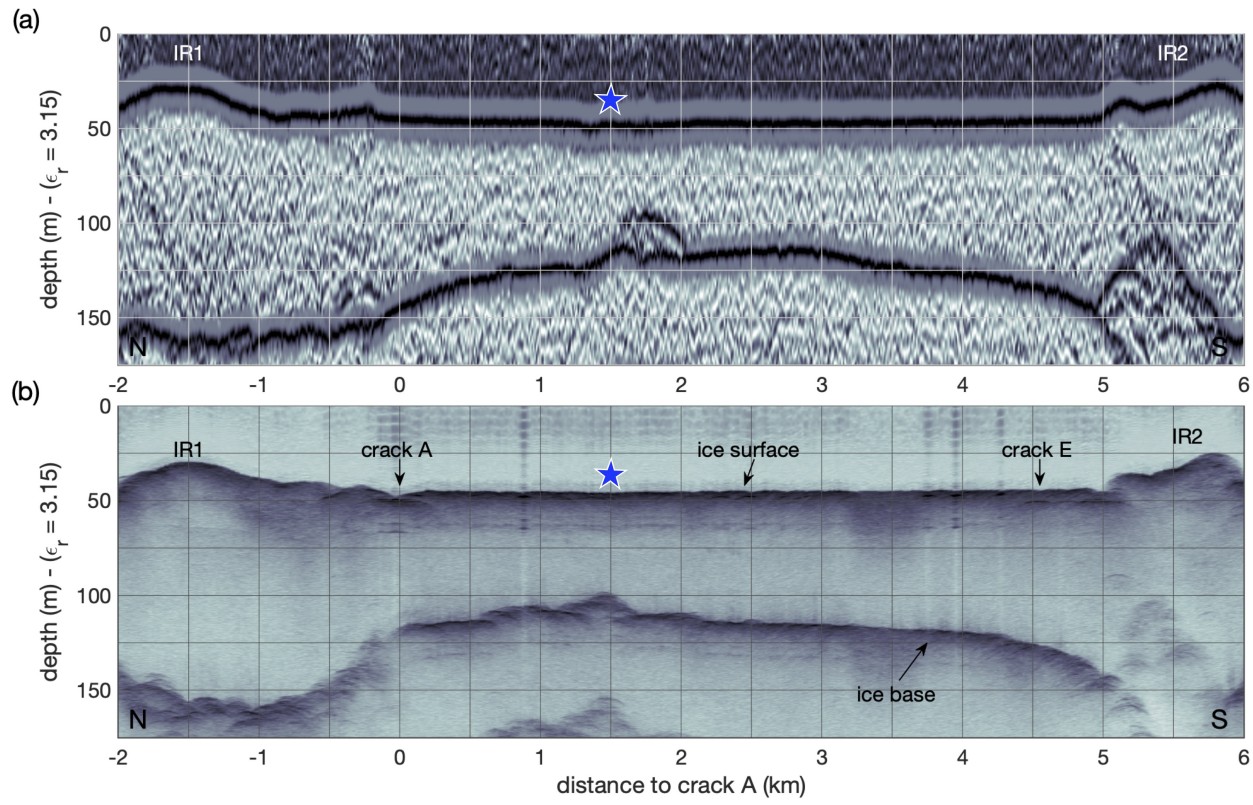

**Figure 5.** Airborne radar echograms recorded with (a) EMR radar from 2013 and (b) UWB radar from 2021 showing crack A and E. The location of the radargrams is shown in Fig. 1d. The blue star marks the location of the grounded spot found in ALS surface elevation from 2013 (Fig. 4).

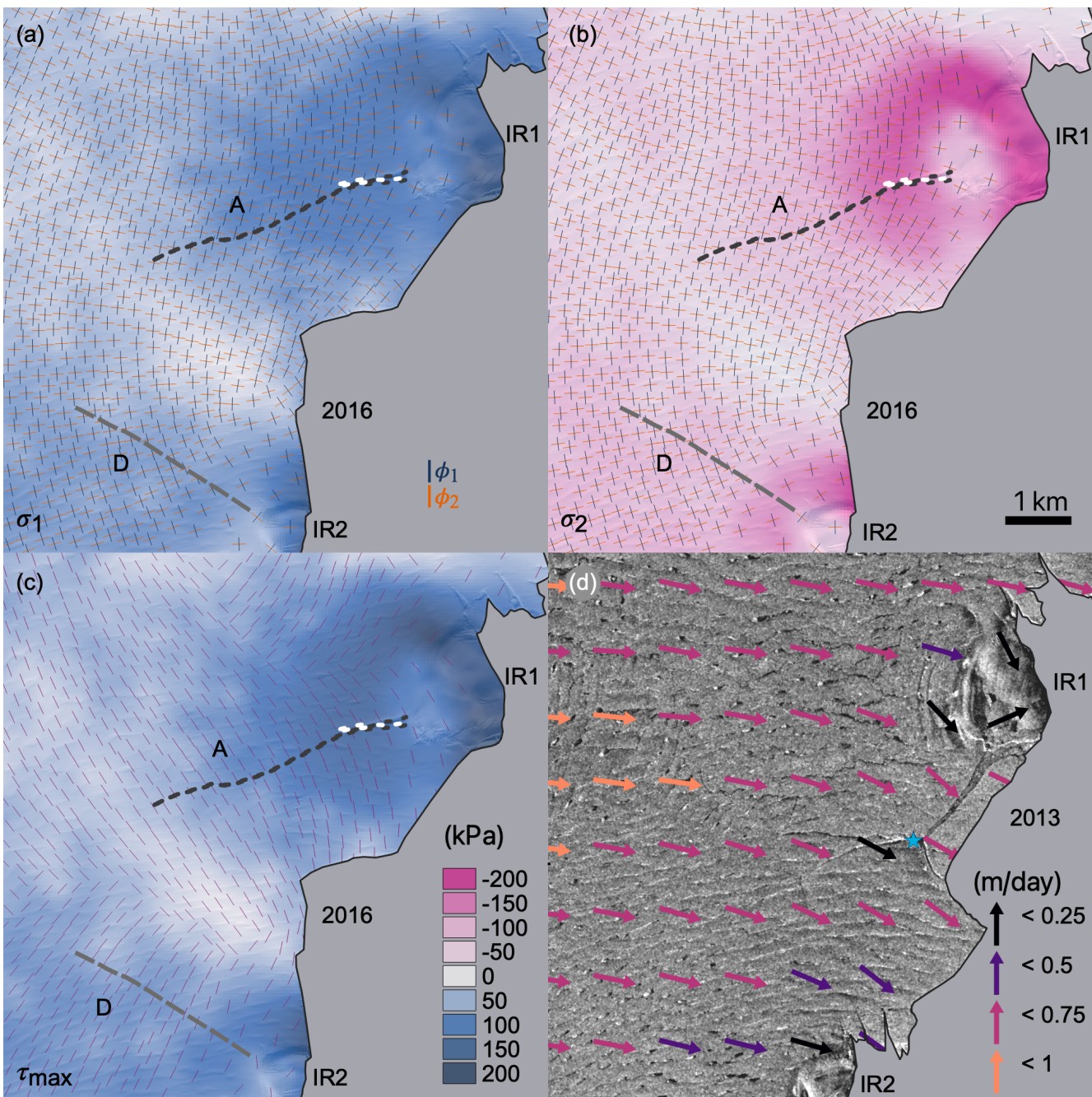

**Figure 6.** Principal stresses in 2016 and velocity field in 2013: (a) first principal normal stress, (b) second principal normal stress, at (a) and (b) crosses indicate principal stress directions ($\phi_1, \phi_2$) (c) maximum shear stress, lines indicate the direction (d) Velocity field in 2013 superimposed on a TerraSAR-X radar image. The blue star denotes the grounded spot. The white dashed line shows the initial Crack A and the black one its length after the propagation in 2016. The grey dashed line represents Crack D right after formation in 2019. The scale bar in panel (c) is valid for panels (a)-(c).

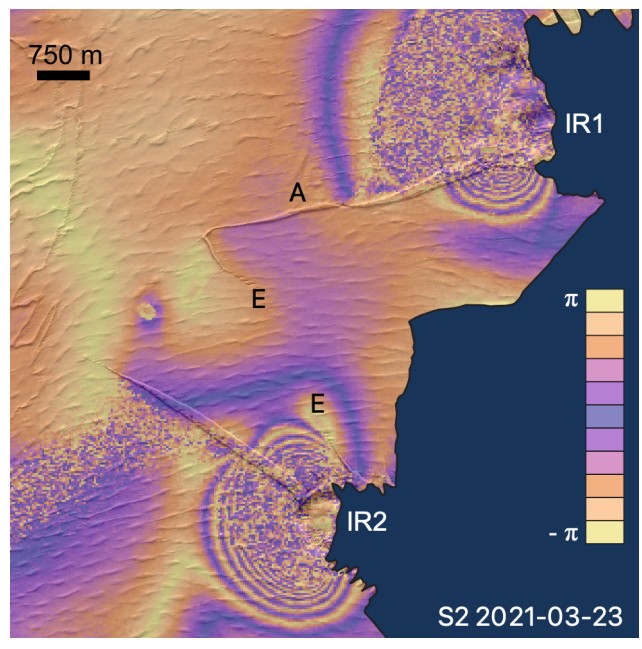

**Figure 7.** Double differential interferogram 2021-03-14, 2021-03-20 and 2021-03-26 based on Sentinel-1 and superimposed on a Sentinel-2 image of 2020-03-23. The color denotes the phase difference.

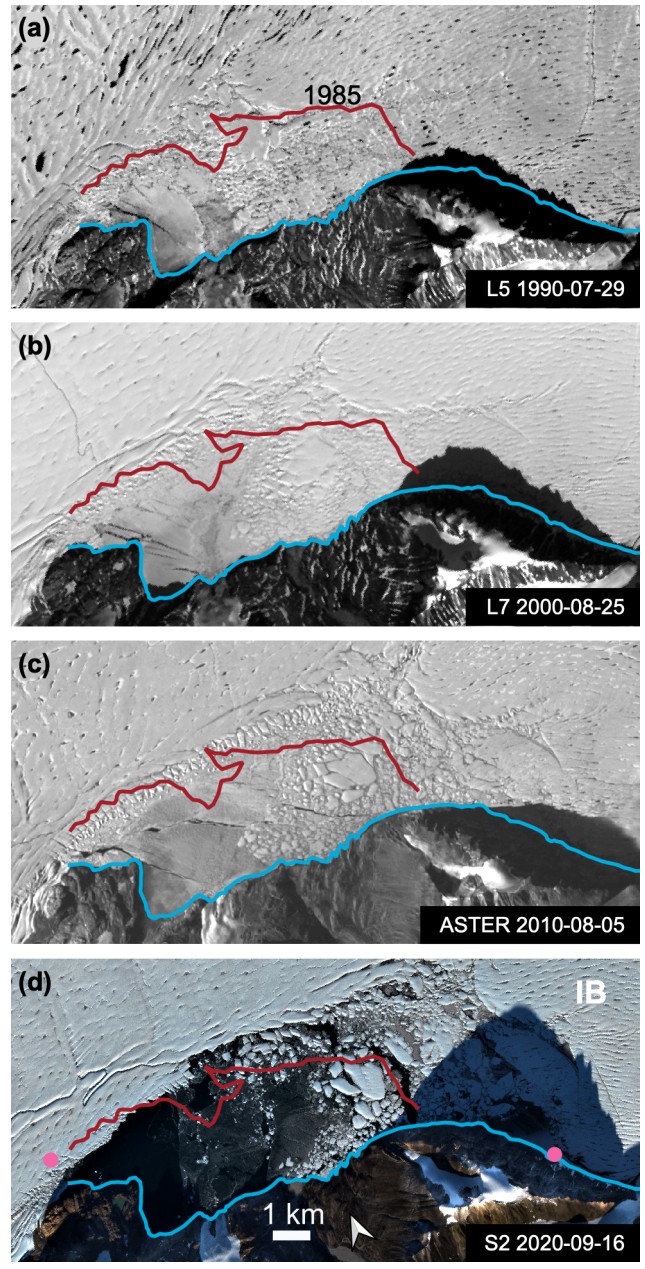

**Figure 8.** Evolution of the chaotic zone since 1990 in optical imagery (Landsat 5, 7, ASTER and Sentinel 2). The blue line marks the shore line and the red line denotes the calving front from 1985. In panel (d), the pink dots mark an area without a hinge zone in interferograms.

## 3 Impact - response of 79NG instability of the calving front

As NEGIS is a fast flowing ice stream and drains a large area of the GrIS (17.23%, Krieger et al., 2020) it has the potential to contribute to sea level rise by an increased ice discharge, once the boundary conditions, such as its calving front, are changing. We address this question, how future large calving events or even a large disintegration event will modify the ice discharge, by using the Ice-sheet and Sea-level System model (ISSM, Larour et al., 2012). According to Rückamp et al. (2022), we employ the full-Stokes (FS) model with a resolution of up to 200 m at the grounding line. We initialise the model using observational data of surface and basal topography as a target to determine the initial conditions by a running a joint inversion for the basal friction coefficient and bulk ice rigidity of the ice (see Appendix B). The modelling domain is outlined in Fig. 1a. The initialisation experiment is denoted *init* and reveals a root mean square error (RMS) of $21.7 \, \mathrm{m \, a^{-1}}$ in the grounded part and $48.9 \, \mathrm{m \, a^{-1}}$ in the floating part (a map view of velocity differences are shown in Fig. B3). The *init* experiment reveals a grounding line flux of $11.9 \, \mathrm{Gt \, a^{-1}}$ (Fig. 9a; note that grounding line fluxes are computed across the main grounding line (white line)).

In order to estimate the back stress exerted by the floating tongue, we calculate a normalised buttressing parameter, $f$, following the method presented by Borstad et al. (2013). Besides geometric and rheologic parameters the buttressing parameter is calculated from contributions of lateral and shear strain rates. In the end, a value of $f = 0$ corresponds to an unbuttressed ice shelf while a value of $f = 1$ represents a fully buttressed ice shelf. Value above 1 represents an overbuttressed ice shelf (i.e longitudinal stresses are negative, i.e. compressive). Up to the bottleneck (NBN to SBN in Fig. 9b) the floating tongue is currently nearly fully buttressed. Downstream of the bottleneck a band of unbuttressed ice exists. The area around the eastern calving front is mainly overbuttressed as a result of the existence of the ice rises.

We are not intending here to simulate future calving events and their impact on the long term behaviour. We are interested in assessing the instantaneous glacier acceleration and increase in grounding line flux to future calving events. Therefore, we perform three diagnostic perturbation experiments in which we remove sections of 79NG's ice tongue. These experiments must not be treated as a snapshot in time of the future evolution, as they do not include any response to changes in oceanic or atmospheric forcing.

The first experiment *calv-iceberg* retreats the calving front position in order to replicate a potential calving event following Crack A towards the front of the chaos zone. The second experiment *calv2fjord* is investigating a larger change in the floating tongue, in which we assume that the calving front retreats up to a point where the fjord geometry is narrowing (denoted southern and northern bottleneck (SBN and NBN), respectively, in Fig. 9b). This scenario mimics a sudden collapse of the frontal part, i.e. 46% of the floating tongue area. A full collapse of 79NG's floating tongue is assumed in a third scenario *collapse*. Next to the simulation results the detached parts are shown in Fig. 10 as purple shading.

With loss of the connection to the two southern ice rises the experiment *calv-iceberg* leads predominantly to ice flow increase (Fig. 10a). In the vicinity of the new developed ice front a speed-up of $118 \, \mathrm{m \, a^{-1}}$ is simulated. Compared to the *init* experiment only the northern tip of the calving front remains overbuttressed, while the central part of the floating tongue is remaining in similar state (Fig. 10d). This scenario shows a grounding line flux of $12.0 \, \mathrm{Gt \, a^{-1}}$, which is an increase of 0.2%. With a further

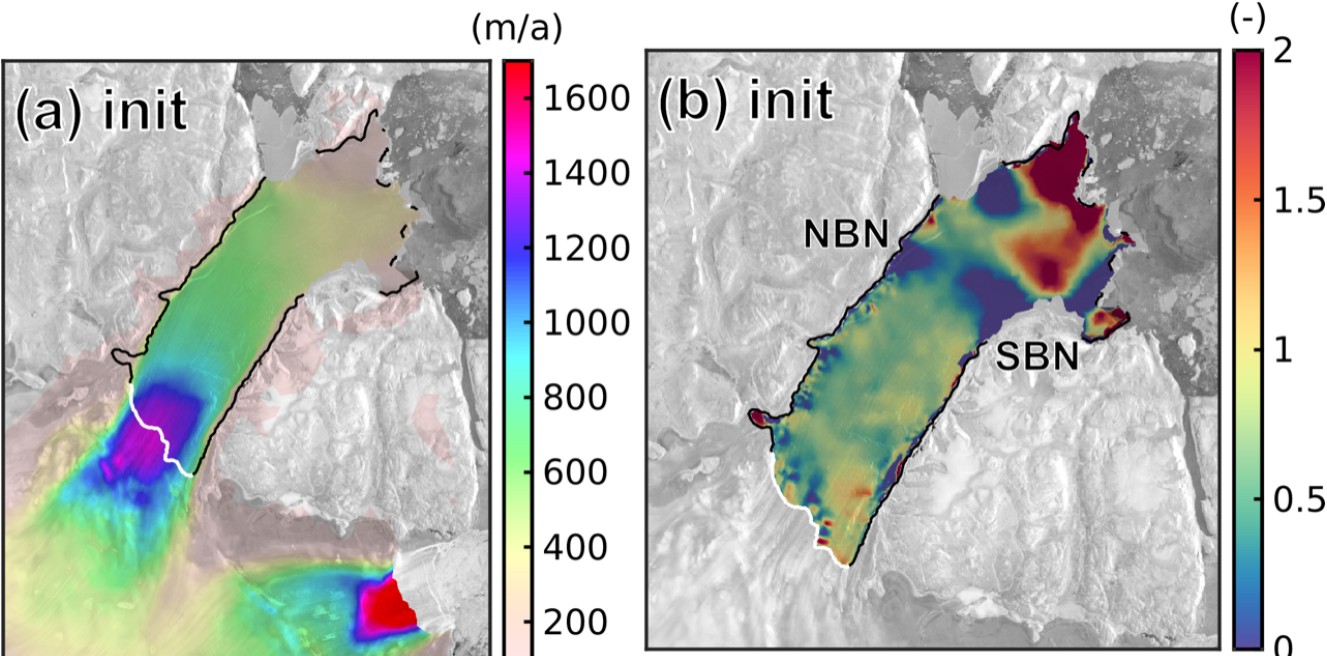

**Figure 9.** Simulation results of the ice flow model for *init* experiments Panel (a) shows simulated surface velocities. Panel (b) shows the buttressing parameter according to Borstad et al. (2013). A value of 0 represent an unbuttressed floating tongue, a value of 1 represent a fully buttressed floating tongue, while a value of >1 represents an overbuttressed ice shelf (i.e. longitudinal stresses are negative, i.e. compressive). The black line indicates the grounding line from the *init* state and the white line the main grounding line at which we compute grounding line flux.

retreat (*calv2fjord*) the situation changes dramatically into an almost entirely underbuttressed floating tongue (Fig. 10e). This comes along with a speed-up of the floating tongue of up to $446\,\mathrm{m\,a^{-1}}$ (Fig. 10b) and an increase in ice discharge of 5.1% ($12.5\,\mathrm{Gt\,a^{-1}}$) compared to the *init* experiment. Eventually, the full *collapse* experiment further enhances the speed-up upstream the grounding line up to $2448\,\mathrm{m\,a^{-1}}$ (Fig. 10c). This leads to 166% ($31.6\,\mathrm{Gt\,a^{-1}}$) increase of the grounding line flux. Comparing the experiments with each other, it becomes evident that there is a substantial difference between *calv-iceberg* and *calv2fjord*. In *calv-iceberg* the floating tongue has still a pinned front, thus the ice is still slower at the calving front than at the grounding line. In the experiment *calv2fjord* there the velocity at the new calving front is much higher, which leads to the highly underbuttressed area near the front.

## 4 Discussion

We start by comparing the situation at 79NG's eastern calving front with other ice shelves and floating tongues that have features similar to the ones at 79NG and have undergone considerable change in which these features played a role. First we discuss the WIS, which has been mentioned above already.

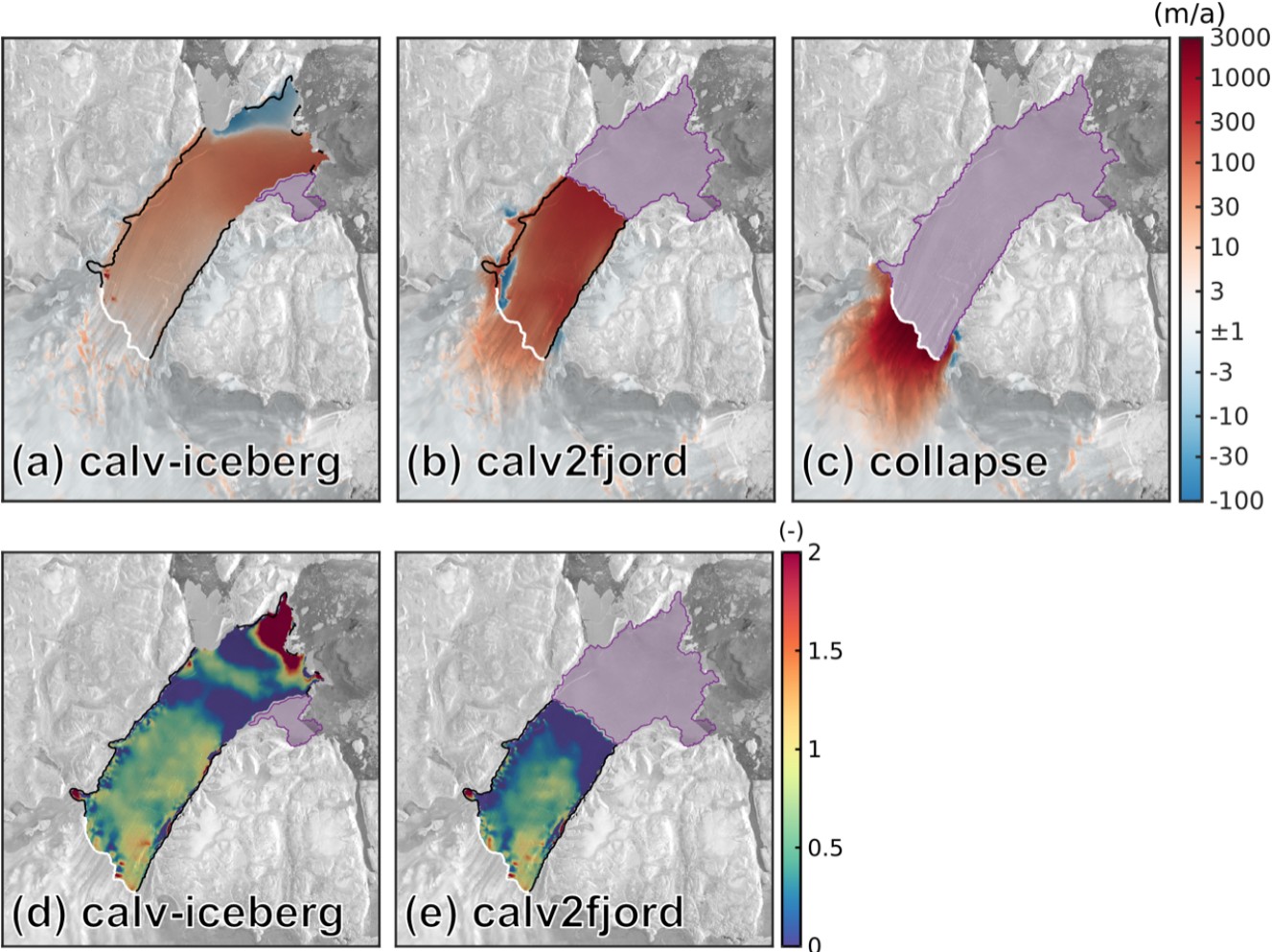

**Figure 10.** Simulation results of the ice flow model for the three perturbation experiments. Panels (a), (b) and (c) show velocity differences of the *calv-iceberg*, *calv2fjord* and *collapse* experiments to simulated velocities from the *init* experiment, respectively. Panels (d) and (e) show the buttressing parameter according to Borstad et al. (2013). A value of 0 represent an unbuttressed floating tongue, a value of 1 represent a fully buttressed floating tongue, while a value of >1 represents an overbuttressed ice shelf (i.e. longitudinal stresses are negative, i.e. compressive). The black line indicates the grounding line from the *init* state and the white line the main grounding line across which we compute grounding line flux. The detached parts for each experiment are highlighted with purple shading.

The ice bridge at WIS was much wider and also the confining islands had a larger size, leading to a broader contact with the confining margins. That ice bridge remained intact in its narrowest form for about 12 month, after which shattering of the ice bridge (Humbert et al., 2010) triggered a sequence of fast crack evolution before today's calving front settled in 2009. This can inform us about the time scale at which we shall expect events at 79NG's calving front to happen. Although 15% of the ice

shelf area has been lost in only 14 months, a new, potentially intermediate, state has been reached that remained now for more than a decade.

Jakobshavn Isbræ (JI) has retreated past a lateral embayment of the fjord and we intend here to investigate if that situation is comparable to 79NG's chaos zone shown in Fig. 8. At JI the embayment had a length of about $12\,\mathrm{km}$, with an ice rise being located at $\sim 5\,\mathrm{km}$ forming a shear margin between the fast glacier flow and the embayment with almost stagnant ice (Joughin et al., 2004). The retreat across this embayment from the ice rise upstream took place in less than 10 months. It is important to note, that the embayment was entirely filled with glacier ice (Csatho et al., 2008), even with buckling along its margin, whereas

at 79NG, the zone is only filled with icebergs and seasonal sea ice, making it by far easier to retreat past this area. Comparing the retreat of the calving front of ZI with 79NG, it becomes evident, that most of the retreat of ZI appeared in a zone that has already been highly heterogeneous already in the 1980's (Thomas et al., 2009; Khan et al., 2014). The northern part of 79NG's eastern calving front is somewhat similar to ZI's north-eastern part (see Fig. 1a, marked with NF), as both are stagnant and show similar surface buckling. In our simulations, this area is overbuttressed at 79NG (see Fig. 10), which one could argue too

for ZI's northeastern part, which is still existing after the massive retreat of ZI's calving front.

     In all three cases, WIS, JI and ZI, the events took place on short time scales, so only days to months after the disintegration has been triggered. This may partly been due to the fact that ice is a brittle material and fractures are propagating very fast (about a third of the speed of sound), but more prominent that ice is responding to change in stress state on short time scales elastically (e.g. Christmann et al., 2021). Once a change in the calving front situation occurs the elastic stress redistribute

instantaneously and can trigger further follow-up events as well as lead to a modified viscous response over month–years, see Rankl et al. (2017) for WIS.

     However, similar to the fact that WIS has not retreated further to date despite high number of annual melt days (Johnson et al., 2022) and JI's calving front retreat and acceleration has slowed down (Joughin et al., 2020), 79NG may retreat in episodes.

     Other glaciers in the North and Northeast, for example Ryder Glacier, Brikkerne, Hagen Bræ and Kofoed-Hansen Bræ show

an even more complex behaviour than we discussed here for 79NG, as they are potentially of surge type (Hill et al., 2017), while no evidence for surges are found for 79NG and ZI.

     Next, we consider different external drivers that might have contributed to the changes observed at the calving front, such as air and ocean temperatures. The changes in air temperature across Greenland have recently been analysed by (Zhang et al., 2022). They present data from the weather station in Danmarkshavn, some $300\,\mathrm{km}$ south of our study area, from 1958–2020,

which shows a step in $2\,\mathrm{m}$-temperatures of more than one degree in the mid 1990's. This makes thinning of the floating tongue at the calving front more likely to arise from increased surface melting, than due to oceanic forcing. Schaffer et al. (2020) found warm water inflow into the cavity of 79NG at $400\,\mathrm{m}$ depth and would thus not get in contact with the ice base at the calving front which consequently also leads to low basal melt rates as shown by Wilson et al. (2017).

## 5 Conclusions

By means of remote sensing data, we detect changes of 79NG's eastern calving front that suggest the onset of destabilisation. Crack evolution is initiated at prominent ice rises and progressing far upstream. We identified crack initiation by shear mode, while crack propagation is in mode I and mode II for Crack D and A, respectively. Interestingly, Crack E is none of these but could be a fatigue crack due to tidal forcing. As these crack patterns are very distinct from normal tongue-type calving we speculate the new crack formations as a precursor of disintegration. Moreover, we present evidence for ungrounding of a small pinning point due to thinning of the ice thickness after 2013. Basal melt rates are presumably small along the calving front, making thinning by surface melting a likely driver for this change. Not only at the eastern calving front changes are ongoing, but also at the southern margin of the floating tongue: in 2022 this margin has a calving front almost double as long as in 1985. The frontal part of 79NG is thus weakened from both sides. An area of ∼5% of the floating tongue is likely to be lost in the near future. Numerical ice flow simulations show that already the loss of such relatively small area leads to an increase of the grounding line discharge of about 0.2% due to the reduction of the exerted buttressing. A sudden collapse of about 46% of the floating tongue further destabilises the glacier and will increase the ice discharge of 79NG's by 5.1%. Eventually, a full collapse of the floating tongue increases the grounding line flux by 166%. Our findings indicate that 79NG is potentially at the onset of a major retreat phase.

*Code and data availability.* The ice flow model ISSM is open source and freely available at https://issm.jpl.nasa.gov/ (Larour et al., 2012, last access: May 26, 2023). Here, ISSM version 4.19 is used. We provide the following data: laser scanner DEM of the region from 2013-08-08 and 2021-07-30, radargrams of the region from 2013-08-08 and 2021-07-30, optical imagery from the onboard camera from 2013-08-08 and 2021-07-30, simulated velocity and buttressing fields, principal strain-rates and principal directions for 2016 Ice flow velocities derived from Landsat are available on the Geodetic data portal of TU Dresden (https://data1.geo.tu-dresden.de/flow_velocity, last access: May 26, 2023, Rosenau et al. (2015)).

## Appendix A: Data

### A1 Satellite data

For the analyses in this paper, we explore a range of different satellite sensors. Optical Landsat-5, -7 and -8, ASTER, as well as Sentinel-2 imagery is used to derive time series of calving front evolution. For this purpose, we use the radiometrically calibrated and orthorectified Landsat Level-1 products provided by the United States Geological Survey and the Sentinel-2 Level-2A products provided by Copernicus, respectively.

Furthermore, we use Landsat-8 imagery to determine ice flow velocity. This is realized through a combined feature tracking approach. We utilize the fast normalized cross correlation as well as a subsequent least squares matching in order to estimate displacement vectors with subpixel accuracy. A detailed description of the processing system is given by Rosenau et al. (2015).

Sentinel-1 synthetic-aperture radar (SAR) imagery is used for detecting calving front and crack positions. For this purpose, the data was first radiometrically calibrated and subsequently a Range-Doppler terrain correction was applied using the GIMP terrain model (Howat et al., 2014). We do not apply speckle filtering on purpose, in order to avoid smoothing of cracks. Mostly we use descending tracks with the relative orbit 170, which we found most useful for detecting crack evolution due to its orientation relative to the cracks. We use one TerraSAR-X scene in stripmap mode as Enhanced Ellipsoid Corrected (EEC) product, that has been reprocessed to 12.5 m resolution with a bicubic spline interpolation.

With the availability of high resolution SAR data with frequent revisiting times, SAR interferometry (InSAR) became an important tool for monitoring the evolution of cracks on floating ice shelves (e.g. Rückamp et al., 2019; Libert et al., 2022). Here, we applied SAR interferometry on Sentinel-1 Interferometric Wide (IW) data following Neckel et al. (2021). In a first step, continuous single-look complex (SLC) images were generated from each set of bursts and swaths. SLC images of orbit 170, slice 2 were co-registered with the help of precise orbit information and the global TanDEM-X DEM gridded to 30 m spatial resolution (Wessel et al., 2016). The accuracy of the co-registration was further refined by employing an iterative offset tracking approach between both SLCs. This is essential in fast flowing regions where phase jumps in burst overlap areas are most common. Two repeat-pass interferograms were generated with data acquired on 2021-03-14, 2021-03-20 and 2021-03-26. Topography induced phase information was removed from both interferograms employing the global TanDEM-X DEM. To minimize the effect of static horizontal ice flow a double differential interferogram was created which is shown in Fig. 7.

### A2 Ice penetrating radar

We used AWI's UltraWideBand (UWB) Multichannel Coherent Radar Depth Sounder (MCoRDS, version 5) onboard of the polar aircraft Polar 5 of the Alfred Wegener Institute Helmholtz Centre for Polar and Marine Research (AWI). The UWB has an array of eight antennas with a total transmit power of $6\,\text{kW}$ and can be operated in the frequency band of $150-600\,\text{MHz}$ (Hale et al., 2016). The radar was operated with a pulse repetition frequency of $10\,\text{kHz}$ and a sampling frequency of $1.6\,\text{GHz}$. We used alternating sequences of different transmission/recording settings (waveforms) to increase the dynamic range: short pulses ($1\,\mu\text{s}$) and low receiver gain ($11-13\,\text{dB}$) to image the glacier surface, and longer pulses (3 and $10\,\mu\text{s}$) with higher receiver gain ($48\,\text{dB}$) to image internal features and the ice base. The waveforms were defined with regard to the thickness of

the floating tongue. In our survey we used a bandwidth of $370\,\mathrm{MHz}$ within the frequency band of $150-520\,\mathrm{MHz}$. Post-flight processing included pulse compression in range direction and synthetic aperture radar focusing in the along-track direction. The final along track resolution was set to $10\,\mathrm{m}$. We assumed a relative permittivity of $\varepsilon_r = 3.15$ in ice for the time-to-depth conversion. The theoretical range resolution in ice after pulse-compression for the chosen bandwidth is about $0.23\,\mathrm{m}$. As there is no thick firn layer, we did not apply any firn correction. We concatenated the echograms of the alternating waveforms to obtain the final echograms covering the ice from the surface to the base with high dynamic range. Fig. 1d gives an overview of the ice penetrating data used in this study.

In addition and to compare to the more recent UWB system (Fig. A1) we used data from the Electromagnetic Reflection System (EMR). The EMR is an airborne radio-echo sounding system used to map ice thicknesses and internal layering of glaciers, ice sheets and ice shelves. The system is capable to penetrate $4000\,\mathrm{m}$ thick ice. It was designed and built by AWI in cooperation with Aerodata Flugmeßtechnik GmbH, Technical University Hamburg-Harburg and the Deutsches Zentrum für Luft- und Raumfahrt e.V. (DLR). The radar signal is a $150\,\mathrm{MHz}$ burst with a signal length of $60\,\mathrm{ns}$ or $600\,\mathrm{ns}$. The maximum performance is $1.6\,\mathrm{kW}$ with a sensitivity of $190\,\mathrm{dB}$ (Nixdorf et al., 1999).

## A3 GNSS

The aircraft's position was measured using a dual-frequency NovAtel OEMV GNSS receiver at a sampling rate of $20\,\mathrm{Hz}$. To determine the flight trajectory we use the Precise Point Positioning (PPP) post-processing option including precise clocks and ephemerides of the commercial GNSS software package Waypoint 8.90. The accuracy of the post processed trajectory is less than $0.1\,\mathrm{m}$ but varies along track.

## A4 Laser scanner

Airborne laser scanner (ALS) data has been acquired in 2013 and 2021 with the laser scanner system (RIEGL LMS-VQ580) and a scan angle of $60°$. The aircraft was flying roughly $300\,\mathrm{m}$ above ground, resulting in a scan width of about $300\,\mathrm{m}$ and a mean point-to-point distance of $\sim 0.5\,\mathrm{m}$. To obtain the final calibrated geo-referenced point cloud (PC) data, the raw laser data was combined with the post-processed GNSS trajectory, corrected for altitude of the aircraft and calibration angles. Crossovers were used to calibrate the system and to derive the elevation accuracy of the final geo-referenced PC to be better than $0.1 \pm 0.1\,\mathrm{m}$. The bias of $< 0.1\,\mathrm{m}$ varies along track and is due to the vertical accuracy of the post-processed GNSS trajectory. The final digital elevation model (DEM) with $1\,\mathrm{m}$ horizontal resolution was derived from the PC by using an inverse distance weighting (IDW) algorithm and a $5\,\mathrm{m}$ search radius. Finally, the freeboard was obtained by reducing the ALS DEM, that has been referenced to WGS84, to the EGM2008 geoid (Pavlis et al., 2012).

Please note that no tidal correction has been applied to the 2013 and 2021 DEM's. The tidal elevation is expected to be in the range of $1\,\mathrm{m}$ based on the measurements of Reeh et al. (2000), Christmann et al. (2021) and the FES2014b ocean tide model (Lyard et al., 2006).

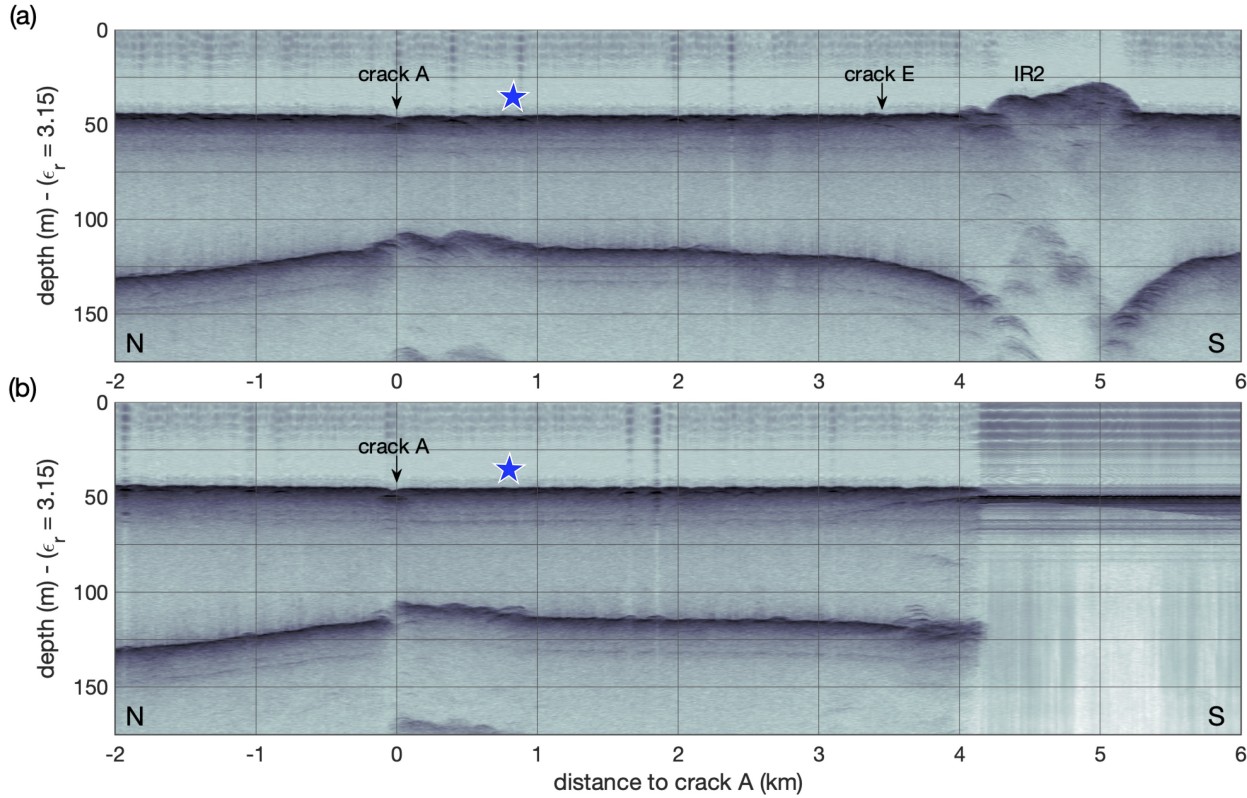

**Figure A1.** Airborne UWB radar echograms from 2021 recorded in study area and crossing today's crack A. The location of both profiles is shown in Fig. 1. The first strong reflection represents the ice surface and the second one the ice base. The blue stars mark the location of the grounded spot found in Fig. 5a and in ALS elevation data from 2013 (Fig. 4).

## A5   RGB camera data

Next to the ALS data a nadir looking CANON EOS-1D Mark III Digital Single Lens Reflex (DSLR) camera in combination with a CANON 14 mm f/2.8L II USM lens is routinely employed on board AWI's research aircrafts. RGB images are acquired at 6 second intervals and are stored together with a GNSS time tag in RAW data format. We selected all images acquired in the vicinity of the crack location and assigned the temporally closest dual-frequency GNSS and Inertial Navigation System (INS) measurement to each image. All images were then corrected for vignetting effects and converted into JPG format preserving the
original high resolution. In the next step we employed the structure-from-motion pipeline of the commercial Agisoft Metashape software (Beyer et al., 2018) to obtain a high resolution DEM and orthomosaic of the 2021 crack area. In order to match the orthomosaic with the timely consistent ALS DEM we coregister both DEMs and employed the derived translation and rotation information to transform the final orthomosaic image. The resulting mosaic is displayed in Fig. A3a.

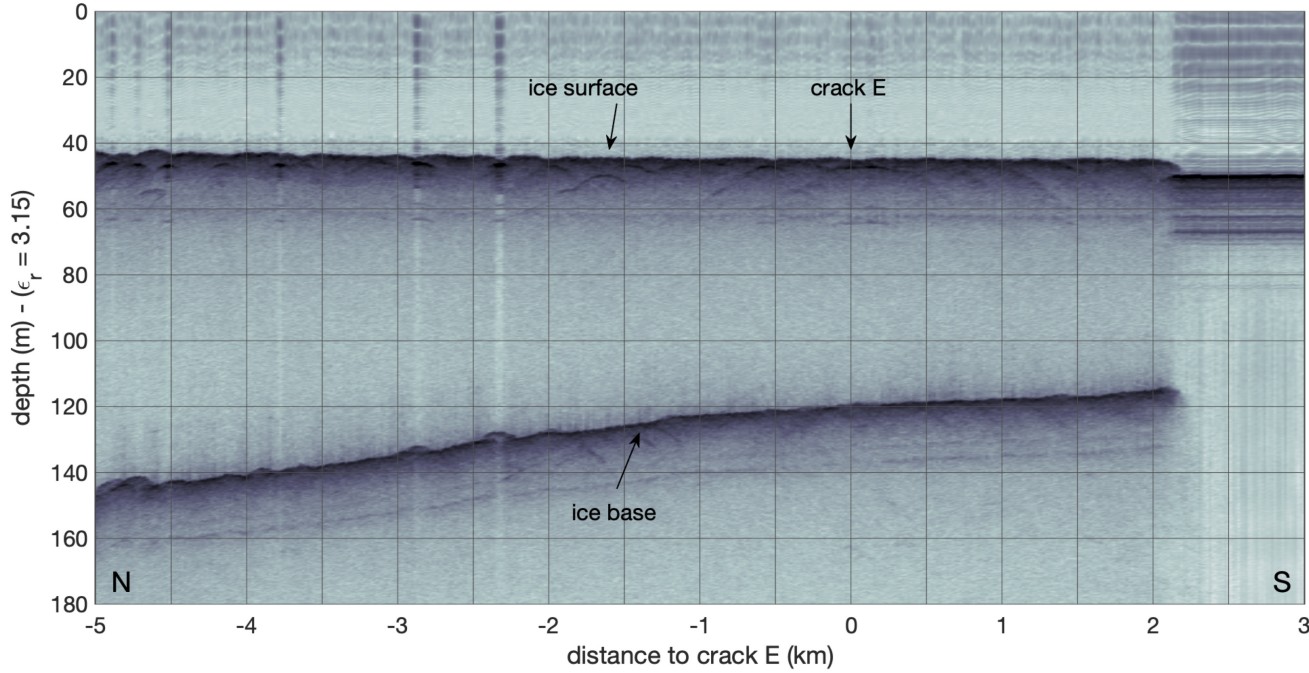

**Figure A2.** UWB echogram from 2021 crossing crack E. For orientation: flight direction is towards the calving front, that is located at 2 km distance from Crack E. The first strong reflection represents the ice surface and the second one the ice base. The location of the profile is shown in Fig. 1d.

## A6 MACS-Polar

The Modular Aerial Camera System (MACS) is a family of optical sensor instruments specifically developed for scientific application in unusual environment. Based on a modular software and hardware design, various project-related demands like geometric and radiometric configurations as well as assembly constraints can be realized. To be carried by AWI's polar aircraft and to be operated in cold regions, a special version named MACS-Polar was established. The sensor head was separated from the control-and-logging unit. Thus, the sensor head consisting of cameras and an inertial measurement unit (IMU) was mounted underfloor in the fuselage while the cabin-mounted data logging unit remained accessible during the campaign. For ice monitoring, a downward oriented sensor configuration was chosen, acquiring images in various optical bands. The red-green-blue spectrum (RGB) is valuable for characterization of melt ponds and for the visual interpretation of the scene by humans. From an altitude of 1000 m above ground level the optical configuration yields to a ground sampling distance (GSD) of 0.15 m in the RGB. In this case the swath width is 700 m. All three cameras take images at the same time and with a continuous rate of up to four frames per second. This enables sufficient along-track image overlap of more than 80% even while flying in very low altitudes less than 300 m. The frame rate is required to achieve a ground pixel resolution better than GSD = 0.05 m. As MACS is a photogrammetric aerial camera, sensors are geometrically and radiometrically calibrated. In conjunction with

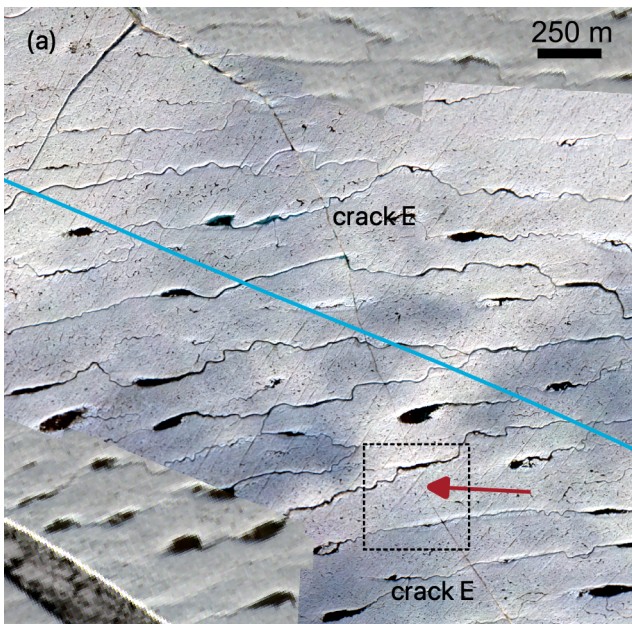 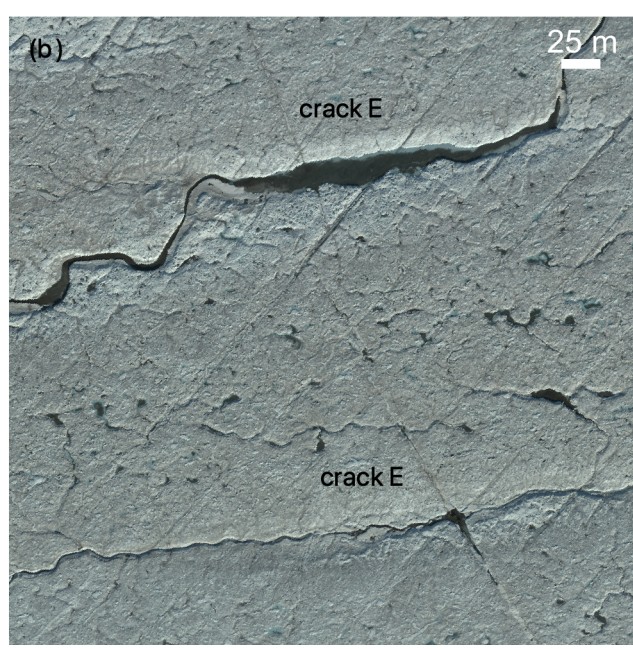

**Figure A3.** (a) Mosaic of the onboard optical Canon camera from a flight on 2021-07-30 superimposed on a Sentinel-2 image of the same date. The red arrow points to the location of the crack tips at the surface. The flight track from Fig. A2 is shown as blue line. Note that the rivers and lakes through which the narrow Crack E has propagated are not drained, pointing out that the crack is not extending through the entire ice thickness. (b) Mosaic of the MACS-polar camera from a flight on 2022-08-20. The area of the is shown in panel (a) as a dashed box.

the inertially aided georeferencing system, all necessary parameters like geographic position, orientation and absolute time of every image's pixel is known. The derivation of a true ortho-photo follows the common photogrammetric approach. First, the aerial triangulation is performed with the initially known approximations of exterior orientation parameters of each image and their a priori accuracies to get the precise and accurate relationships between the individual image coordinate systems and a defined datum and projection. The surface elevation of the glacier is required for the correct positioning of individual color pixels. The resulting mosaic for the crack tip of crack E is displayed in Fig. A3b.

## Appendix B:  ISSM model setup

The ice flow modelling of the NEGIS is conducted with the Ice-Sheet and Sea-level System Model (ISSM, Larour et al., 2012). We employ the full-Stokes (FS) model as it was shown to be most accurate (Rückamp et al., 2022). Model calculations are performed on an unstructured finite element grid with a varying horizontal resolution between 0.2 km and 10 km (Fig. B1). The base mesh consists of 1000 m resolution refined to 200 m next to the main grounding line, over fast flowing ice at 79NG (i.e. $>300\,\mathrm{m\,a^{-1}}$) and at the frontal pinning points. The domain is vertically extruded with 15 layers refined to the base.

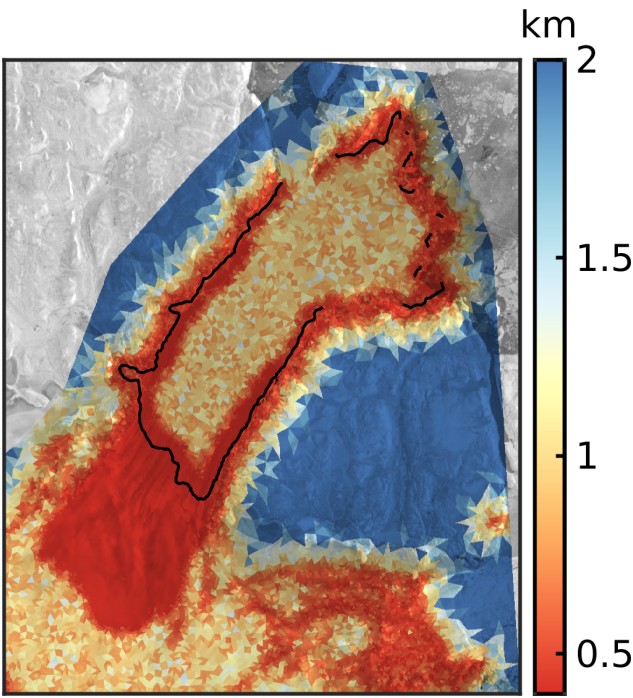

**Figure B1.** Horizontal mesh resolution (km). Data are clipped at 0.4 and 2 km. The horizontal resolution of a triangle is defined by its minimum edge length. The black line delineates the grounding line.

We initialize the model using the present-day ice geometry from BedMachine Greenland version 4 (Morlighem et al., 2017). We reconstruct basal friction, $k^2$, and bulk ice rigidity, $B$, using data assimilation of satellite measurements of surface ice velocity (Fig. B2) (AWI-S1 velocities as in Krieger et al., 2020, but with Sentinel-1 winter (November-March) data from the years 2014 to 2016). Since the surface velocity field has a different coverage as the BedMachine mask, we clipped the calving front to the coverage of the surface velocity field. We use a friction power law (Budd-like) on grounded ice that relates the

basal shear stress, $\boldsymbol{\tau}_b$, to the sliding velocity, $\boldsymbol{v}_b$,

$$\boldsymbol{\tau}_b = -N^{1/m} k^2 \left|\boldsymbol{v}_b\right|^{1/m-1} \boldsymbol{v}_b \tag{B1}$$

with the stress exponent $m = 3$. The effective pressure, $N$, is assumed to be the difference of ice overburden pressure, $p_i$, and the subglacial water pressure, $p_w$, i.e. $N = p_i - p_w$. The basal water pressure is computed in marine parts, i.e. where the ice base, $z_b$, is below the sea-level ($z_b < z_{sl}$), i.e. $p_w = -\min(\varrho_w g z_b, 0)$, where $\varrho_w = 1023\,\mathrm{kg\,m^{-3}}$ is the density of the ocean water.

The viscosity is given by the Glen-Steinemann flow law (Glen, 1955; Steinemann, 1954)

$$\eta = \frac{1}{2} B \dot{\varepsilon}_e^{(1-n)/n}, \tag{B2}$$

with the flow law exponent $n = 3$, the bulk ice rigidity $B$, and the effective strain rate $\dot{\varepsilon}_e$ being the second invariant of the strain-rate tensor.

To avoid having to invert for both bulk ice rigidity and basal friction at the same location, we apply an inversion of bulk ice rigidity to floating ice and basal friction to grounded ice only. We assume that the bulk ice rigidity is a constant value on grounded ice equivalent to a temperature of -5°C (using a constant temperature in grounded ice is a common approach for the inversion of bulk ice rigidity, e.g. Choi et al., 2017; Åkesson et al., 2022).

Within the inverse problem a cost function, $J$, that measures the misfit between observed, $v_{x,y}^{obs}$, and modelled velocities, $v_{x,y}$, is minimised. The cost function is composed of two terms which fit the velocities in fast- and slow-moving areas. A third term is a Tikhonov regularisation to avoid oscillations. The cost function is defined as follows:

$$J_0(\mathbf{v}) = \gamma_1 \frac{1}{2} \int_{d\Gamma_s} \ln \left( \frac{\sqrt{v_x^2 + v_y^2} + \varepsilon}{\sqrt{v_x^{obs\,2} + v_y^{obs\,2}} + \varepsilon} \right) d\Gamma_s, \tag{B3}$$

$$J_{reg}(B \text{ or } k) = \gamma_t \frac{1}{2} \int_{\Gamma_b} \nabla(B \text{ or } k) \cdot \nabla(B \text{ or } k)\, d\Gamma_b, \tag{B4}$$

$$J(\mathbf{v}, B \text{ or } k) = J_0(\mathbf{v}) + J_{reg}(B \text{ or } k), \tag{B5}$$

where $\varepsilon$ is a minimum velocity used to avoid singularities and $\Gamma_s$ and $\Gamma_b$ are the ice surface and ice base, respectively. An L-curve analysis was performed to pick the Tikhonov parameter $\gamma_t$. We obtained excellent agreement to the observed velocities by choosing $\gamma_1 = 1$ and $\gamma_t = 4 \times 10^{-9}$ (Fig. B3).

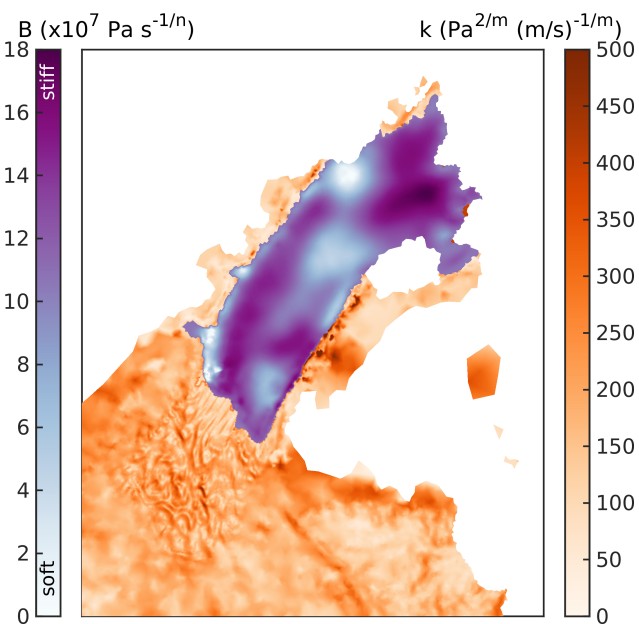

**Figure B2.** Inferred values for the basal friction coefficient $k$ and the bulk ice rigidity $B$ by the joint inversion.

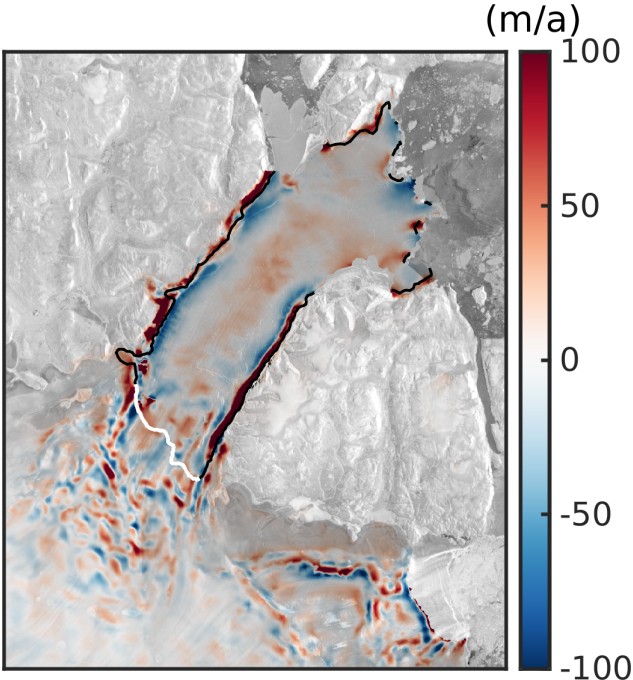

**Figure B3.** Surface velocity differences of simulated and observed velocities.

*Author contributions.* A.H. designed the study, processed and analysed the optical satellite data, M.R. conducted the ISSM simulations, D.G., R.M., R.S., A.H. conducted the fracture mechanical analysis, E.L. contributed satellite remote sensing velocity fields, N.N. computed interferograms and mosaicked the optical airborne imagery, V.H. processed the airborne laser scanner data, O.Z. and V.H. processed the airborne ice penetrating data, S.A.K. contributed findings from his previous studies. J.B. and K.S. processed the MACS data. All authors discussed the findings and contributed to writing the manuscript.

*Competing interests.* We declare that no competing interests are present.

*Acknowledgements.* Parts of this study have been supported by the collaborative project GROCE2 (Greenland Ice Sheet Ocean Interaction) funded by the German Federal Ministry of Research and Education under the Grant No. 03F0855A. E.L. has been funded by the Helmholtz Association Project AI-CORE 'Artificial Intelligence for COld REgions'. R.S. is supported by the German Research Foundation (DFG) under MU 1370/21-1. We want to thank Daniel Steinhage, the crew of Polar 5, Dean Emberly, Marc-André Verner, Luke Cirtwill, Ryan Schrader, the team of Villum Research Station and Station Nord for their support of the airborne campaign NG21. We thank Gerit Birnbaum and Thomas Krumpen for conducting the flight with the MACS-polar system in 2022 as part of the airborne campaign MACSNG.

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
