# Peer review of "Precursor of disintegration of Greenland's largest floating ice tongue"

_The Cryosphere, 2022_

## Referee Comment (RC1)

**Review of Humbert et al. 'Precursor of disintegration of Greenland's largest floating ice tongue'**

**General comments**

The authors combine remotely sensed datasets and numerical modelling experiments to assess the stability of the 79N Glacier ice tongue. Based on an observed switch in the calving style in a region of the terminus between two pinning points, they conclude that the tongue appears to be primed for an imminent disintegration. Using diagnostic modelling experiments, they go on to assess the impact on ice discharge, of removing this near terminus region, and regions of the ice tongue further upstream. From this they find the near terminus region would not increase discharge by much, but that further removal could increase discharge by 8%. This paper is a useful and timely contribution, presenting the evidence for the potential destabilisation of the 79N Glacier ice tongue, and more broadly assessing the impact buttressed ice shelves have on increased ice discharge and sea level rise. In general the paper is well structured and written and the figures are appropriate. I have a couple of general comments outlined below prior to publication, followed by some specific comments line by line.

I read the paper in the order it is written and at several points found it difficult to understand the results presented because the methods are hidden in the appendices. I would suggest moving the methods into the main text, either as a dedicated methods section or as subsections to your observational results and modelling results sections. If the methods are not moved to the main text, at the very least there needs to be better signposting, or summary methods sentences at the start of the results sections to make it clear what data/experiments were used to make statements in the results. I've included some suggestions in the specific comments below. The paragraph at the beginning of Section 3 is very clear, so perhaps a similar paragraph at the beginning of Section 2 (stating the type/date range of imagery used) would be sufficient.

The modelling experiments are well thought out and the second experiment was a useful addition to the first that just removed the region that appears to be about to collapse. However, I did wonder if a third experiment would be interesting, where you remove the entire ice tongue to see what the total/maximum impact of ice tongue loss would be. This would be particularly interesting given that you show that the regions close to the grounding line appear highly buttressed.

**Specific comments**

**Line 3:** I think the link to sea level rise in the future due to ice tongue disintegration could be more convincing here

**Line 5:** Specify what 'recent changes'

**Line 6:** Perhaps state when these cracks progress further upstream compared to 2010.

**Line 17:** Rephrase 'since more than a decade', perhaps 'In the last decade, mass loss has reached northern Greenland...'

**Line 21:** Would be useful to refer to Figure 1 when you state the ice front is in contact with the ice rises. The second half of this sentence is a little vague, can the statement be more convincing, along the lines of 'the impact of ice rises on stabilising the ice front. Consider changing 'object' to 'location'

**Line 23:** You mention basal melting as a mass loss mechanism but do not then discuss it elsewhere in the Introduction. While I appreciate the focus is on calving, I think at least a sentence on melt

rates beneath the ice tongue would be useful, making reference to previous work e.g. Wilson et al., 2017

**Line 26:** I think floating ice sheet should instead be 'ice shelf/tongue'

**Lines 27-31:** Given that a number of ice tongues in northern Greenland have recently disintegrated (ZI), or calved large tabular icebergs (Petermann Gl.), it would be good to include some context for the calving style you are referring to here, and how this compares/contrasts with calving elsewhere in Greenland from floating ice shelves/ice tongues.

**line 35:** Given that Zacharias Isbrae is often also referred to as Zachariae Isstrom it would be useful to include both names here. Also state the time period (decade) over which the ice tongue was lost.

**Line 39:** Add in the size/length of the ice tongue, either here of line 20 where you state it is the largest ice tongue.

**Line 42:** Re-state what style of calving that was. Also where is the evidence for the style of calving in the 1980s, from satellite imagery? that was examined as part of this study? It needs a reference if not.

**Lines 47-50:** Given that you calculate buttressing, and discuss this when conducting your modelling experiments, it would be nice to include a sentence here to state how it impacts upstream flow.

**Lines 51-54:** Before launching into the summary of the methods and imagery used, I think there should be a clearer statement of the research aim and justification. It would also be useful to state the time periods over which the study is conducted, e.g. datasets used for xx to xx

**Figure 1:** Most people may know where in Greenland 79NG is, but I think an inset map of greenland and a box/marker showing the study region would be useful. This inset map could also include the modelling domain used too. Also, include the time period of the crack formations A-F so it is clear when the 'oldest' and 'newest' refer to. Consider adding 'IR1' and 'IR2' labels to panel a) as well.

**Figure 2:** It's quite difficult to see the calving fronts in panels (a) and (b), can you increase the contrast of these images? and perhaps also digitise the calving fronts so that it is obvious where they are.

**Line 55:** Given that the material on the modelling simulations is all in the appendix, I think a clear summary sentence on the experimental design and purpose of these experiments is needed here.

**Lines 68-69:** It is not clear to me where the evidence for the statement that the tongue type calving style has been present for 28 years. It was also not clear from Appendix A1 what the date range of imagery that was used in the study. Did you examine imagery from 1975 in order to make this statement? Please clarify, both here, and in Appendix A1 (see later comment on this).

**Line 71:** Can you instead provide a summary/overarching statement about how the the calving front has changed, instead of using 'tremendously'

**Figure 3:** Some minor suggestions: add scale bars to (a) and (d), or state in the caption they have the same scales as (b) and (c). (b) state the time period of the surface elevation data. Change to 'photo from the Canon camera'

**Line 76:** As mentioned above, where does the evidence for tongue-type calving in 1975 come from?

**Figure 4:** It would be useful to show this track on a study figure or inset map so it is clear where it is located.

**Line 82:** Restate here when it was formerly grounded and when 'recent past' refers to.

**Line 87:** State the tidal range here, including references, or refer to the Appendix section where it is mentioned.

**Line 89:** Consider making Fig.C2 a panel of Fig.1, it would be nice to see the location of the profiles in the main text.

**Line 89:** The second half of this sentence needs rephrasing for clarity.

**Line 90, Figure 5:** As you talk about radar profiles for 2021 and 2013 in this section, I think it would be useful if both appear in the main text rather than the appendix. Why not add a second panel to Fig.5 with the 2013 radargram?

**Line 94:** Can you include a measurement of how much crack A has widened?

**Line 96:** I don't find 'intersect the ice entirely' to be particularly clear, consider rephrasing, perhaps 'propagate through the full ice depth'

**Line 99:** see previous comment related to 'intersect the ice shelf'

**Line 106:** What evidence is this statement based on? the timing of lake formation/drainage did not coincide with changes in the cracks? Please expand on this.

**Line 108:** State why this eastern part is 'interesting'

**Lines 110-113:** I wonder if these sentences on crack modes would be better placed in the introduction when you first use the term 'tongue-type calving style'

**Lines 127-128:** Rephrase these sentences for clarity.

**Line 145:** Explain the characteristics of a 'kind of bridge'. I appreciate you do this on line 148, but I think it would be better to explain the ice bridge at 79NG first, and then refer to the Wilkins Ice Shelf.

**Line 155:** Restate the time over which the calving style changed.

**Line 158:** I'm not sure how useful this comparison to Wilkins Ice Shelf is, given that they have different settings, is there anything to suggest they would be similar in size? Perhaps make it clear why this comparison is necessary. Also, refer to a Figure or the location of glacier draining into the ice bridge from the south on Line 160.

**Line 170:** Given that the methods are in the Appendices, please add a sentence to the main text that presents how your model replicates observations, e.g. the misfit between observed and modelled velocities after the inversion or how the grounding line flux calculated by the model compares to published estimates of grounding line flux/ice discharge.

**Line 175:** It seems that the impact of removing the ice tongue from the ice rises, and even removing it half way up the fjord, was relatively small, <10% increase in ice discharge. Therefore I'm wondering why you stopped there, and didn't follow up with a third potentially 'high-end' scenario in which you remove the entire ice tongue. This would be particularly interesting as it seems from your buttressing maps that the region closest to the grounding line is highly buttressed. This would be complementary to the impact of removing sections of ice tongues/ice shelves close to the grounding line in other regions (e.g. Petermann Glacier Hill et al., 2018 and Larsen C (Mitcham et al., 2022).

**Line 179:** Again, as the methods are hidden in the Appendix (see major comment), there needs to be summary statements or better signposting in the main text. Here for example, it would be better to say something along the lines of 'We calculated ice shelf buttressing following the method presented in (ref) and found that...'

**Line 180:** Explain the term 'overbuttressed'

**Lines 193-194:** Re-state the 14 month period that this 15% area was lost, and the decade over which it as remained in this 'intermediate state'.

**Figure 9:** (a) It looks like the velocities near the northern part of the calving front go off the bottom of the scale? Are they zero here? or excluded from your domain? In (b) can you somehow highlight (e.g. polygon, shading, arrow) the region that has been removed during the experiment. Same for (c). Then refer to this Figure/panels when presenting the experiments in the main text. Panels (d-f): white is not a good colour for the highly buttressed regions given that the background is white, consider changing the color bar. In the caption, state what a value of 2 represents.

**Lines 201-202:** Is there a reference for this statement that the tongue of ZI was highly heterogeneous in the 1980s.

**Line 204:** It is a little bit confusing to refer to locations (e.g. 'northeast part') of ZI's ice tongue without referring to a figure. Consider labelling these locations on existing figure or creating a new figure/panel on existing figure (e.g. Fig.1).

**Line 213:** Can an 8% increase in ice discharge be considered a 'significant contribution to sea level rise'? If so, some additional context/justification is needed. Consider referring to other examples of observed or modelled ice tongue calving/collapse elsewhere in Greenland and Antarctica.

**Line 213:** Change 'is already' to 'would' because the calving in Fig.9c,f has not happened yet.

**Line 217:** What about air temperature changes between 1990 and 2020? Also in the following sentence, make a clearer link between thinning at the calving front and the potential for collapse in the near-future.

**Line 217-218:** I think it would be useful to expand on this discussion about thinning of the ice tongue near the calving front into the context of the rest of the ice tongue, i.e. the oceanic forcing is important for high rates of basal melting elsewhere on the ice tongue and thus responsible for the observed thinning (e.g. Mayer et al., 2018).

**Line 227:** Can you obtain measurements of basal melt rates to clarify this statement 'presumably small'

**Line 243:** You use a number of different satellite senors and I think it would be useful to include a table somewhere of the dates and sensors used. Also include the full time period for which your study covers.

**Line 278:** Why not include Figure C2 in Appendix A? Same for other figures in Appendix C and D.

**Line 284:** 'usually better' by how much?

**Line 311:** It looks like the resolution is also refined based on distance to the grounding line and calving front? If this is the case it would be worth mentioning.

**Lines 329-330:** Is it a common approach to assume the ice rigidity is constant over the grounded areas? Perhaps refer to other studies using this approach/temperature value.

**Line 338:** Rather than state excellent agreement, can you instead include the final value of the cost function, or some measure of the misfit between observed and modelled velocities.

**Technical comments**

**Line 117:** Change 'figure' to 'Fig 6c'

**Line 125:** Add 'the' before 'direction'

**Line 130:** 'at the end' instead of 'in the end'

**Line 136:** 'Remarkably there are'

**Line 156:** Delete 'with changing'

**Figure 6:** (d), there are no arrows shown in light yellow (<10) so this could be removed from the legend.

**Line 188:** Change 'featured' to 'features'

**Line 307:** 'mosaic' spelling

**References**

Hill, E.A., Gudmundsson, G.H., Carr, J.R. and Stokes, C.R., 2018. Velocity response of Petermann Glacier, northwest Greenland, to past and future calving events. The Cryosphere, 12(12), pp.3907-3921.

Mayer, C., Schaffer, J., Hattermann, T., Floricioiu, D., Krieger, L., Dodd, P.A., Kanzow, T., Licciulli, C. and Schannwell, C., 2018. Large ice loss variability at Nioghalvfjerdsfjorden glacier, northeast-Greenland. Nature communications, 9(1), pp.1-11.

Mitcham, T., Gudmundsson, G.H. and Bamber, J.L., 2022. The instantaneous impact of calving and thinning on the Larsen C Ice Shelf. The Cryosphere, 16(3), pp.883-901.

Wilson, N., Straneo, F. and Heimbach, P., 2017. Satellite-derived submarine melt rates and mass balance (2011–2015) for Greenland's largest remaining ice tongues. The Cryosphere, 11(6), pp.2773-2782.

---

## Referee Comment (RC2)

Comments on Precursor of disintegration of Greenland's largest floating ice tongue, by Humbert et al.

Nina Kirchner, December 2022

This manuscript contains a lot of interesting findings for recent observed change at the floating tongue of 79 NG, northern Greenland. For instance, how frontal changes are connected to position of ice rises and ungrounding of previously grounded areas, how tides lead to potential weaking of the floating tongue, and how crack development can be explained in a fracture mechanical framework. Connections between supraglacial melt water features and crack propagation are mentioned, but not considered to have a large impact. Data analysis is complemented by diagnostic numerical modelling focusing on the effects which possible future mass loss due to calving at the terminus of 79 NG will have on its stability.

In my opinion, major revisions are needed before the manuscript can be re-assessed and considered for possible publication in TC. Major points and detailed comments are given below. The manuscript will benefit from language edits throughout, and there is room for considerable improvement in almost all figures.

**Major comments**

The **introduction** is not well structured and not concise (for detailed comment, pls see below). It is essential to better explain the configurational change at 79 NG, and why this motivates to study a regime change.

The section **Transition in calving regime** needs to better streamlined and organized, too, and both figures and text need to be improved (for detailed comments, pls see below). Hydrofracture as a potential link between supraglacial melt features and crack propagation is dismissed, however, with insufficient arguments. The fracture mechanical context employed to describe crack formation lacks context necessary to follow the argumentation. It is essential to rework this section so that the wealth of useful information can be correctly understood.

The **Impact** session suffers from two drawbacks. First, there is repetition as some information is already given in Appendix B and can be removed from the impact session. At the same time, Appendix B is incomplete and would benefit from additional information. Second, the text in the impact section is lacking important detail concerning the justification of model choice, description of results and metrics used, and that needs to be added (for detailed comments, pls see below) to allow for a better understanding of the results presented.

The **Discussion** section appears incomplete and needs to be expanded, for suggestions pls see the detailed comments.

The **Conclusions** reflect the manuscript in its present form, and may need to be revised slightly during the revision.

The **Appendices** appear to replace a full-fledged "Methods" section, which I found a bit surprising for this type of manuscript. Appendix A and B are introduced in the main text at appropriate locations, however, Appendix C and D are not, and actually should not, either, because they belong

to Appendix A. Pls see detailed comments below. Almost all figures need improvements, and Appendix B lacks necessary detail describing the diagnostic model setup.

**Detailed comments (text from the manus is given in italics)**

**Abstract**

**Line 9:** *"are a precursor"* -> may be a precursor

**Section 1 - Introduction**

**Lines 18-19:** provide names of the three floating tongues left. Return to these in the Discussion, see my comment lateron.

**Line 21:** I understand ice rises as features at the surface of an ice mass that are caused by iceflow over a bathymetric high ate the seafloor. Could the text *"Its ice front is still in contact with ice rises…"* be reformulated along the lines of e.g.: Parts of the front are grounded on ice rises, acting as pinning points, …. also because later (eg. in line 24 and in Fig.1) the ice rises are referred to as pinning point.

**Line 23:** *"Calving and basal melting…"* References for this should be added. Also, basal melting is barely addressed later in the manuscript, so it should be mentioned that the focus here is on calving only.

**Line 24:** A ref is needed re *"calving … is often governed by the existence of ice rises"*, and a more nuanced formulation is needed as well. Yes, ice rises could be related to calving if they are in the frontal zone of an ice tongue. But I do not see how ice rises that are far from the frontal zone should be related to calving? Pls clarify.

**Lines 24-25:** *"When the ice mass moves passes by …"* -> When the ice mass moves past …

**Line 25:** *" … rifts (cracks that are separating the ice entirely…."* -> rifts (cracks that penetrate through the entire ice thickness)

**Line 26**: *"which grow laterally into the floating ice sheet"* -> which may also propagate laterally.

**Line 27:** *"This calving style can be found at many locations…."* Pls provide references.

**Lines 27-28:** *"Tongue-type calving is a normal process and is very distinct from break-up of disintegration events"* -> Shorten to: Tongue-type calving is very distinct from disintegration events

**Line 29:** *"During these events, a large part is shattered…"* -> During these events, a large part of the floating tongue is shattered.

**Lines 30-32:** In addition to modern disintegration, there is also evidence of similar ice shelf collapses in the past that should be mentioned, see eg. Jakobsson, M, et al. (2011). *Geological Record of Ice Shelf Break-up and Grounding Line Retreat, Pine Island Bay, West Antarctica*. Geology 39(7), pp. 691-694, doi:10.1130/G32153.1

**Line 35:** Insert space before "(ZI)"

**Lines 39-44:** Several issues here. 1/ This info would fit better in the context of the description of 79 NG that is initiated in line 19 ff. 2/ Also, some basic numbers (length and thickness of floating tongue, depth at grounding line etc) would be useful to have upfront around lines 19 ff. 3/ line 41: remove "act as pinning points", has been said above (eg. line 24). 4/ "…. *Calving event ... in 2020, still in the same style as in the 1980's*": this is not helpful because the "1980's style" isn't described (yet). 5/ Incorrect grammatical reference in "*Changes of calving front … style… have not gone beyond the line of these pinning point*" (how can a style go beyond a line?) 6/ "Line of pinning points" should be reformulated. 7/ Incorrect tense: "A transition … can…destabilize … and eventually triggering " -> A transition ..can.. destabilize … and eventually trigger.

**Lines 45-50:** Consider combining this paragraph with lines 33-38. Explain "*load situation*". Reformulate "*response … in stresses ... of glaciers*". Mention "Antarctica" in connection with Pine Island Glacier. That Petermann is in Greenland is known once you introduce the glaciers name in the context of lines 18, see my comment above.

**Lines 51-52:** "*We leverage…of the floating tongue in the vicinity …*" -> Here, we leverage ... of the floating tongue of 79 NG in the vicinity….

**Line 55:** *"… are complemented by numerous numerical "* -> complemented by numerical ….

**Line 56:** What do you mean by recent changes of 79NG configuration? From line 42 we know that something changed CE 2020, but not really what exactly. What exactly is the configurational change that motivates to study a regime change? This needs to be better explained, see my major comment.

**Line 60:** Decide whether to refer to 79NGs tongue as a tongue or an ice shelf. This needs to be checked throughout the manuscript.

**Figure 1:** Several issues. 1/ Pls provide an overview map of Greenland with the location of 79 NG marked. 2/ Caption (a): "*IB marks an ice bridge (see text for details)."* -> add more precisely where in the text, eg. Section 2. 3/ Panel b: Pls provide some context for Fig 1b already in the introduction. Currently, the reader needs to wait for this until Section 2. Label (b) and scale bar are poorly visible. Consider rendering them in different color and at position with better contrast. 4/ Panel b: Could cracks A-F be colored with individual colors? Explain in the caption what a stippled line vs a solid line means for the crack. Also, cracks are referred to as rifts in Fig 2. Decide on one terminology. 5/ Panel b: I guess IR1 and IR2 stand for IceRise1 and IceRise2. In panel a, they are referred to as pinning points. This should be homogenized, alternatively, say that you use the terms as synonyms. Consider adding the outline of the blue ice rises in panel a to panel b - may be useful for orientation. 6/ Add something about the supraglacial lakes.

**Section 2 – Transition in calving regime**

**Figure 2:** This figure shows changes in the frontal configuration 79NG. Pls consider modifying the figure such that configurations (outlines, prominent features) from 2010 and 2020 are overlain onto the 2000 configuration. This will make the comparison more visually straightforward. A central flowline could be added, as you refer to it later on (eg. line 72).

**Lines 64-74:** Generally, the text here would benefit from streamlining and better organization. A simple guideline could be the chronology of events/calving front changes, from 1975 to present. Specifically, I have the following comments. 1/ Line 64: In this generality, I don't think this is correct.

One can eg. not say anything about changes in subaqueous calving based on calving front positions from optical satellite imagery. I think this needs to be reformulated.  2/ Ensure easy matching between text and abbreviations used in figures. Replace "*between the two ice rises*" -> between the two ice rises IR 1 and IR2. 3/ Lines 66-67: *"Calving is initiated by the lateral rifts":* It would be helpful to indicate the rifts in Fig 2. Pls see comment above re Fig 2. Length of rifts in 2020 need not be given in the text if rifts A and D are indicated in Fig 2. 4/ Line 70-71: "*From the mid 2010's onwards, the calving front situation has changed tremendously*" – How does that go together with line what is stated in line 43: "*calving … took place in 2020, in the same style as in 1980*". Is the 2020 event an exception from the new, post 2010 state? Pls clarify.

**Lines 75-91:** These paragraphs link to Figs. 3  to 5 and focus of the floating tongue becoming ungrounded at a certain location. I have several comments. 1/ "*Based on our database*" -> Based on the data analysed here. 2/ "*between the two ice rises a small grounded spot…*"  What you are saying is that there is another ice rise, at the location marked by a blue star, correct? The formulation is not clear. Why don't you refer to this grounded area as an ice rise? Does an ice rise have to be of a certain size? Other criteria? This is confusing, pls clarify. 3/ "*Our assumption …*"  do you mean: our suggestion? To avoid repetition, you can replace "suggest" by "propose" in line 75. 4/ The text and **Fig. 3** don't go very well together.  Starting point is 8 Aug 2013 (panels a and b). Pls add length scale to panel a and date to panel b, or combine them into one panel. Legend in panel b is m asl, pls add. Panel c has date 1 july 2013 so is showing a state prior to the one shown in panel a. When panel c is mentioned in line 80, no date is mentioned but it is natural to assume that it is a date after august 2013, which causes confusion. In line 80, reference is made to panel d, providing evidence for the ungrounding which is mentioned in line 76, however, without referring to panel d. Rather, because the sentence continues timewise with describing 2021 situation, there is risk that confusion arises (again) as to what actually is shown in panel d (obs scale is missing) and whether it is really necessary (from the caption to Fig. 3, it appears to provide geographical context only?). Also, it would be helpful if current Fig. 6d would be combined into current Fig. 3. Pls review Fig. 3 and caption and harmonize with the text. 5/ Line 81: "*Since 2013 such dome-like structure…*" -> Since 2013 such a dome-like…. 6/ Line 82: "*An alteration…*": Wouldn't that fit better in the discussion section? Alongside an explanation why? 7/ Line 83: Grammar: become –> became.  8/ Line 84: "*Ungrounding can be a result from two instances:*" Ungrounding can results from at least two processes. And yes, indeed, at the ice rises, a lowering of ice tongue surface elevation must come from surface melt or dynamic thinning. However, at floating locations, even basal melt should reduce to surface elevation lowering because reduced buoyant forces cannot sustain the same ice thickness above the waterline? You have mentioned basal melt in line 23 but don't seem to pick up on it any further? Pls clarify. There are some remarks later in the discussion re suspected low basal melt rates where the issue could be discussed nicely in appropriate detail. 9/ **Fig 4.** Please provide an inset to the current figure showing where the profiles are located.  The calving front position of 2013 is nowhere shown, and the 2021 position could be inferred from Fig 1.a, but it gets cumbersome if this becomes a to-do for the reader. Why are supraglacial lakes included in the figure? If they were introduced in Fig 1, it would be easier to see a red thread (since lakes are mentioned later, too). What about lakes in 2013? Pls provide sufficient information regarding Fig. 4. Caption: Pls check grammar. A location cannot correspond to a Figure. 10/ Lines 89-91: Could this be better placed in line 82, before "*we conclude*"?  11/ Fig. 5. Pls add lines showing the approximate location of the underside of the floating tongue, and the seafloor.  Same applies to Figs. C1 (a-c).

**Line 92**: "For the sake of …" This is repetition from Fig.1 and can be replaced by a ref to the Fig instead.

**Line 95:** *"lower ice rise"* Do you mean "southern"?

**Line 98:** "*unification*" -> do you mean that the northern and southern branch of crack E joined? Pls reformulate.

**Lines 103-106:** "*None of the cracks is a hydrofracture*". How can you be sure? This is quite a strong statement, and the simple mention *"we did not find evidence"* is in this brevity not a satisfactory argument. What did you do to arrive at you conclusion? You have bothered to indicate lakes in Fig. 4, why? "either from surface melt of facilitation supraglacial lake drainage": Even the latter are related to surface melt. Pls reformulate and check grammar, and also add a ref to at least Fig. 1b.

**Line 108:** *"at 79 NG's interesting part …."* -> remove interesting and reformulate sentence.

**Lines 109 - 124:** Several issues here. 1/ line 109-111: Provide a reference. Why mention mode III – it is not relevant for the following? 2/ Lines 111-112 and Fig 6: Explain briefly how shear and tensile stresses are related to mode I and mode II, and to first and second principal normal stress and maximum stress mentioned only in the caption to Fig. 6.   Otherwise, this part and the following discussion cannot be understood. 3/ Why do ice rises induce shear stress and why does tensile stresses increase downstream of the ice rise? How is this related to velocities? This must not be left to the reader to find out but needs to be explained. 4/ Line 114. Grammar and terminology. What is an unstable crack and can a crack (a void) detach an ice berg? The latter is detached from the tongue and crack propagation accelerates – is that what you mean?  Reformulate.  5/ Line 117 ff: *"Velocity field of 2014-2016"* Why a two-year averaged one? From where are these stresses? From ISSM? This needs to be explained. Line 129: "descend" -> decreases. 6/ Fig 6. Add to the caption that the scale in panel c is valid even for panels a and b. I think that would be helpful. Consider having panel d in Fig 3 instead, see my earlier comment. Pls show the floating tongue in panel d  a lighter grey so a better contrast to the arrows and the star is achieved. 7/ Did you provide evidence for your statement that (line 121) *"Loss of contact to the grounded spot is leading to an increase in main flow"*?  If yes, please refer to it, if not, pls add and/or explain. Pls also check grammar in lines 121-124, there is a mix of tenses that should be checked for correct use.

**Lines 125 – 140:** Here cracks and supraglacial water and fatigue failure is discussed. Several issues: 1/"River" – has nowhere been introduced. You should mention (eg. in the context of Fig 1b) that supraglacial lakes can be connected by supraglacial streams. Is it "stream" or "river"? There is a lot of literature on supraglacial hydrology at the Greenland Ice Sheet so it should be easy to find the correct terminology and add a couple of references. 2/ *"de-watered*" -> drained. Also in Fig. D2. 3/ Add a line showing the base of the floating ice tongue to Figs 5, D1. Otherwise it is hard to see that the crack is not intersecting the entire ice thickness. 4/ *"Forming a non-intersecting crack".*  I don't understand the terminology chosen. Crack E is rendered as a stippled line in Fig 1b, with individual ends propagating towards each other. In Fig. D2 their expected joining location is highlighted (btw how did you predict this location? And pls add a north arrow to Fig. D2) So why "non-intersecting"? And how is that linking to the fatigue fracture? I understand the latter, but not how the connection to the not-yet joined crack is made. 5/ Maybe it would be better to group this part as section on supragglacial meltwater features and cracks (lines 125-133) and a section on oceanic forcing of crack propagation (lines 133-140)? 6/ Pls add missing information to the caption of Fig. 7. What is ranging between -pi and pi (if the symbol is pi?)   Pls add a pointer to cracks A and E. 7/ Line 137: *"The ice plate"* do you mean the floating tongue?  I don't fully understand why you need to argue that grounded and floating ice are connected at the ice rises (lines 136-139). Because you have fractures around the ice rises? Isnt it sufficient that the tides work on the floating part? Pls clarify.

**Lines 141-143:** This is a bit unconnected, pls consider moving, providing additional context, or removing.

**Line 145:** *"detaching…."* -> reformulate eg as: will detach about 20 km^2 of ice but leave the ice bridge (IB, Fig. 1a) unaffected.

**Fig 8 and caption.** Use either chaos zone or chaotic zone, it is not homogenous in the text. What does the red line denote? Add north arrow. Panel d shows locations for which no hinge zone is visible in interferograms. Are the latter shown? What is the meaning of the easternmost interferogram location – it is in the ocean?

**Lines 152-157**. Consider providing this information as a diagram (staples, or similar). This provides a more attractive means of conveying the info than in repetitive text.

**Line 158-159:** Add info on WIS IB to line 147 instead (and remove here).

**Line 159-161:** Lambert Land has not been introduced. Perhaps just remove? Sufficient that two glaciers drain into the 79 NG from the south? Do you show evidence of the bulging zone?

**Section 3 Impact – response of the 79 NG instability of the calving front**

**Lines 166-167:** "*We attempt …discharge*" -> This is repetition from line 163. Shorten and mention ISSM directly, eg: We address this question using ISSM (add ref to App. B). Then: continue to describe the three experiments (line 170 onward).

**Line 167-170:** Repetition from Appendix B. Remove.

**Lines 170-178:** Here, the three experiments should be described, as well as model output variables and metrics used to derive conclusions, eg. the buttressing parameter that is so far only mentioned in the caption to Fig. 9. Pls justify why you run diagnostic experiments only when the temporal evolution of the changes at 79 NG clearly are in the overall focus. Pls also explain why a Blatter-Pattyn (BP) model was favored over a Full Stokes (FS) model, when in a recent study of the NEGIS, , 'considerable differences at the grounding line' were found and that 'results from non-FS models should therefore be viewed with caution' (tc.copernicus.org/articles/16/1675/2022/). Maybe the error is systematic and so not so important in terms of % increases in discharge, which is the main takeaway in the following. But, I would suggest that either running an FS simulation as a comparison or including a discussion of the potential errors / motivation of the model set-up is necessary. Moreover: 1/ Line 171: Remove number concerning grounding line flux for the init run. 2/line 175: "Bottleneck": Do you mean in term of the fjord geometry narrowing? Pls explain better. 3/ Line 176: Incorrect Fig ref, needs to be corrected to Fig 9f. 4/ Fig 9: parts of the floating tongue that are removed in the two perturbation runs should be rendered hatched (or otherwise highlighted) in panels b,c,e,f. All panels can be cropped to instead focus on the relevant model domain. Insert a frame showing the extent of eg. Fig 1a in Fig 9a.

**Lines 179-185:** Several issues. 1/ Explain buttressed vs overbuttressed. The Borstadt parameter ranges between 0 and 1 according to the caption to Fig 9. So why is the scale in Fig 9d,e,f from 0 to 2? 2/ Move grounding line flux for init here. Consider comparing your findings to the other modelling study which you cite in the Introduction wrt grounding line fluxes (de Rydt et al., 2021). 3/ Velocities are not discussed at all, this needs to be added. Especially since velocities and their changes after ice tongue breakups are mentioned several times in the Introduction. Modelled results are presented at the ice surface only, correct? What about their distribution in the vertical which is also a direction in

which the cracks evolve. 4/ Panel 9a shows a modelled region southeast of 79 NG, in all other panels, this is not shown. As it is not relevant, I suggest to remove it in panel a, and also in Fig B1. If not, pls motivate and explain why it is kept.

**Section 4 – Discussion**

**Line 187:** "First…" does this opening refer to lines 190-210? One does expect a "Second", which would perhaps be the discussion of the modelled ISSM results in terms of experiment design (lines 213-214), as well as a "Third" which would be changes in regional climate and their potential impact on 79 NG (lines 215-220)? And what about adding a "Forth", see below. Re "first", my specific questions are: 1/ line 194: provide the year during which this mass loss occurred. 2/ Line 200: For consistency, star a new paragraph when ZI is described. Regarding "Second", the discussion should be extended to include more nuanced reflections on the limitations of diagnostic simulations and to offer an explanation as to why prognostic simulations were not address here. Also, it would be useful to include reflections on whether or not a crack propagation (instead of removing parts of the floating tongue based on observational evidence and some extrapolation) could be captured in a prognostic setup?  Also, are results from similar numerical simulations regarding the buttressing of JI, ZI and WIS available in the literature? This would be an interesting comparison. Likewise, the grounding line fluxes modelled here could be placed in observational (and modelling, if available) context, which would provide a broader picture with regards to future expected changes at 79 NG. If no additional FS simulations are run for comparison (see my comment above) a discussion concerning the potential shortcomings of the BP model simulations needs to be included.  Regarding "Third", why is the impact of expected future climatic change on 79 NG in a rapidly changing Arctic not discussed, especially with a focus on supraglacial melt features that have been mention a number of times in the manuscript? Regarding "Fourth": in the introduction it is mentioned that 79 NG is one of three remaining ice tongues at the northern Greenland margin. I suggest adding a discussion focusing on a comparison with the other two, Ryder and Petermann. That is at least as relevant as comparing to ZI, WIS and JI.

**Section 5 – Conclusions**

**Line 234.** I recommend to weaken this to "may be at the onset…"

**Appendix A: Data**

**A1** is very well written. But I find the figures not as helpful as they could be because of a lack of overall structure. So, before setting out with A1, pls consider presenting an overview figure like the present Fig. C2 (or a variant thereof, perhaps with more zoom into the relevant region?) as a main orientation figure for the Appendix, before you present data along the various profiles in subsequent figures and dive into the various appendices A1, 2, 3 etc. Minor changes: Line 260: "*was formed which is shown in Fig 7*" -> was created. Line 247: "*Furthermore, we apply*" -> Furthermore, we use. Line 257: "*Here we apply SAR interfermetry*" -> Here we apply InSAR.

**A2** is very well written. My major question is: Why is there a separate Appendix C with Figs. C1 and C2, and a separate Appendix D with Fig, D1 which all clearly belong to A2?  Here in A2, you should continue to present Fig C1 (C2 should be taken care of at the start of the appendix, see above), and

then present Fig. D1.  Line 272. What is the thickness of the floating tongue? I suggest to include that as base info somewhere in the introduction, see one of my comments above. Figure C2: Please add map showing location, orientation and length of the profiles in panels a, b c. Alternatively, perhaps introduce a new overview figure where all profile locations are given, see my comments below in relation to the Appendix. Pls specify whether the same equipment was used in 2013 (data in panel a), or add a ref where the data description can be found. Abbreviation EMR has not been introduced. Line 268: "antenna" -> antennas.

 **A5**: Line 307 *"mosais"* -> mosaic

**Appendix B: ISSM model setup**

The ISSM model setup for the diagnostic runs in Section 3 is insufficiently described. Pls provide a more complete description.  Specific comments/questions: Fig. B1: add location of 79 NG to the figure. Consider cropping to the relevant area, cf. comments in the context of Fig. 9. Line 315: Why are only winter surface ice velocities used for the inversion? Line 316 *"Since the surface velocity field have"* -> Since the surface velocity field has . Equation B3: What is the meaning of the plus side in front of gamma_1?

---

## Author Comment (AC1)

Dear Reviewer 1,
many thanks for your very useful review. We will follow the vast majority of your points in the revised version
and give detailed answers to your comments below. We are very grateful that you have put a lot of effort -
and time - into giving us recommendations.
Some of your points were also raised by Reviewer 2 and we thus refer occasionally also to our answer to
those answers. You will find below 'Done.' as an answer to some of your points. In that case we have
directly included your suggestion into the manuscript and elaborate more on it, but this is more for grammar,
typos or smaller issues. All other things are answered in detail.
Again, many thanks for helping to improve our manuscript!
Best wishes,
Angelika and all co-authors

I read the paper in the order it is written and at several points found it difficult to understand the results
presented because the methods are hidden in the appendices. I would suggest moving the methods into
the main text, either as a dedicated methods section or as subsections to your observational results and
modelling results sections. If the methods are not moved to the main text, at the very least there needs to
be better signposting, or summary methods sentences at the start of the results sections to make it clear
what data/experiments were used to make statements in the results. I've included some suggestions in
the specific comments below. The paragraph at the beginning of Section 3 is very clear, so perhaps a
similar paragraph at the beginning of Section 2 (stating the type/date range of imagery used) would be
sufficient.

We have thought lengthy about including the methods in the main text while we wrote the original version
and did it now again. It is the number of different datasets and with those methods why we have put it into
the appendix. We wanted to tell the story from the past situation, the current changes to the impact in the
main part, as we thought readers with very different own background may read it and a modeler might find
the details on n instruments/dataset distracting and a scientist from observational glaciology to be distracted
by the details on the modeling. If we would use only with up to ~three different data sources, a method
section would be our choice, but with this larger number, we still want to keep it in the appendix.

The modelling experiments are well thought out and the second experiment was a useful addition to the
first that just removed the region that appears to be about to collapse. However, I did wonder if a third
experiment would be interesting, where you remove the entire ice tongue to see what the total/maximum
impact of ice tongue loss would be. This would be particularly interesting given that you show that the
regions close to the grounding line appear highly buttressed.

Yes, we fully agree that this is an additional interesting experiment and we have completed the simulation
and will include them in the revised version. Currently, we are trying to increase the mesh resolution in
order to also incorporate Reviewer 2's suggestions. This will also then lead to an overhaul of the figure.

Specific comments
Line 3: I think the link to sea level rise in the future due to ice tongue disintegration could be more
convincing here

We agree that his can be improved and will present a better sentence in the revised version.

Line 5: Specify what 'recent changes'

Very good suggestion and will be included in the revised version.

Line 6: Perhaps state when these cracks progress further upstream compared to 2010.

We will include 'since 2016' in the revised version.

Line 17: Rephrase 'since more than a decade', perhaps 'In the last decade, mass loss has reached northern Greenland...'
Many thanks, this is definitely the better wording and we will include this into the revised version.

Line 21: Would be useful to refer to Figure 1 when you state the ice front is in contact with the ice rises.
Very good suggestion and will be included in the revised version.
The second half of this sentence is a little vague, can the statement be more convincing, along the lines of 'the impact of ice rises on stabilising the ice front.
That is indeed a by far better formulation and will be included in the revised version.
Consider changing 'object' to 'location'
Yes, indeed location is by far more suitable and will be rephrased accordingly in the revised version.
Many thanks for reading so carefully and making so useful suggestions!

Line 23: You mention basal melting as a mass loss mechanism but do not then discuss it elsewhere in the Introduction. While I appreciate the focus is on calving, I think at least a sentence on melt rates beneath the ice tongue would be useful, making reference to previous work e.g. Wilson et al., 2017
We apologize that we mixed up the citation of Wilson's work. It should actually have been Wilson et al. 2017 and we cited Wilson and Straneo (2015) in line 220. The revised version is correcting this to Wilson et al., 2017.

Line 26: I think floating ice sheet should instead be 'ice shelf/tongue'
This is indeed correct and is reworded accordingly.

Lines 27-31: Given that a number of ice tongues in northern Greenland have recently disintegrated (ZI), or calved large tabular icebergs (Petermann Gl.), it would be good to include some context for the calving style you are referring to here, and how this compares/contrasts with calving elsewhere in Greenland from floating ice shelves/ice tongues.
This is also mentioned by Reviewer 2 and we discuss that in more detail below. We agree that it is important to mention other examples. The revised version will contain references to examples of calving fronts with pinning points from Antarctica, as the floating tongues mentioned here are not exhibiting any pinned calving fronts. We will also add text to compare that to floating tongues.

line 35: Given that Zacharias Isbrae is often also referred to as Zachariae Isstrom it would be useful to include both names here. Also state the time period (decade) over which the ice tongue was lost.
That is a good suggestion, many thanks. It will be included in the revised version as suggested.

Line 39: Add in the size/length of the ice tongue, either here of line 20 where you state it is the largest ice tongue.
We have decided to include it in line 20. Many thanks for pointing out that this was missing!

Line 42: Re-state what style of calving that was. Also where is the evidence for the style of calving in the 1980s, from satellite imagery? that was examined as part of this study? It needs a reference if not.
This is a good suggestion, in particular as this is no tongue-style calving. We will incorporate 'still in the same style as in the 1980's, with one lateral rift growing and widening over numerous years, similar to calving at Petermann Glacier.' in the revised version.

Lines 47-50: Given that you calculate buttressing, and discuss this when conducting your modelling experiments, it would be nice to include a sentence here to state how it impacts upstream flow.
We will include: " A retreat of the floating tongue might imply a reduction of the buttressing exerted on the upstream part and perhaps lead to increased ice discharge."

Lines 51-54: Before launching into the summary of the methods and imagery used, I think there should be a clearer statement of the research aim and justification.
This is indeed a very good suggestion. We will add here a paragraph doing exactly that.
It would also be useful to state the time periods over which the study is conducted, e.g. datasets used for xx to xx
This has also been suggested by the second reviewer and we will include a table in the Appendix in which we list the datasets and the respective time period.

Figure 1: Most people may know where in Greenland 79NG is, but I think an inset map of greenland and a box/marker showing the study region would be useful. This inset map could also include the modelling domain used too. Also, include the time period of the crack formations A-F so it is clear when the 'oldest' and 'newest' refer to. Consider adding 'IR1' and 'IR2' labels to panel a) as well.
We agree, the overview figure is needed. We discussed if we could also fit the modeling part into it, but it seems to us to become too crowded, therefore, we suggest having another inset figure in the modeling section.
The suggestion to have a legend with A-F and the respective years is very good and we will definitely include this in the revised version!

Figure 2: It's quite difficult to see the calving fronts in panels (a) and (b), can you increase the contrast of these images? and perhaps also digitise the calving fronts so that it is obvious where they are.
Unfortunately, even with higher contrast the reflectance of the snow on sea ice and on the floating tongue is very similar. What we suggest doing is to add a thin colored, dashed line, marking the calving front position. With the dashed style the front can still be visible in the image, but is highlighted.

Line 55: Given that the material on the modelling simulations is all in the appendix, I think a clear summary sentence on the experimental design and purpose of these experiments is needed here.
Done. The sentence is rewritten to "The observational datasets are complemented by numerical modeling simulations which aim to assess the instantaneous  velocity response to a retreating floating tongue (Section 3 and Appendix B)".

Lines 68-69: It is not clear to me where the evidence for the statement that the tongue type calving style has been present for 28 years. It was also not clear from Appendix A1 what the date range of imagery that was used in the study. Did you examine imagery from 1975 in order to make this statement? Please clarify, both here, and in Appendix A1 (see later comment on this).
Yes, we did examine the Landsat 2 image from 1975-08-26 for the statement, but we decided to not include that in a figure, as this was also part of other studies. We suggest moving the sentence 'This is in line with the findings of Khan et al. (2014).' to line 69, as it is a very good reference for it.

Line 71: Can you instead provide a summary/overarching statement about how the the calving front has changed, instead of using 'tremendously'
You are right: the sentence is somewhat misplaced here, as it is in the middle of the discussion of what changed and how it changed. We suggest removing it entirely in the revised version.

Figure 3: Some minor suggestions: add scale bars to (a) and (d), or state in the caption they have the same scales as (b) and (c). (b) state the time period of the surface elevation data.
We will add a scale bar to (d), however, the optical image from the onboard camera cannot be perfectly georeferenced, as it was our old system. We have tried to scale it similar to panel (b), but no exact scale can be given. We will state the date of the laser scanner DEM in the revised version - many thanks for pointing this out!
Change to 'photo from the Canon camera'
Done.

Line 76: As mentioned above, where does the evidence for tongue-type calving in 1975 come from?
From the Landsat 2 scene acquired on 1975-08-26.

Figure 4: It would be useful to show this track on a study figure or inset map so it is clear where it is located.
This track is shown in Fig. C2, but we forgot to mention it in the caption, which we will do in the revised version.

Line 82: Restate here when it was formerly grounded and when 'recent past' refers to.
We rephrase this sentence to 'we infer that a grounded spot existed at least for four decades and became afloat early 2014.'

Line 87: State the tidal range here, including references, or refer to the Appendix section where it is mentioned.
We added the sentence: , expected to be in the range of 1 m based on measurements of Reeh et al. (2000), Christmann et al. (2021) and FES2014b ocean tide model (Lyard et al., 2006),

Line 89: Consider making Fig.C2 a panel of Fig.1, it would be nice to see the location of the profiles in the main text.
In the meantime we are convinced that this is a good suggestion! Many thanks for pointing this out. We are planning this as Fig 1c and also take Reviewer 2's comments into account.

Line 89: The second half of this sentence needs rephrasing for clarity.
The sentence will be rephrased to 'Interestingly enough, ice penetrating radar shows in 2021 at the location that was formerly grounded still thinner ice (see Fig.*, location of profiles shown in Fig.*).' Figure numbers will change in the revised version.

Line 90, Figure 5: As you talk about radar profiles for 2021 and 2013 in this section, I think it would be useful if both appear in the main text rather than the appendix. Why not add a second panel to Fig.5 with the 2013 radargram?
Yes, this is indeed a very good idea. We will prepare a figure with two panels for the revised version.

Line 94: Can you include a measurement of how much crack A has widened?
In principle, this is a good suggestion, but the motion of the floating tongue is pushing the central part towards crack A. However, we think it is definitely worth it to give the reader an impression of how much the rifts changed and we suggest including the change in width for crack D instead. This will be included in the revised version.

Line 96: I don't find 'intersect the ice entirely' to be particularly clear, consider rephrasing, perhaps 'propagate through the full ice depth'

Done.

Line 99: see previous comment related to 'intersect the ice shelf'
We will rephrase this to '.. there is no indication that they are rifts that propagate through the full ice depth…'

Line 106: What evidence is this statement based on? the timing of lake formation/drainage did not coincide with changes in the cracks? Please expand on this.
Also Reviewer 2 has mentioned that we shall elaborate more on the role of hydrofracture and we do understand that this needs further discussion. We will definitely do this in the revised version and are actually happy to expand the text with a more thorough discussion of that topic..

*None of the cracks is a hydrofracture. Hydrofractures are basically crevasses (prexisting fractures) filled with water, either from surface melt draining into them as on Antarctic ice shelves (e.g. Scambos et al. 2000) or are facilitating supraglacial lake drainage (Das et al. 2008 ,Chudley et al. 2019). Crack A-D and F are newly formed cracks that grow vertically and horizontally at once - they are rifts. As an example, Crack D was initiated in March when no melt water was available at all. Also, the newly formed cracks do not follow the remnants of old crevasses, which hydrofractures would have done. Crack E is the only crack that has a potential for becoming a hydrofracture. Formed in 2019 it has so far survived three years without propagating vertically through. Although the floating tongue at the calving front is densely covered with melt ponds in summer, the lakes are small and shallow so the stress due to the water filling the crack has not yet been large enough to initiate hydrofracture. Hence, we conclude that the rifts are not hydrofractures but initiated by stresses (not due to water pressure) exceeding the material strength.*

Line 108: State why this eastern part is 'interesting'
This was also mentioned by the second reviewer and will be rephrased in the revised version of the manuscript.

Lines 110-113: I wonder if these sentences on crack modes would be better placed in the introduction when you first use the term 'tongue-type calving style'
Yes, very good point! We will move these sentences to the introduction where we add an overview of fracture mechanics and explain the different modes and correlated stresses. We hope this will give the reader a better understanding of the terminology. This was also suggested by Reviewer 2.

Lines 127-128: Rephrase these sentences for clarity.
Indeed, the reviewer is absolutely right in requesting a clearer text. We have rephrased this text to:
*'Crack arrest is not coinciding with rivers or lakes, as we find evidence for cracks to propagate across meandering rivers and lakes, and propagating further. One may wonder why rivers and lakes are not disturbing crack propagation.*
*In comparison with the ice thickness of about 80m an 1-2m deep river or 3-4m (based on ALS data) deep lake is still a minor change in thickness. From our perspective, this surface topography can be seen as a surface roughness, but without an additional stress concentration, a surface roughness alone, is not controlling the propagation of a crack.'*

Line 145: Explain the characteristics of a 'kind of bridge'. I appreciate you do this on line 148, but I think it would be better to explain the ice bridge at 79NG first, and then refer to the Wilkins Ice Shelf.

This is a good suggestion. We will change the order in the revised version by first describing the ice bridge at 79NG and then going into the example of the Wilkins Ice Shelf - and it actually reads now much better! Thanks for suggesting this!

Line 155: Restate the time over which the calving style changed.
Yes, this is indeed very useful information to the reader. We will include that in the revised version.

Line 158: I'm not sure how useful this comparison to Wilkins Ice Shelf is, given that they have different settings, is there anything to suggest they would be similar in size? Perhaps make it clear why this comparison is necessary. Also, refer to a Figure or the location of glacier draining into the ice bridge from the south on Line 160.
We think it makes sense to mention all commonalities as well as differences of the two ice bridges to gain a detailed understanding and provide an explanation why the 79NG ice bridge is stable although it is narrower than the WIS bridge. We will elaborate that in the revised version and also include the reviewer's suggestion and refer to an adequate figure.

Line 170: Given that the methods are in the Appendices, please add a sentence to the main text that presents how your model replicates observations, e.g. the misfit between observed and modelled velocities after the inversion or how the grounding line flux calculated by the model compares to published estimates of grounding line flux/ice discharge.
We added a RMS between observed and modeled surface velocity; we differentiate between the grounded and floating part. We don't think that a comparison between simulated grounding line fluxes and other cited values is a good measure of how the model replicates the observations because it is strongly dependent on the location and length of the grounding line or flux gate.

Line 175: It seems that the impact of removing the ice tongue from the ice rises, and even removing it half way up the fjord, was relatively small, <10% increase in ice discharge. Therefore I'm wondering why you stopped there, and didn't follow up with a third potentially 'high-end' scenario in which you remove the entire ice tongue. This would be particularly interesting as it seems from your buttressing maps that the region closest to the grounding line is highly buttressed. This would be complementary to the impact of removing sections of ice tongues/ice shelves close to the grounding line in other regions (e.g. Petermann Glacier Hill et al., 2018 and Larsen C (Mitcham et al., 2022).
Done. See answer above in the preamble.

Line 179: Again, as the methods are hidden in the Appendix (see major comment), there needs to be summary statements or better signposting in the main text. Here for example, it would be better to say something along the lines of 'We calculated ice shelf buttressing following the method presented in (ref) and found that...'
Done. We agree with the reviewer that this part needs improvements. We have extensively rewritten this part in order to better guide the reader through the experiments and findings etc. See also reply to RC2 179-185.

Line 180: Explain the term 'overbuttressed'
Done. See answer to RC2 Line 179-185.

Lines 193-194: Re-state the 14 month period that this 15% area was lost, and the decade over which it as remained in this 'intermediate state'.

We understand that the actual date is what the reviewer asks us to include in the text and we will do so in the revised version.

Figure 9: (a) It looks like the velocities near the northern part of the calving front go off the bottom of the scale? Are they zero here? or excluded from your domain? In (b) can you somehow highlight (e.g. polygon, shading, arrow) the region that has been removed during the experiment. Same for (c). Then refer to this Figure/panels when presenting the experiments in the main text.

Panels (d-f): white is not a good colour for the highly buttressed regions given that the background is white, consider changing the color bar.
Done, we used an improved colormap
In the caption, state what a value of 2 represents.
Done. We update the figure caption.

Lines 201-202: Is there a reference for this statement that the tongue of ZI was highly heterogeneous in the 1980s.
Done. We have added reference to Khan et al., 2014, which studied ZI using aerial photos from 1978 and Thomas et al. 2009, which studied ZI using NASAs ATM data from 1994-2001.

Line 204: It is a little bit confusing to refer to locations (e.g. 'northeast part') of ZI's ice tongue without referring to a figure. Consider labelling these locations on existing figure or creating a new figure/panel on existing figure (e.g. Fig.1).
This study is focussing on 79NG, this is why we do not want to introduce figures on ZI's ice tongue and it is far outside the range of Fig. 1. However, the new references included in the revised version provide context for these statements and provide a useful discussion for readers.

Line 213: Can an 8% increase in ice discharge be considered a 'significant contribution to sea level rise'? If so, some additional context/justification is needed. Consider referring to other examples of observed or modelled ice tongue calving/collapse elsewhere in Greenland and Antarctica.
We understand the reviewers point and will delete significantly in the revised version.

Line 213: Change 'is already' to 'would' because the calving in Fig.9c,f has not happened yet.
Indeed, this is the correct formulation and is included in the revised version.

Line 217: What about air temperature changes between 1990 and 2020?
This is somewhat unclear to us. In line 215-217 we explicitly write about the changes in air temperature and refer to Zhang et al., 2022.
Also in the following sentence, make a clearer link between thinning at the calving front and the potential for collapse in the near-future.
We have evolved this link throughout the manuscript, from presenting evidence for thinning, ungrounding and subsequent change in rift formation. It is unclear how the entire route shall be summarized again in one sentence. These lines 215-220 are providing the background to assign the thinning to surface melting and exclude basal melting being a driver here.

Line 217-218: I think it would be useful to expand on this discussion about thinning of the ice tongue near the calving front into the context of the rest of the ice tongue, i.e. the oceanic forcing is important for high rates of basal melting elsewhere on the ice tongue and thus responsible for the observed thinning (e.g. Mayer et al., 2018).

It is correct that oceanic forcing is important for high basal melt rates and thinning of the tongue, but the warm water sits at the lower part of the water column and is thus not in contact with the ice at the calving front. To discuss oceanic forcing elsewhere along the floating tongue would thus not give any additional information on the scenario at the calving front.

Line 227: Can you obtain measurements of basal melt rates to clarify this statement 'presumably small'
Measurements of basal melt rates require two expedition minimum, which is nothing we can achieve easily in this remote area. In addition, we would not be able to obtain past basal melt rates, which would be needed in the context of this manuscript. In addition, we refer in line 220 to the remote sensing based dataset of Wilson and Straneo (2015).

Line 243: You use a number of different satellite senors and I think it would be useful to include a table somewhere of the dates and sensors used. Also include the full time period for which your study covers.
This is a very good suggestion - we will add that to the appendix.

Line 278: Why not include Figure C2 in Appendix A? Same for other figures in Appendix C and D.
This is a very good suggestion! We will incorporate the figures in the respective subchapters of Appendix A, so that in the revised version we have all figures from observational data in Appendix A and have only Appendix B with modelling.

Line 284: 'usually better' by how much?
Changed to: less than

Line 311: It looks like the resolution is also refined based on distance to the grounding line and calving front? If this is the case it would be worth mentioning.
Done. You are right, the description of how the mesh is generated is incomplete and we have rewritten this part to: "Model calculations are performed on an unstructured finite element grid with a varying horizontal resolution between (Fig. B1). Over fast flowing ice (i.e. >300 m/a) we employ a resolution of 1000 m. Around the grounding line we refine the mesh up to 500 m. The domain is vertically extruded with 15 layers refined to the base."

Lines 329-330: Is it a common approach to assume the ice rigidity is constant over the grounded areas? Perhaps refer to other studies using this approach/temperature value.
We agree with the reviewer that the constant ice rigidity is a very crude approach for the grounded part. A better approach would be to use a depth-dependent ice rigidity from other products, like thermal spin-ups over a glacial cycle. However such an assumption is very common when running an inversion for the basal friction coefficient

Line 338: Rather than state excellent agreement, can you instead include the final value of the cost function, or some measure of the misfit between observed and modelled velocities.
Done. See comment to Line 170. We also provide the RMS values here.

Technical comments
Line 117: Change 'figure' to 'Fig 6c'
Done.

Line 125: Add 'the' before 'direction'
Done.

Line 130: 'at the end' instead of 'in the end'
Done.

Line 136: 'Remarkably there are'
Done.

Line 156: Delete 'with changing'
Done.

Figure 6: (d), there are no arrows shown in light yellow (<10) so this could be removed from the legend.
This is a good suggestion, we will do this in the revised version of the manuscript.

Line 188: Change 'featured' to 'features'
Done.

Line 307: 'mosaic' spelling
Done.

---

## Author Comment (AC2)

Dear Reviewer 2, dear Nina,
many thanks for your review that helped a lot to improve our manuscript! We find the detailed comments very useful and give below detailed answers and our plans for the revised version.
Your review was definitely very labor intensive and you have put a lot of time in it, for which we are very grateful for your effort!
Below you will find our detailed response to your comments. In some cases Reviewer 1 raised a similar point and we then refer also to our answer to Reviewer 1. In some lines you will find a short 'Done.' only, which means that we incorporated it right away in the revised version, but this is for grammar, typos etc only, all other things have detailed answers.
We again want to thank for your help to improve our manuscript!
Best wishes,
Angelika and all co-authors

**Major comments**

The **introduction** is not well structured and not concise (for detailed comment, pls see below). It is essential to better explain the configurational change at 79 NG, and why this motivates to study a regime change.

We understand that the reviewer wants to have a better structured and more concise introduction. In addition there were numerous requests to include an in depth introduction of fracture mechanical topics into the introduction. We will present both in the revised version of the manuscript.

The section **Transition in calving regime** needs to better streamlined and organized, too, and both figures and text need to be improved (for detailed comments, pls see below). Hydrofracture as a potential link between supraglacial melt features and crack propagation is dismissed, however, with insufficient arguments. The fracture mechanical context employed to describe crack formation lacks context necessary to follow the argumentation. It is essential to rework this section so that the wealth of useful information can be correctly understood.

We elaborate more on hydrofracture in the revised version and this can now already be found in the answers to the comments of Reviewer 1.

The **Impact** session suffers from two drawbacks. First, there is repetition as some information is already given in Appendix B and can be removed from the impact session. At the same time, Appendix B is incomplete and would benefit from additional information. Second, the text in the impact section is lacking important detail concerning the justification of model choice, description of results and metrics used, and that needs to be added (for detailed comments, pls see below) to allow for a better understanding of the results presented.

We will present a thorough updated version of the modeling part and our answers to the detailed comments below give more information on what we will present in the revised version. We will also include suggested modeling experiments.

The **Discussion** section appears incomplete and needs to be expanded, for suggestions pls see the detailed comments.

We are grateful for the detailed comments below, which we have answered all below.

The **Conclusions** reflect the manuscript in its present form, and may need to be revised slightly during the revision.

The **Appendices** appear to replace a full-fledged "Methods" section, which I found a bit surprising for this type of manuscript. Appendix A and B are introduced in the main text at appropriate locations, however, Appendix C and D are not, and actually should not, either, because they belong
to Appendix A. Pls see detailed comments below. Almost all figures need improvements, and Appendix B lacks necessary detail describing the diagnostic model setup.

We are restructuring the appendix in the revised version, so that all methods and figures related to observations appear in Appendix A and have then only Appendix B. We will also move the figures accordingly into the newly structured Appendix A.

The figures will be updated and we will incorporate all the detailed comments given below - also taking Reviewer 1's suggestions into account.

**Detailed comments (text from the manus is given in italics) Abstract**

**Line 9:** *"are a precursor"* -> may be a precursor

Done.

**Section 1 - Introduction**

**Lines 18-19:** provide names of the three floating tongues left. Return to these in the Discussion, see my comment later on.

The revised version will contain the names and we will refer to them in the discussion.

**Line 21:** I understand ice rises as features at the surface of an ice mass that are caused by iceflow over a bathymetric high ate the seafloor. Could the text *"Its ice front is still in contact with ice rises..."* be reformulated along the lines of e.g.: Parts of the front are grounded on ice rises, acting as pinning points, .... also because later (eg. in line 24 and in Fig.1) the ice rises are referred to as pinning point.

Thank you, this is a very good suggestion that is incorporated in the revised version.

**Line 23:** "*Calving and basal melting*..." References for this should be added. Also, basal melting is barely addressed later in the manuscript, so it should be mentioned that the focus here is on calving only.

We will include 'This study is focusing on calving only.' in the revised version of the manuscript.

**Line 24:** A ref is needed re "*calving ... is often governed by the existence of ice rises*", and a more nuanced formulation is needed as well. Yes, ice rises could be related to calving if they are in the frontal zone of an ice tongue. But I do not see how ice rises that are far from the frontal zone should be related to calving? Pls clarify.

Indeed ice rises at the calving front are directly affecting calving. Ice rises far away from the calving front have an indirect effect, as they lead to damage, which may then influence calving. This has not yet been studied intensively. We will include two references in the revised version, Thomas et al. 1979 and Wang et al. 2022.

**Lines 24-25:** "*When the ice mass moves passes by ...*" -> When the ice mass moves past ...

Done.

**Line 25:** " *... rifts (cracks that are separating the ice entirely....*" -> rifts (cracks that penetrate through the entire ice thickness)

Done.

**Line 26**: "*which grow laterally into the floating ice sheet*" -> which may also propagate laterally.

We suggest to include 'typically' instead of 'may'.

**Line 27:** "*This calving style can be found at many locations....*" Pls provide references.

It is actually non trivial to find references to this calving style for Greenland, the calving fronts of Petermann Glacier and Ryder Glacier are not pinned, and no reference can be found for the northern calving front of Zacharias Isbræ prior to the recent changes. Many examples can be found at Antarctic ice shelves and outlet glaciers, and we will refer to this work in the revised version of the manuscript:

BERGER, S., FAVIER, L., DREWS, R., DERWAEL, J., & PATTYN, F. (2016). The control of an uncharted pinning point on the flow of an Antarctic ice shelf. *Journal of Glaciology, 62*(231), 37-45. doi:10.1017/jog.2016.7

**Lines 27-28:** "*Tongue-type calving is a normal process and is very distinct from break-up of disintegration events*" -> Shorten to: Tongue-type calving is very distinct from disintegration events
Done.

**Line 29:** "During these events, a large part is shattered..." -> During these events, a large part of the floating tongue is shattered.
Done.

**Lines 30-32:** In addition to modern disintegration, there is also evidence of similar ice shelf collapses in the past that should be mentioned, see eg. Jakobsson, M, et al. (2011). *Geological Record of Ice Shelf Break-up and Grounding Line Retreat, Pine Island Bay, West Antarctica*. Geology 39(7), pp. 691- 694, doi:10.1130/G32153.1
Many thanks for keeping an 'paleo-eye' on the introduction! We will cite the mentioned paper in the revised version.

**Line 35:** Insert space before "(ZI)"
Done.

**Lines 39-44:** Several issues here.
1/ This info would fit better in the context of the description of 79 NG that is initiated in line 19 ff.
Many thanks, this is indeed a very good suggestion that we follow. The new text will read as follows:

*The largest floating tongue in Greenland is the 79NG (~70 km length and ~20 km width), draining an ice sheet area of 6.28% containing an ice volume of 0.58 m sea level equivalent SLE (Krieger et al., 2020). The floating tongue of 79NG has two calving fronts (see Fig. 1a), one in the north towards the Djimphna Sound (earlier this part of the 79NG was named Spalte Glacier) and one eastern calving front. The latest calving event at the northern front was taking place in 2020, still in the same style as in the 1980's, with one lateral rift growing and widening over numerous years.*

*Parts of the eastern calving front are grounded on ice rises, acting as pinning points (blue areas in Fig. 1a, denoted with IR in Fig. 1b) and it is hence an ideal location to understand the impact of ice rises on stabilising the ice front. The floating ice has an ice thickness of about 80-100 m in the vicinity of the calving front. Variations of the eastern calving front position in the past have not gone beyond an imaginary line between these pinning points (Khan et al., 2014) until 2013.*

2/ Also, some basic numbers (length and thickness of floating tongue, depth at grounding line etc) would be useful to have upfront around lines 19 ff.

Numbers will be included in the revised version. Depth at the grounding line seems to us not very useful in a study that focuses on the calving front.
3/ line 41: remove "act as pinning points", has been said above (eg. line 24).
Yes, very good point and will be changed accordingly in the revised version.

4/ "*.... Calving event ... in 2020, still in the same style as in the 1980's*": this is not helpful because the "1980's style" isn't described (yet).

We rephrase this in the revised version: 'The latest calving event at the northern front was taking place in 2020, still in the same style as in the 1980's, with one lateral rift growing and widening over numerous years, similar to calving at Petermann Glacier.'

5/ Incorrect grammatical reference in "*Changes of calving front ... style... have not gone beyond the line of these pinning point*" (how can a style go beyond a line?)

Yes, that is correct. We will rephrase this to 'have not gone beyond an imaginary line between these pinning points'

6/ "Line of pinning points" should be reformulated.

This has been rephrased to 'Variations of the eastern calving front position in the past have not gone beyond an imaginary line between these pinning points \citep{Khan2014} until 2013.'

7/ Incorrect tense: "A transition ... can...destabilize ... and eventually triggering " -> A transition ..can.. destabilize ... and eventually trigger.

Done.

**Lines 45-50:** Consider combining this paragraph with lines 33-38. Explain "*load situation*". Reformulate "*response ... in stresses ... of glaciers*".

Mention "Antarctica" in connection with Pine Island Glacier.

That Petermann is in Greenland is known once you introduce the glaciers name in the context of lines 18, see my comment above.

Done.

**Lines 51-52:** "*We leverage...of the floating tongue in the vicinity ...*" -> Here, we leverage ... of the floating tongue of 79 NG in the vicinity....

Done.

**Line 55:** "*... are complemented by numerous numerical *" -> complemented by numerical ....

Done.

**Line 56:** What do you mean by recent changes of 79NG configuration? From line 42 we know that something changed CE 2020, but not really what exactly. What exactly is the configurational change that motivates to study a regime change? This needs to be better explained, see my major comment.

We apologize, 'configuration' is a remnant from another draft sentence.

**Line 60:** Decide whether to refer to 79NGs tongue as a tongue or an ice shelf. This needs to be checked throughout the manuscript.

We have checked that throughout the manuscript and are now mentioning only floating tongue. However, we want to state here that it is well justified to use both in a text. While the floating parts of outlet glaciers in Greenland are normally called floating tongues, it is rather unusual to speak of over- or underbuttressed floating tongues. We also think that the readers are fully capable of dealing with both terms without becoming confused.

**Figure 1:** Several issues.

1/ Pls provide an overview map of Greenland with the location of 79 NG marked.

This has also been mentioned by Reviewer 1 and we will include this in the revised Fig. 1.

2/ Caption (a): "*IB marks an ice bridge (see text for details).*" -> add more precisely where in the text, eg. Section 2.

We will also highlight the area of the ice bridge in color in the revised version and rephrase the related sentences to make it more clear where the ice bridge is located. As we will add the former Fig C2 to Fig 1 as a third panel, we have also a better zoom into that area.

3/ Panel b: Pls provide some context for Fig 1b already in the introduction. Currently, the reader needs to wait for this until Section 2. Label (b) and scale bar are poorly visible. Consider rendering them in different color and at position with better contrast.

We will improve the figure as suggested. However, given that we move the introduction of fracture modes into the introduction already, we think that we can only afford a brief mentioning, as we otherwise overload the introduction.

4/ Panel b: Could cracks A-F be colored with individual colors? Explain in the caption what a stippled line vs a solid line means for the crack. Also, cracks are referred to as rifts in Fig 2. Decide on one terminology.

Indeed, obviously the cracks A-F could be colored with individual colors (six well distinguishable colors with each a good contrast to gray may be challenging), but we are not totally convinced if there is a lot of benefit from it. Reviewer 1 suggested including a legend with the date the cracks were formed, which we plan to do in the revised version.

Many thanks for mentioning that we missed to explain the dashed line in the caption! We will include that in the revised version. The dashed line denotes that the crack is not penetrating through the entire ice thickness, whereas all solid lines are rifts.

5/ Panel b: I guess IR1 and IR2 stand for IceRise1 and IceRise2. In panel a, they are referred to as pinning points. This should be homogenized, alternatively, say that you use the terms as synonyms. Consider adding the outline of the blue ice rises in panel a to panel b - may be useful for orientation. 6/ Add something about the supraglacial lakes.

We prefer to state that we use ice rise and pinning point as synonyms. One is the classification and one refers to its mechanical effect. We will state that in the text.

The reason why we did not add the outline into panel b is that it would overlay them and we find it useful to show their appearance.

**Section 2 – Transition in calving regime**

**Figure 2:** This figure shows changes in the frontal configuration 79NG. Pls consider modifying the figure such that configurations (outlines, prominent features) from 2010 and 2020 are overlain onto the 2000 configuration. This will make the comparison more visually straightforward. A central flowline could be added, as you refer to it later on (eg. line 72).

This is a very good suggestion and we will include that in the revised version of the manuscript.

**Lines 64-74:** Generally, the text here would benefit from streamlining and better organization. A simple guideline could be the chronology of events/calving front changes, from 1975 to present. Specifically, I have the following comments.

1/ Line 64: In this generality, I don't think this is correct.

One can eg. not say anything about changes in subaqueous calving based on calving front positions from optical satellite imagery. I think this needs to be reformulated.

We are not sure if we understand the reviewer correctly. Icebergs of all sizes are becoming afloat and we have never heard of icebergs being trapped underneath a floating tongue. Therefore all icebergs are visible in optical satellite imagery eventually. If fragments broken off from tidewater glacier calving fronts become larger or smaller or if formation of larger icebergs like at Petermann Glacier has changed can all be investigated using optical satellite imagery. It could well also be investigated by radar satellite imagery, therefore we will rephrase the sentence in the revised version and leave out 'optical'.

2/ Ensure easy matching between text and abbreviations used in figures. Replace "*between the two ice rises*" -> between the two ice rises IR 1 and IR2.

We will change it in this sentence as suggested in the revised version.

3/ Lines 66-67: *"Calving is initiated by the lateral rifts":* It would be helpful to indicate the rifts in Fig 2. Pls see comment above re Fig 2. Length of rifts in 2020 need not be given in the text if rifts A and D are indicated in Fig 2.

It is a very good idea to indicate rifts A and D in Fig. 2c and we will do that in the revised version. We do not understand why the length should not be given as it is not a repetition, but we will follow the reviewers suggestion in the revised version.

4/ Line 70-71: "*From the mid 2010's onwards, the calving front situation has changed tremendously*" – How does that go together with line what is stated in line 43: "*calving ... took place in 2020, in the same style as in 1980*". Is the 2020 event an exception from the new, post 2010 state? Pls clarify.

In line 43 we discuss the northern calving front and not the eastern calving front.

**Lines 75-91:** These paragraphs link to Figs. 3 to 5 and focus of the floating tongue becoming ungrounded at a certain location. I have several comments.

1/ "*Based on our database*" -> Based on the data analysed here.

Done.

2/ "*between the two ice rises a small grounded spot..."* What you are saying is that there is another ice rise, at the location marked by a blue star, correct? The formulation is not clear. Why don't you refer to this grounded area as an ice rise? Does an ice rise have to be of a certain size? Other criteria? This is confusing, pls clarify.

There has never been a formal definition of what characteristics an ice rise has, but grounded and ice flow moving around the ice rises rather than across, are the most referred to definitions. Given that we find that the location is not permanently grounded, we avoided calling it ice rise. As we have not high resolution velocity field (the one shown in Fig.6 is too coarse for it and there is no data available to produce any higher resolution velocity field) we can also not identify it as an ice rumple, which is normally characterized by ice flow being disturbed, but still moving across the grounded area. Therefore, we called it a grounded spot.

3/ "*Our assumption ...*" do you mean: our suggestion? To avoid repetition, you can replace "suggest" by "propose" in line 75.

We assume that the reviewer refers to line 76. We will rephrase this to 'Our inference is … '

4/ The text and **Fig. 3** don't go very well together. Starting point is 8 Aug 2013 (panels a and b). Pls add length scale to panel a and date to panel b, or combine them into one panel. Legend in panel b is m asl, pls add. Panel c has date 1 july 2013 so is showing a state prior to the one shown in panel a. When panel c is mentioned in line 80, no date is mentioned but it is natural to assume that it is a date after august 2013, which causes confusion. In line 80, reference is made to panel d, providing evidence for the ungrounding which is mentioned in line 76, however, without referring to panel d. Rather, because the sentence continues timewise with describing 2021 situation, there is risk that confusion arises (again) as to what actually is shown in panel d (obs scale is missing) and whether it is really necessary (from the caption to Fig. 3, it appears to provide geographical context only?). Also, it would be helpful if current Fig. 6d would be combined into current Fig. 3. Pls review Fig. 3 and caption and harmonize with the text.

Reviewer 1 also suggested adding acquisition date  to panel b and scale to panel d, which we will do. Please see our answer to Reviewer 1 why we do not add a length scale to panel a. We will also change (m) to (m asl) in the revised version, thanks for the suggestion. We will also add more information into the

caption to avoid any misunderstanding. Fig 6d has a wider coverage and if we incorporate it into Fig 3, there would only be very few velocity vectors.

Because the time of acquisition of panel a and b is the same, we have decided that they shall be next to each other and with four panels, we cannot do that by arranging it differently.

As the Fig 3c and d have both dates given in the panel, it is unclear where the confusion about dates arises from. We begin this paragraph with briefly summarizing the important information (line 75 and 76) and then present the details. This is why panel d is not mentioned in line 76. We guess that we can solve this by ending the sentence in line 80 after the reference to Fig 3c,d and start a new one with 'Furthermore, ALS data [...]' .

To summarize, we will do our best to harmonize the figure and text in the revised version.

5/ Line 81: "*Since 2013 such dome-like structure...*" -> Since 2013 such a dome-like....
Done.

6/Line 82: "*An alteration...*": Wouldn't that fit better in the discussion section? Alongside an explanation why?
We think that this shall remain in the presentation of the results, as the discussion section is taking a different perspective, comparing the situation at 79NG to other locations like Jakobshavn Isbrae and Wilkins Ice Shelf.

7/ Line 83: Grammar: become –> became.
Done.

8/ Line 84: "*Ungrounding can be a result from two instances:*" Ungrounding can results from at least two processes.
We wrote 'Ungrounding can be a result from two instances: (i) thinner ice approaching the shallow bathymetry or (ii) thinning of the ice locally.' and we do not see more than these two possibilities. Either thinner ice is advected or the ice thins locally. Please advise which other possibilities are given. The sentence is not stating what may cause local or non-local thinning.

And yes, indeed, at the ice rises, a lowering of ice tongue surface elevation must come from surface melt or dynamic thinning. However, at floating locations, even basal melt should reduce to surface elevation lowering because reduced buoyant forces cannot sustain the same ice thickness above the waterline?
We are not sure what the critical point is. Please see the answer above, potentially this leads already to clarification.

You have mentioned basal melt in line 23 but don't seem to pick up on it any further? Pls clarify. There are some remarks later in the discussion re suspected low basal melt rates where the issue could be discussed nicely in appropriate detail.
Basal melt rates are picked up in the last paragraph of the discussion section and the conclusion. As we can exclude this being the reason for thinning, we have not had a motivation to go into more detail.

9/ **Fig 4.** Please provide an inset to the current figure showing where the profiles are located.
Fig. C2 does show the location of the profile. We will mention in the revised version in the caption where to find the location of the profiles.

The calving front position of 2013 is nowhere shown, and the 2021 position could be inferred from Fig 1.a, but it gets cumbersome if this becomes a to-do for the reader. Why are supraglacial lakes included in the figure? If they were introduced in Fig 1, it would be easier to see a red thread (since lakes are mentioned later, too). What about lakes in 2013? Pls provide sufficient information regarding Fig. 4. Caption: Pls check grammar. A location cannot correspond to a Figure.
It is unclear to us why the reviewer refers to the calving front position here and we imagine that there is a misunderstanding. This profile (as shown in Fig. C2) is an exact repeat profile, flown in 2013 and 2021. The calving front position is obviously different in both years, but this is not the point we discuss here. This figure

is about thinning - and lowering of the surface. We assume that we confused the reviewer by the caption that has stated 'Thinning along the calving front … ' and we are rephrasing this to 'Thinning along a profile in the vicinity of the calving front …' When looking with the aim of analyzing thinning on this figure, two features may confuse: the rift and lakes. As the surface of the floating tongue is densely covered with small lakes, a profile may by chance cross a lake in year 1 and not cross lakes in year 2. This is why we mark these features to make it easier to understand. We think that the reader will also benefit here from moving Fig. C2 into Fig. 1 in the revised version, as the location of the profile is then already introduced when coming to this figure in the manuscript.
The calving front position of 2013 is shown in Fig 6d.

10/ Lines 89-91: Could this be better placed in line 82, before "*we conclude*"?
This is a very good suggestion and will be followed in the revised version.

11/ Fig. 5. Pls add lines showing the approximate location of the underside of the floating tongue, and the seafloor. Same applies to Figs. C1 (a-c).
We will insert arrows denoting the location of the ice base. As this is a radargram, the seafloor is not observed - ocean water is not transparent to radar waves.

**Line 92**: "For the sake of ..." This is repetition from Fig.1 and can be replaced by a ref to the Fig instead.
Actually, this is the location where the rifts A-F are first discussed. We can shorten the sentence by leaving out 'For the sake of', but it would be odd not to introduce the naming of the rifts.

**Line 95:** *"lower ice rise"* Do you mean "southern"?
Done.

**Line 98:** "*unification*" -> do you mean that the northern and southern branch of crack E joined? Pls Reformulate.
We reformulated that sentence to
'Although optical data show in 2021 that the northern and southern branches are close to join (shown in high resolution in Fig.\,\ref{fig:crackE-optical} marked by a red arrow), …'

**Lines 103-106:** "*None of the cracks is a hydrofracture*". How can you be sure? This is quite a strong statement, and the simple mention *"we did not find evidence"* is in this brevity not a satisfactory argument. What did you do to arrive at you conclusion?
We agree with both reviewers that we did not provide a good explanation for our conclusion that none of the cracks is a hydrofracture.
We can exclude Crack D easily to be a hydrofracture because it arises before the melt season. As the melt ponds at the 79NG were small and shallow at the time the cracks appeared, the water amount was not large enough to initiate hydrofracture. Furthermore hydrofracture is a secondary fracture process, because crevasses must exist beforehand. As cracks did not follow remnants of historic crevasses but correspond to higher stresses we conclude that exceeding the material strength is the reason for most of the cracks. We will elaborate our conclusions in the revised version to give a satisfactory argumentation.

You have bothered to indicate lakes in Fig. 4, why? "either from surface melt of facilitation supraglacial lake drainage": Even the latter are related to surface melt. Pls reformulate and check grammar, and also add a ref to at least Fig. 1b.
This has been answered above under 9/ **Fig 4.**

**Line 108:** *"at 79 NG's interesting part ...."* -> remove interesting and reformulate sentence.
Yes, indeed, this is not well formulated and will be rephrased in the revised version.

**Lines 109 - 124:** Several issues here. 1/ line 109-111: Provide a reference. Why mention mode III – it is not relevant for the following?
We include a reference to a text book in the revised version. As there are only three crack modes (and not a large number), we find it useful to introduce all three.

2/ Lines 111-112 and Fig 6: Explain briefly how shear and tensile stresses are related to mode I and mode II, and to first and second principal normal stress and maximum stress mentioned only in the caption to Fig. 6. Otherwise, this part and the following discussion cannot be understood.
We find this indeed a very useful point and have restructured our introduction and include a larger text part on general introduction to fracture mechanics. This is certainly useful for the reader and we are happy to do so. So, these sentences will be moved to the introduction where we will provide an overview of fracture mechanics including the different modes and their correlated stresses. Therefore we will expand this text passage in the revised version (see also our answer to comment of RC1).

3/ Why do ice rises induce shear stress and why does tensile stresses increase downstream of the ice rise? How is this related to velocities? This must not be left to the reader to find out but needs to be explained.
Yes, indeed it is absolutely worth it to elaborate more on this and we will do that in the revised version.

4/ Line 114. Grammar and terminology. What is an unstable crack and can a crack (a void) detach an ice berg? The latter is detached from the tongue and crack propagation accelerates – is that what you mean? Reformulate.
Many thanks for pointing us to this! Indeed the usage of unstable can become entirely misleading here. We intended to say that the crack propagates, but we did not intend to say if the crack propagation is stable or unstable. We have rephrased this for the revised version.
'The floating tongue downstream the ice rises is incised laterally on both sides and eventually one of those initial cracks is reaching a critical limit and propagates further, disrupting the entire ice vertically as well as horizontally and leading to the detachment of an iceberg.'

5/ Line 117 ff: *"Velocity field of 2014-2016"* Why a two-year averaged one? From where are these stresses? From ISSM? This needs to be explained.
The stresses are an output from ISSM. They correspond to the inversion state with the 2014-2016 winter velocity field as target. We will explain that in the revised version
This is a winter velocity field, meaning only scenes with frozen surface conditions are considered, as also explained in the Appendix for the inversion.
Please also see our answer below to line 315.

Line 129: "descend" -> decreases.
There is no mention of 'descend' in Line 129 or adjacent lines.

6/ Fig 6. Add to the caption that the scale in panel c is valid even for panels a and b. I think that would be helpful.
Many thanks for mentioning this. It will be added to the caption in the revised version.
Consider having panel d in Fig 3 instead, see my earlier comment.
There are only two data points in the area covered by Fig. 3, which is why we suggest leaving it here.

Pls show the floating tongue in panel d a lighter grey so a better contrast to the arrows and the star is achieved.
We are afraid that it would then be difficult to separate it well from the ocean mask.

7/ Did you provide evidence for your statement that (line 121) *"Loss of contact to the grounded spot is leading to an increase in main flow"*? If yes, please refer to it, if not, pls add and/or explain. Pls also check grammar in lines 121- 124, there is a mix of tenses that should be checked for correct use.
Fair point - we have checked the entire database at TU Dresden and have to conclude that we cannot present evidence for it. We will delete this statement in the manuscript accordingly.

**Lines 125 – 140:** Here cracks and supraglacial water and fatigue failure is discussed. Several issues:
1/"River" – has nowhere been introduced. You should mention (eg. in the context of Fig 1b) that supraglacial lakes can be connected by supraglacial streams. Is it "stream" or "river"? There is a lot of literature on supraglacial hydrology at the Greenland Ice Sheet so it should be easy to find the correct terminology and add a couple of references.
We conducted a literature search, but none of the studies we found is giving a definition for the width of a river or stream, which we understand being prone to resolution in the available imagery. In our text, the main info is, that the surface water body is not drained. If it is a river, a stream, a lake or a small pond is secondary from our perspective. In the papers we read, the term river is also not introduced. It is likely assumed that the term river is known to readers, which we assume here, too.

2/ *"de-watered"* -> drained. Also in Fig. D2.
Done.
3/ Add a line showing the base of the floating ice tongue to Figs 5, D1. Otherwise it is hard to see that the crack is not intersecting the entire ice thickness.
We will add arrows marking the ice base.

4/ *"Forming a non-intersecting crack".* I don't understand the terminology chosen. Crack E is rendered as a stippled line in Fig 1b, with individual ends propagating towards each other. In Fig. D2 their expected joining location is highlighted (btw how did you predict this location? And pls add a north arrow to Fig. D2) So why "non-intersecting"? And how is that linking to the fatigue fracture? I understand the latter, but not how the connection to the not-yet joined crack is made.
We have rephrased this to 'vertically non-intersecting crack', which describes the situation accurately. The caption of Fig. D2 is rephrased to 'The red arrow points to the location of the crack tips at the surface.'. We will add a north arrow in the revised version.

5/ Maybe it would be better to group this part as section on supraglacial meltwater features and cracks (lines 125-133) and a section on oceanic forcing of crack propagation (lines 133-140)?
We have discussed that lengthy, but given the fact that meltwater features have no effect on the fracture formation, we do not want to include a section, as it would be misleading. We have rephrased the text where it was mentioned before to make it more clear.

6/ Pls add missing information to the caption of Fig. 7. What is ranging between -pi and pi (if the symbol is pi?) Pls add a pointer to cracks A and E.
The phase is ranging between +-pi - it is an interferogram. Yes, the symbol is pi and we try to change the font. We will add a pointer to both cracks.

7/ Line 137: *"The ice plate"* do you mean the floating tongue? I don't fully understand why you need to argue that grounded and floating ice are connected at the ice rises (lines 136-139). Because you have fractures around the ice rises? Isnt it sufficient that the tides work on the floating part? Pls clarify.

As above, we will change the terminology 'ice plate' for clarification in the revised version.

To understand the stress situation as well as the kinematics it is necessary to describe the boundary conditions.

**Lines 141-143:** This is a bit unconnected, pls consider moving, providing additional context, or removing.

This is a good suggestion. We will move this further up, before crack E and the chaotic zone are discussed.

**Line 145:** *"detaching...."* -> reformulate eg as: will detach about 20 km^2 of ice but leave the ice bridge (IB, Fig. 1a) unaffected.

This will be incorporated in the revised version as following: We anticipate that calving along the cracks D–F–C–A detaching about 20 km$^2$ of ice but leave the ice bridge (IB, Fig. 1a; area ~ 55 km$^2$) unaffected.

Reviewer 1 had an additional useful comments to this text block. As a consequence we have restructured this paragraph and give more information on that in the answer to RC1.

**Fig 8 and caption.** Use either chaos zone or chaotic zone, it is not homogenous in the text. What does the red line denote? Add north arrow. Panel d shows locations for which no hinge zone is visible in interferograms. Are the latter shown? What is the meaning of the easternmost interferogram location – it is in the ocean?

We will add a north arrow. It is unclear what with 'What is the meaning of the easternmost interferogram location - it is in the ocean?' is meant. If this refers to the right pink dot, it shall be noted that the blue line marks the shore line and the pink point is located on the shore line.

**Lines 152-157**. Consider providing this information as a diagram (staples, or similar). This provides a more attractive means of conveying the info than in repetitive text.

We understand that repetitive text is not very appealing, but we give only four time snapshots here, which does not make a very attractive figure. In addition, the manuscript already has a large number of figures. So, we suggest leaving it as text.

**Line 158-159:** Add info on WIS IB to line 147 instead (and remove here).

We suggest leaving the text at its current position, as the lines 157-159 are giving the numbers for the ice bridge on 79NG, which is not discussed at line 147 yet.

**Line 159-161:** Lambert Land has not been introduced. Perhaps just remove? Sufficient that two glaciers drain into the 79 NG from the south? Do you show evidence of the bulging zone?

We agree - the revised version will contain 'from the south' instead of Lambert Land.

**Section 3 Impact – response of the 79 NG instability of the calving front**
**Lines 166-167:** *"We attempt ...discharge"* -> This is repetition from line 163. Shorten and mention ISSM directly, eg: We address this question using ISSM (add ref to App. B). Then: continue to describe the three experiments (line 170 onward).

Done. This part is rewritten accordingly.

**Line 167-170:** Repetition from Appendix B. Remove.

We do not fully agree with the reviewer to remove this part from the main text although it is a repetition from the Appendix. We think running an inversion is an important information to understand the initialisation state without reading the technical part of the Appendix. However, we deleted some parts to make the text better readable. Together with the comment above, this part reads: "As NEGIS is a fast flowing ice stream and drains a large area of the GrIS (17.23%, Krieger et al., 2020) it has the potential to contribute to sea level rise by an increased ice discharge, once the boundary conditions, such as its calving front, are changing. We address this question, how future large calving events or even a large disintegration event will modify the ice discharge, by using the Ice-sheet and Sea-level System model (ISSM, Larour et al., 2012). First, we initialise the model using observational data of surface and basal topography as a target to determine the initial conditions by a running a joint inversion for the basal friction coefficient and rigidity of the ice (see Appendix B)."

**Lines 170-178:** Here, the three experiments should be described, as well as model output variables and metrics used to derive conclusions, eg. the buttressing parameter that is so far only mentioned in the caption to Fig. 9.
We agree, the revised version will contain a better description of the experiments and the buttressing parameter.
Pls justify why you run diagnostic experiments only when the temporal evolution of the changes at 79 NG clearly are in the overall focus.
We are not intending here to simulate the retreat itself, but the long term impact it will have. To this end a diagnostic simulation of the new state is sufficient. But we fully understand that retreat simulations are very interesting and we are looking forward to go into that direction in a future study.

Pls also explain why a Blatter-Pattyn (BP) model was favored over a Full Stokes (FS) model, when in a recent study of the NEGIS, , 'considerable differences at the grounding line' were found and that 'results from non-FS models should therefore be viewed with caution' (tc.copernicus.org/articles/16/1675/2022/). Maybe the error is systematic and so not so important in terms of % increases in discharge, which is the main takeaway in the following. But, I would suggest that either running an FS simulation as a comparison or including a discussion of the potential errors / motivation of the model set-up is necessary.
Done. We will either include a FS simulation or discuss the potential errors as suggested. See also answer to major commnet by RC1.
Moreover: 1/ Line 171: Remove number concerning grounding line flux for the init run.
Done
2/line 175: "Bottleneck": Do you mean in term of the fjord geometry narrowing? Pls explain better.
We have rephrased this into '[...]in which we assume that the calving front retreats up to a point where the fjord geometry is narrowing (denoted southern and northern bottleneck (SBN and NBN), respectively, in Fig. [...]'.

3/ Line 176: Incorrect Fig ref, needs to be corrected to Fig 9f.
Done.
4/ Fig 9: parts of the floating tongue that are removed in the two perturbation runs should be rendered hatched (or otherwise highlighted) in panels b,c,e,f.
Done.
All panels can be cropped to instead focus on the relevant model domain.
As mentioned above, we do not crop the images as the new experiment "collapse" shows speed-up far upstream the grounding line. Sure, Figures could (d), (e) and (f) could be cropped, but we would like to keep the same figure extension for all figures of the modelling results.
Insert a frame showing the extent of eg. Fig 1a in Fig 9a.
Done.

**Lines 179-185:** Several issues. 1/ Explain buttressed vs overbuttressed. The Borstadt parameter ranges between 0 and 1 according to the caption to Fig 9. So why is the scale in Fig 9d,e,f from 0 to 2?

Maybe the caption is misleading. The Borstad buttressing parameter ranges from 0 to >1. A value of > 1 corresponds to overbuttressed ice shelf. An overbuttresed ice shelf means that the longitudinal stresses are negative, i.e. compressive. We added this to the text and clarified the figure caption.

2/ Move grounding line flux for init here.

Done

Consider comparing your findings to the other modelling study which you cite in the Introduction wrt grounding line fluxes (de Rydt et al., 2021).

We actually find it more useful to compare it with the changes in grounding line presented in Mouginot et al. (2019) as that study is presenting values for the same glacier. The revised version will include this.

3/ Velocities are not discussed at all, this needs to be added. Especially since velocities and their changes after ice tongue breakups are mentioned several times in the Introduction.

Done. See answer to RC1 line 179.

Modelled results are presented at the ice surface only, correct? What about their distribution in the vertical which is also a direction in which the cracks evolve.

Indeed, we presented surface velocities in the figures, but not in Fig 9f that is discussed here. Of course we simulate the 3D velocity field. Two things to clarify here: (i) the rifts evolve in horizontal direction, but through the entire vertical direction. (ii) the discussion in like 179-185 is about buttressing and grounding line flux and not velocities at the calving front.

4/ Panel 9a shows a modelled region southeast of 79 NG, in all other panels, this is not shown. As it is not relevant, I suggest to remove it in panel a, and also in Fig B1. If not, pls motivate and explain why it is kept.

First of all, all panels show the same region (also Figure B1). Since the other panels show Zeros of velocity differences and the buttressing parameter there is nothing to show. So, generally we would agree with the reviewer to crop the images. Since RC1 requested a third experiment, a full collapse of the floating tongue, we do not crop the images. This experiment shows large velocity differences upstream the grounding line which would be hidden once we crop the image.

**Section 4 – Discussion**
**Line 187:** "First..." does this opening refer to lines 190-210? One does expect a "Second", which would perhaps be the discussion of the modelled ISSM results in terms of experiment design (lines 213-214), as well as a "Third" which would be changes in regional climate and their potential impact on 79 NG (lines 215-220)? And what about adding a "Forth", see below. Re "first", my specific questions are:

We have rephrased this for the revised version to 'We start by comparing…'

1/ line 194: provide the year during which this mass loss occurred.

We guess you mean before the calving front settled. Because it is actually not a mass loss as it is a floating ice body. We will incorporate the year in the revised version.

2/ Line 200: For consistency, star a new paragraph when ZI is described.

Regarding "Second", the discussion should be extended to include more nuanced reflections on the limitations of diagnostic simulations and to offer an explanation as to why prognostic simulations were not address here. Also, it would be useful to include reflections on whether or not a crack propagation (instead of removing parts of the floating tongue based on observational evidence and some extrapolation) could be captured in a prognostic setup? Also, are results from similar numerical simulations regarding the buttressing of JI, ZI and WIS available in the literature? This would be an interesting comparison.

Likewise, the grounding line fluxes modelled here could be placed in observational (and modelling, if available) context, which would provide a broader picture with regards to future expected changes at 79 NG.

This is a very good suggestion and we will include a comparison with the values in Mouginot et al. (2019) in the revised version.

If no additional FS simulations are run for comparison (see my comment above) a discussion concerning the potential shortcomings of the BP model simulations needs to be included.

Please see answer above

Regarding "Third", why is the impact of expected future climatic change on 79 NG in a rapidly changing Arctic not discussed, especially with a focus on supraglacial melt features that have been mention a number of times in the manuscript?

We do discuss the changes in air temperature in line 215-219. It is actually the paragraph in which we also discuss why increase in air temperature and not basal melting is the cause for reduction in ice thickness.

Regarding "Fourth": in the introduction it is mentioned that 79 NG is one of three remaining ice tongues at the northern Greenland margin. I suggest adding a discussion focusing on a comparison with the other two, Ryder and Petermann. That is at least as relevant as comparing to ZI, WIS and JI.

This is a very good suggestion and we are happy to do so in the revised version.

**Section 5 – Conclusions**
**Line 234.** I recommend to weaken this to "may be at the onset..."

We discussed that in depth and came to the conclusion that we do not agree to include 'may' in that sentence. This sentence says that our findings *'indicate* that 79NG is at the onset of a transition from stability to instability', which is a moderate statement from our perspective.

**Appendix A: Data**
**A1** is very well written. But I find the figures not as helpful as they could be because of a lack of overall structure. So, before setting out with A1, pls consider presenting an overview figure like the present Fig. C2 (or a variant thereof, perhaps with more zoom into the relevant region?) as a main orientation figure for the Appendix, before you present data along the various profiles in subsequent figures and dive into the various appendices A1, 2, 3 etc.

We also got other comments about Fig. C2 being better presented in the introduction already and agree to move it further up. We will then also zoom more into the relevant area, but leave the southern extent of the figure as is, because we want to mark the ice bridge in that new figure then, too.

Minor changes: Line 260: "*was formed which is shown in Fig 7*" -> was created.
Done.
Line 247: "*Furthermore, we apply*" -> Furthermore, we use.
Done.
Line 257: *"Here we apply SAR interfermetry"* -> Here we apply InSAR.
Will be added in brackets in the revised version, see comments RC1.

**A2** is very well written. My major question is: Why is there a separate Appendix C with Figs. C1 and C2, and a separate Appendix D with Fig, D1 which all clearly belong to A2? Here in A2, you should continue to present Fig C1 (C2 should be taken care of at the start of the appendix, see above), and then present Fig. D1.

Reviewer 1 also suggested to rearrange the appendix and we find both reviewers suggestions very good. Our plan is to have two sections in the appendix, one on data and one with modelling. We will then also present all figures in Appendix A (the data appendix).

Line 272. What is the thickness of the floating tongue? I suggest to include that as base info somewhere in the introduction, see one of my comments above.

This is a very good suggestion and we incorporate this in the revised version.

Figure C2: Please add map showing location, orientation and length of the profiles in panels a, b c.

Figure C2 has only one panel.

Alternatively, perhaps introduce a new overview figure where all profile locations are given, see my comments below in relation to the Appendix.

Fig. C2 is the figure in which all profile locations are given.

Pls specify whether the same equipment was used in 2013 (data in panel a), or add a ref where the data description can be found.

The section A4 explicitly states that the laser scanner was the same instrument. Fig. C1 explicitly states which data is used in the different panels. We can, however, add a legend, in which we can list the data for the different profiles.

Abbreviation EMR has not been introduced.

We recognised that we did not only not introduce the abbreviation, we entirely missed to introduce this radar instrument! The revised version will contain a paragraph on this radar in Chapter A2. Many thanks for pointing this out, we would have missed that entirely and are very grateful for the careful reading!

We added in A2:

In addition and to compare to the more recent UWB system (Figure C1) we used data from the Electromagnetic Reflection System (EMR). The EMR is an airborne radio-echo sounding system used to map ice thicknesses and internal layering of glaciers, ice sheets and ice shelves. The system is capable to penetrate 4000 m thick ice. It was designed and built by AWI in cooperation with Aerodata Flugmeßtechnik GmbH, Technical University Hamburg-Harburg and the Deutsches Zentrum für Luft- und Raumfahrt e.V. (DLR). The radar signal is a 150 MHz burst with a signal length of 60 ns or 600 ns. The maximum performance is 1.6 kW with a sensitvity of 190 dB (Nixdorf et a

Line 268: "antenna" -> antennas.
Done.

**A5**: Line 307 *"mosais"* -> mosaic
Done.

**Appendix B: ISSM model setup**
The ISSM model setup for the diagnostic runs in Section 3 is insufficiently described. Pls provide a more complete description.

We apologize that the description was incomplete. In the revised version we envisage to provide a more detailed description.

Specific comments/questions:

Fig. B1: add location of 79 NG to the figure.

Done

Consider cropping to the relevant area, cf. comments in the context of Fig. 9.

We do not crop the images because of the new 'collapse' experiment which shows large velocity differences upstream the grounding line. See answer above to Line 179-185 /4.

Line 315: Why are only winter surface ice velocities used for the inversion?

The offset tracking methods have issues in summer as coherence is easily lost by the changing surface conditions. This leads to bigger areas not covered with data and also noisy velocity fields. Therefore we (and other studies, too) rely on 'winter velocity fields', so fields outside the melt period.

Line 316 *"Since the surface velocity field have"* -> Since the surface velocity field has .

Done

Equation B3: What is the meaning of the plus side in front of gamma_1?

Done. The plus sign is a typo and is removed.

---

## Author Response (AR2)

Dear Editor, dear Jan,

Many thanks for the very, very  helpful suggestions!

Before presenting below a point-2-point answer, we want to mention that we recognised that we did not include the new simulation results in the conclusion and abstract. Apologies for that! The new version has now an update of the numbers in abstract and conclusion.

We give detailed answers to all points below and prepared a new version of the manuscript with tracked changes.

Best wishes,
Angelika and all co-authors

LINE-BY-LINE COMMENTS:

 (all line numbers refer to the manuscript with tracked changes)

 L1 replace 'so far' by the timeframe of available observations
Done

 L1-2 The first sentence of the abstract talks about changes to the ice tongue of 79N, whereas the second sentence compares this to changes to the grounded ice of ZI. This seems like comparing apples and oranges. The third sentence then jumps back to 79N. Is the comment about ZI needed here?
Right, it is indeed not needed and we deleted it for the new version of the manuscript.

 L4 'will' lead to sea level rise, or 'is likely' to contribute to future sea level rise?
We have rephrased it to likely.

 L5 employed
Done
 L6 add comma: …calving front, from…
Done
 L7 unclear what is meant by 'normal' tongue-type calving. Is this commonly used jargon?
Correct, there is no normal or abnormal tongue-type calving, only tongue-type calving. We have deleted normal for the new version of the manuscript.

 L8 maybe replace 'increase' by 'expansion'
Done.

 L8 the description of the melange and ice bridge is difficult to comprehend without the help of Fig1 and 8.

Could you reformulate to say that damage upstream of the main calving front has created areas of open water and an ice-berg melange?

Very good suggestion. We incorporated this into the text as follows: The calving front area is further weakened by an area upstream of the main calving front that consists of open water and an ice melange that has substantial expanded, leading to the formation of a narrow ice bridge.

L11 maybe say 'future ice-front retreat and complete ice-shelf disintegration' rather than 'break-up or disintegration', which sound identical.

Done.

L11 'discharge of grounded ice' instead of 'ice discharge'

Done.

L12 add …'south-eastern area of the ice shelf…' for clarity

Done.

L26 'was taking place' or 'took place'?

'took place' is the correct description. Many thanks.

L26 The reference to calving style might need some further context. For example, can you say that since the first observations in the 1980s, all calving events have followed a similar pattern, following the extension of a lateral rift over several years?

Yes, that is a good suggestion. Indeed all calving events have followed a similar pattern. The new version is: All calving events still since the 1980's followed a similar pattern, with one lateral rift growing and widening over numerous years.

L31 'gone beyond' in the upstream or downstream direction?

In upstream direction - changed for the new version of the manuscript.

L42-45 by 'different mechanisms' do you mean a 'variety of mechanisms', or do you mean that crack formation is controlled by different mechanisms compared to crack propagation? In the latter case, I think references are needed, and a clear distinction needs to be made between what mechanisms are responsible for crack formation, and what mechanisms are responsible for crack propagation (or deformation).

Very well spotted! We indeed meant a variety of mechanism. It is changed in the new version of the manuscript.

L46 and following: some of these concepts are explored in detail in
https://tc.copernicus.org/articles/14/1673/2020/

It is correct that the reference proposed is exploring model I cracks, but many other references do too and at this stage in the text the only thing we want to do is to introduce the definition of mode I, II and III to the reader and therefore we refer to a textbook only.

L64 …'are' stabilizing... I would prefer: 'are buttressing'

Done.

L65 …'an' increase…

Done.

L66 'most recent' instead of 'latest'?

Done.

Can you add a specific time instead of staying 'in the past decade'?

We rephrased this to 'since 2012' and cited Khan et al. (2014).

L69 I think you use 'stable' to mean 'unchanged' throughout the manuscript. It might be good to clarify this somewhere, in order to avoid confusion with the term 'stability', which comes from dynamical systems theory and means something different: a steady system is stable if it returns to the same steady state following a small perturbation.

This is a very good point. We have incorporated in the introduction section this note: (we use stable as synonym to unchanged)

L71 'an' ice discharge increase

Many thanks - and Petermann Glacier was not in capital letters, too. Both changed now.

L72 Why do you start the sentence with 'However', if the measurements and modelling results are consistent?

Correct, the use of 'however' was not appropriate and we have deleted it.

You will also need to include Hill et al. 2018 (https://doi.org/10.5194/tc-12-3907-2018) here.

Indeed! It is now included.

L73-82 These sentences sound rather fragmented. Can you reformat into a coherent paragraph?

We rephrased this paragraph and moved the part on Petermann Glacier to a paragraph below, which was already discussing Petermann Glacier. A reviewer asked us to introduce Petermann Glacier in this paragraph, but as we are focussing in that paragraph on observed disintegration events, it was there anyway not at the right location.

L80 'mandatory' is a strong use of words; perhaps say 'informative' or 'a first step towards'

Changed to 'a first step towards'

L81-81 Do you have evidence from the published literature (or even anecdotal evidence) for this statement?

This is a very good point. It is indeed very difficult to find a suitable reference for this (which should be an inspiration for future modelling studies). One reference that presents some evidence is in Matsuoka et al (2015, https://doi.org/10.1016/j.earscirev.2015.09.004) which we cite in the new version of the manuscript. 'The area of ice shelves in this region decreased by 6.8% between 1963 and 1997, mostly in regions without ice rises and rumples near the calving front (Kim et al., 2001). This observation supports the hypothesis that ice rises generally stabilize ice shelves.'

L83 what do you mean by 'catastrophic'? I would remove this part of the sentence.

'Catastrophic events' is a well defined terminology (e.g. MacAyeal et al., 2003; Hulbe et al., 2004; Braun et al., 2009)

Also, by 'change in load situation', do you mean 'change in ice-shelf geometry'?

No change in stress situation. This terminology is very often used in the field of mechanics.

L88 Do you mean 'a moderate retreat of the ice front' instead of 'a moderate calving rate'? Pine Island is one of the fasted flowing Antarctic glaciers, and even a steady ice front location would require the glacier to have one of the highest calving rates on Earth.

This is correct and we have changed it to 'moderate retreat of the ice front'.

L92 'Set this into the context of the stress regime': can you be more precise? Are you linking changes in calving rate / calving flux / calving extent / … to changes in principle deviatoric stress / …?

We changed this to 'and compare this to principal stresses prior to the crack formation'

L94 replace 'destabilisation' with 'further retreat'

Done.

L95 'numerical perturbation experiments' instead of 'simulations'.

Done.

L117 I think you can already refer to IR1 and IR2 in line 112

Done.

L129 'ALS elevation data ARE lacking…'

Done.

L131-132 An 'alteration between grounding and ungrounding': this sentence confused me, because earlier on you say that this area has ungrounded. Do you mean 'episodic' grounding and ungrounding?

Yes, we want to say, that the location is not switching back and forth from grounded to ungrounded to grounded again. We have rephrased it to: An episodic change between grounding and ungrounding seems to be rather unlikely based on our data.

L131-134 I would fold this into the discussion above, where you provide evidence from other sources. I don't see why the radar data should be treated separately.

This is a very good suggestion. We have now first presented the observations and draw subsequently the conclusion.

L137 '…result from two processes' instead of '…from two instances'

Done.

L139 Unless I'm looking at the wrong feature in Fig4 (distance along track ~2.4km) it appears to me that the difference in surface elevation is about 7 or 8 meters?

You are definitely looking at the right image, but we compare the area outside the grounded spot to infer the thinning. The grounded spot was a domelike feature and once moved the elevation change is 7-8m, but to get information about thinning the ungrounded spots are the one to look at.

L142 You suggest surface melting as a process. How about dynamic thinning?

We divided the causes into an advection process or local thinning. As there is no evidence of a velocity change of the floating tongue in that area, but we find massive surface melting to take place (see the surface elevation change at the ice rises, also mentioned in the text), local thinning is definitely taking place. This is our way to identify local thinning to be the cause.

L146 'In addition' instead of 'Next'

Done.

L156 '…is similar in summer 2022,…'

Done.

L166 when you refer to the stresses in the floating tongue, say that this will be discussed later.

Done.

L176 How did you estimate components of the stress tensor? From ISSM?

Exactly, we used the remote sensing velocity product to conduct an inversion in ISSM. We have added to the sentence that we do this by means of inverse modelling.

L176-180 Can you show the evolution of the stress field as the calving front & rifts develop between the 1990s and 2020s? So reproduce Fig 6 but for different snapshots in time? You might find evidence in the stress field that could help explain the changes in rift propagation direction (or 'calving style' as it is referred to in the paper) after 2014. The following work might be of interest in that context: https://tc.copernicus.org/articles/13/2771/2019/.

This is exactly the problem here: the coverage is not sufficient to have an inversion before / after the rift evolution. We discuss this also in more detail below. We would have wanted to go exactly the route as in that reference, but the data coverage is just too poor for that. The TerraSAR-X supersite was 'just' the grounding line area, which is repeatedly covered, but no such coverage is available of the ice front in those years.

L183 The propagation into an area with lower stresses might well lead to the arrest of the rift, although the

behaviour also depends on the spatially heterogeneous material properties (ice thickness, critical stress intensity etc.) so I think a more nuanced statement is needed here.

The change in critical stress intensity factor is very unlikely over this short spatial distance - there is no indication that the density is different, or other factors varying over this short distance. The ice thickness is higher closer to the ice rise and has then a constant ice surface elevation over the length of the crack in a TanDEM-X DEM. If the thickness difference would be the reason for crack arrestment, the crack should have been arrested earlier. Nevertheless, we do not have any measurement of what exactly lead to crack arrestment, therefore we rephrase it to 'which is likely the reason for why the crack stops propagating'.

L182-184 Inferences about the type of fracture are provided as facts, but some explanation that links back to the explanation of the different modes of fracture in L46 etc. would make it easier to understand this paragraph. E.g. why is crack D 'clearly' a mode I crack?

Indeed we only threw a half sentence into the air, which was not adequate - apologies for that! As we only have the stress field before the occurrence of crack A, we cannot draw any direct comparison to the principle stress fields. That crack D is a mode I crack thus comes from the form of the crack opening. This is visible in Fig1c and clearly shows that the crack faces are moving apart like a mode I crack would do. We will include this in the new version of the manuscript. We are indeed very grateful that you mentioned this, because our text was very incomplete.

L185 '…along SURFACE rivers or lakes…'

Done.

L198-205 I find this paragraph hard to follow. Do you infer evidence for tidal bending at the location of rift E from the density of fringes in the interferogram? If so, I would expect there to be tidal bending -and hence 'fatigue cracks' - in many other locations with a similarly strong phase gradient in the interferogram. Can you explain this in more detail in the manuscript please?

It is a very good idea to include more detail in the manuscript! If the tidal bending would be large, crack E would have already become critical. In the interferogram one does indeed see a small change in phase, but no real fringe belt. Fatigue cracks could indeed very well be found at other locations of cyclic loading on short time scales. As cracks are often formed upstream the grounding lines already, and hence the grounding zone is often a zone of crack propagation rather than initiation. It might be difficult to distinguish between old mode I cracks and new fatigue cracks, but is would be a very interesting study indeed.

L206 pre-existing

Done.

L218 'We further consider THE potential future evolution OF the calving front'

Done.

L219 'will detach' instead of 'detaching'

Done.

L234 'was last observed' instead of 'disappeared'

Very good suggestion!

L236 'area' instead of width

We replaced length with area.

L238 What do you mean by a 'normal' calving front?

The WIS ice bridge had also two calving fronts, one supported by a thick ice melange, while the other had only open ocean or winter sea ice and thus no support.

L241 remove 'from south' before the reference to the figure.

Done.

L242 'A' bulging zone. Also refer to the relevant figure here.

Done.

Fig1 caption. In the description of panel b, refer to IR1 & 2 when you talk about the pinning points.

Fig1b shows all pinning points, not only IR1 & 2. Therefore, we would suggest to leave the caption of thins panel as is.

Replace 'marked in blue' by 'shaded in blue'

Done.

Fig3 caption. Is 'The location of the profile is presented in Fig 1d' needed here?

We do think that it is important to mention where the profile is actually located and suggest to keep the reference. In the revised version we moved this sentence to the end of the description of panel (b), as the panels © and (d) are not showing data from that profile.

Fig5 caption. Can you indicate the location of the radar profiles?

Yes, we added a reference to Fig.1d now.

Fig6 caption. 'scale bar' instead of 'scale bare'

Done

L267 'larger' instead of 'more dramatic'

Done

L276 'predominantly' instead of 'in the majority'

Done

L282 is 0.8% correct here?

Thanks for catching this, the number is not correct. We have now corrected the numbers.

L284 'significant increase' or '166% increase' instead of 'massive'

Done

L288 'First we discuss the WIS, which has …'

Done

L292 '…15% of area…': do you mean the area of the ice bridge?

No, it was the ice shelf area, the ice bridge collapsed entirely. We included 'ice shelf' now.

L289-294 The comparison based on broad-brush similarities in the geographical setting for both ice tongues seems rather speculative, given the detailed stress arguments that are invoked in the remainder of the manuscript. Is there any strong evidence from the literature that ice tongue area and other geographical descriptors such as lateral confinement, are good metrics to determine the vulnerability to collapse?

We are convinced that it is not speculative. The ice bridge on Wilkins Ice Shelf was astonishingly similar to what we find on 79NG. Yes, you are right, that there are different levels of detail in the arguments in the different sections/topics, but we would be reluctant to leave this comparison out. Actually, we instrumented the ice bridge with an autonomous GPS.

L295 'retreated past A lateral embayment'.

Done

Also: it is not clear to me what is meant by a lateral embayment. Can you describe the geographical setting in more detail?

In Csatho et al. (2017) it is called bay, in Johnson et al. (2004) ice lobe. We added now 'of the fjord'.

L307-310 Again, this sounds rather speculative. Do you have any evidence that the stress field changed significantly between calving events, i.e. once the ice front retreated beyond the pinning points?

We do not understand what you are referring to. Do you refer to 'Once a change in the calving front situation occurs the elastic stress redistribute instantaneously and can trigger further follow up events as well as lead to a modified viscous response over month–years, see Rankl et al. (2017) for WIS.'? The publication of Rankl et al (2017) presents stress in Fig. 4. So, yes, we have evidence that the stress field changed significantly at WIS.

L300 '…far easier to retreat past this area': Do you have a reference to the published literature for this statement?
We hope we understood the request correctly: you are asking for a reference for the statement about the Jakobshavn Isbræ. We included a reference to Csatho et al., 2008.

L320-325 The discussion about air and ocean temperatures is rather detached from the remainder of the discussion and reads like an afterthought. Can this be integrated better, e.g. by saying that you consider the different external drivers that might have contributed to the changes observed at the ice front, and then systematically discuss the different possibilities?
This is a great idea. We start the paragraph now with 'Next we consider different external drivers that might have contributed to the changes observed at the calving front, such as air and ocean temperatures.'

L329 '…crack E is none of these but 'COULD BE' a tidal fatigue crack…'
Done
L330 'speculate' instead of 'interpret'
Done
L334 'long' instead of 'large'
Done
L339 The use of terminology 'transition from stability to instability' is misleading in the context of dynamical system theory, since no stability analysis was performed in this paper. Please consider replacing this statement with 'is potentially at the onset of a major retreat phase' or similar.
It is correct that it is misleading in the context of dynamical system theory, but we never draw a link to dynamical system theory. The great works of Chris Doake often discussed the stability of ice shelves, also without any dynamical system theory context. However, we are happy to replace the text as suggested.

General comment about the discussion section: I'd like to echo Reviewer 2 and say I'm missing a more in-depth discussion about the model results. Besides the changes in GL flux, it would be interesting to explore the underlying reasons for the modelled behaviour. For example: if I understand correctly, your model is initialized with velocities prior to the formation/propagation of some of the rifts. Since you simulate a 5% increase in GL flux as a result of calving following the formation of the rifts, I wonder if that is consistent with observations?
You are right in saying that we initialise with velocities prior to the rift formation, but the first experiment calv2iceberg detaches also the ice bridge and this has not yet happened in reality, thus no observational comparison is possible at this stage. Indeed, once that happens the change in velocity will be the way to benchmark our results.

Can you also provide some further intuition about the changes in the buttressing parameters, e.g. what is the reason for the large areas of unbuttressed ice shelf in the calv2fjord experiment? Would one not expect shear contact with the lateral mountain slopes to provide some amount of buttressing?
The snapshots in time that we are looking at here, include (1) the initial state, a state in which the ice shelf is pinned along the calving front and (2) a state without pinning of the calving front. While in case 1 the velocity gradient in flow direction is negative - the velocity is getting lower towards the ice front, in case 2 the velocity towards the calving front is having little gradient. The change in lateral shear stress has less

influence, than the change in longitudinal stresses. The change in stress in along flow direction is what reduces buttressing.

In L102 you state that "Based on this variety of datasets we aim to investigate whether recent changes of the 79NG configuration are indicating a regime change, what exactly is the cause of the changes and how this will impact the stability of 79NG." yet I find little direct evidence in the text for the 'cause of the changes'. Are you referring to the ungrounding of the pinning point? If so, I think stress maps at different times before/after the ungrounding might help you identify how the glaciological conditions have changed, and how this has impacted on the preferred direction of fracture propagation.

We fully agree that this would be what one wants to see, but it is unfortunately not achievable. We have looked into every available velocity dataset, have tried to compute velocity fields from all missions we could get data from, but the coverage is poor and the scatter in the velocity fields from Landsat so large, that one has to smooth the fields that much, that one cannot use it for our purpose. During writing the original manuscript we have spent really a lot of energy on that and finally concluded that the one we present is the only dataset we can use for the stress field.

We think that in future, phase field modeling can be used to investigate this further, but currently phase field modeling for long time periods is still quite tricky. Sondershaus et al. (2023, https://doi.org/10.1002/pamm.202200256 ) is an example for such an approach, but lacks large deformations. However, that is the route to go to compare stress fields for different boundary conditions like grounded/ungrounded and crack formation. We will be working on this in future and hope to submit a manuscript on this in the next years.